# Diversity in Notch ligand-receptor signaling interactions

**Rachael Kuintzle[1], Leah A Santat[1,2], Michael B Elowitz[1,2]***

[1]Division of Biology and Biological Engineering, California Institute of Technology, Pasadena, United States; [2]Howard Hughes Medical Institute, California Institute of Technology, Pasadena, United States

## eLife Assessment

This **valuable** study significantly enhances our understanding of how various ligands and receptors interact within the Notch signaling pathway. By developing novel cell-based assay systems, the authors systematically analyzed the effects of different ligand-receptor combinations on pathway activation. The **convincing** data reveal intriguing and unexpected differences and provide a foundation for interpreting Notch signalling in both normal and disease-related contexts.

***For correspondence:**
melowitz@gmail.com

**Competing interest:** The authors declare that no competing interests exist.

**Abstract** The Notch signaling pathway uses families of ligands and receptors to transmit signals to nearby cells. These components are expressed in diverse combinations in different cell types, interact in a many-to-many fashion, both within the same cell (in cis) and between cells (in trans), and their interactions are modulated by Fringe glycosyltransferases. A fundamental question is how the strength of Notch signaling depends on which pathway components are expressed, at what levels, and in which cells. Here, we used a quantitative, bottom-up, cell-based approach to systematically characterize trans-activation, cis-inhibition, and cis-activation signaling efficiencies across a range of ligand and Fringe expression levels in Chinese hamster and mouse cell lines. Each ligand (Dll1, Dll4, Jag1, and Jag2) and receptor variant (Notch1 and Notch2) analyzed here exhibited a unique profile of interactions, Fringe dependence, and signaling outcomes. All four ligands were able to bind receptors in cis and in trans, and all ligands trans-activated both receptors, although Jag1-Notch1 signaling was substantially weaker than other ligand-receptor combinations. Cis-interactions were predominantly inhibitory, with the exception of the Dll1- and Dll4-Notch2 pairs, which exhibited cis-activation stronger than trans-activation. Lfng strengthened Delta-mediated trans-activation and weakened Jagged-mediated trans-activation for both receptors. Finally, cis-ligands showed diverse cis-inhibition strengths, which depended on the identity of the trans-ligand as well as the receptor. The map of receptor-ligand-Fringe interaction outcomes revealed here should help guide rational perturbation and control of the Notch pathway.

## Introduction

The Notch signaling pathway controls stem cell differentiation and proliferation, plays key roles in numerous diseases, and represents a major drug target. It uses multiple membrane-bound ligands and receptors that interact with one another in a many-to-many fashion, as well as Fringe glycosyltransferases (Fringes) that modulate those interactions. In mammals, the Notch pathway consists of four receptors (Notch1–4), four canonical activating ligands (Dll1, Dll4, Jag1, and Jag2), at least one predominantly inhibitory ligand (Dll3), and non-canonical ligands (*D'Souza et al., 2008*; *Falix et al., 2012*; *Fiddes et al., 2018*; *Gera and Dighe, 2018*; *Ladi et al., 2005*; *Serth et al., 2015*). Ligands and receptors interact both within the same cell (in cis) and between adjacent cells (in trans).

Either configuration has the potential to activate or inhibit signaling (*del Álamo et al., 2011*; *Nandagopal et al., 2019*; *Sprinzak et al., 2010*). The level of signaling in a Notch-expressing cell generally depends on which ligand, receptor, and Fringe variants are expressed in the cell and its neighbors. These components are expressed in various combinations in different cell types, often including coexpression of ligands, receptors, and at least one Fringe enzyme (*Granados et al., 2022*). However, it remains difficult to predict signaling strength—how strongly a given cell will signal to another cell—based on the expression profiles of Notch pathway components. It similarly remains challenging to rationally and predictably perturb signaling for therapeutic and tissue engineering purposes.

Transcriptional responses are sensitive to the amplitude and duration of Notch signaling (*Falo-Sanjuan et al., 2019*; *Kuang et al., 2020*; *Nandagopal et al., 2018*). In the canonical trans-activation mechanism, binding of ligands on one cell to receptors on an adjacent cell triggers ligand endocytosis, which generates mechanical strain on the receptor, exposing a metalloproteinase recognition site (*Langridge and Struhl, 2017*; *Lovendahl et al., 2018*). S2 cleavage by ADAM10 results in shedding of the receptor extracellular domain (NECD) and permits a subsequent S3 cleavage by γ-secretase to release the Notch intracellular domain (NICD). The free NICD directly translocates into the nucleus and binds cofactors MAML and RBPjκ to activate target genes. Downstream target genes respond similarly to NICD originating from the Notch1 or Notch2 receptors (*Kraman and McCright, 2005*; *Liu et al., 2015*; *Liu et al., 2013*). However, they respond differently to distinct concentrations of NICD. In pancreatic progenitors (*Ninov et al., 2012*) and central nervous system stem cells (*Guentchev and McKay, 2006*), complete Notch inhibition permitted differentiation, low levels of NICD promoted proliferation, and higher levels induced quiescence. Notch signaling amplitude can also influence the timing of developmental transitions, with higher NICD concentrations activating master transcription factors earlier than lower concentrations, depending on the target's enhancer architecture (*Falo-Sanjuan et al., 2019*).

The Notch pathway provides numerous ways to tune signaling amplitude. Fringe enzymes can alter receptor-ligand binding and activation strengths, sometimes in opposite directions (*Hicks et al., 2000*; *Kakuda et al., 2020*; *Kakuda and Haltiwanger, 2017*). Upregulating expression of a ligand can suppress Notch signaling cell-autonomously, through a process termed cis-inhibition (*Becam et al., 2010*; *Fiuza et al., 2010*; *Sprinzak et al., 2010*; *Thambyrajah et al., 2024*), or trans-activate receptors on neighboring cells. In some cases, ligands can also activate signaling by receptors in the same cell (*Nandagopal et al., 2019*). Finally, growing evidence suggests that some ligands can inhibit signaling intercellularly ('trans-inhibition') by binding receptors strongly without activating them (*Benedito et al., 2009*; *Golson et al., 2009*; *Luna-Escalante et al., 2018*), as may be the case for Jag1 and Lunatic Fringe (Lfng)-modified Notch1 (*Hicks et al., 2000*; *Kakuda and Haltiwanger, 2017*). Thus, the Notch pathway architecture allows for receptor-ligand interactions to result in either activation or inhibition of signaling, in either cis or trans. However, for the majority of possible receptor-ligand interaction pairs, the relative activation and inhibition strengths have not been measured in both cis and trans orientations, making it difficult to predict signaling outcomes in natural contexts and in applications such as tissue engineering.

Here, to address these challenges, we developed a set of engineered cell lines and coculture reporter assays that allow systematic characterization of trans-activation, cis-inhibition, and cis-activation efficiencies across a range of cis-ligand and Fringe expression levels. We focused on interactions among four ligands—Dll1, Dll4, Jag1, and Jag2—with two receptors—Notch1 and Notch2—and their modulation by Lfng. We verified that key features are consistent between two distinct cell lines: Chinese hamster ovary (CHO-K1) fibroblasts and C2C12 myoblasts. Each receptor and ligand had a unique profile of Lfng-dependent cis- and trans-interactions with other components. In trans, all ligands were capable of activating Notch1 and Notch2. However, Jag1 trans-activated Notch1 inefficiently, despite the strong Notch1-activating ability of recombinant Jag1 fragments. In cis, with competition from trans-activating ligands, Dll1 and Dll4 inhibited Notch1 and further activated Notch2 signaling, while Jagged ligands cis-inhibited both receptors. Lfng modulated most receptor-ligand interactions. It increased Jag1's inhibitory potential by strengthening binding but weakening activation of Notch1, as seen previously (*Hicks et al., 2000*; *Kakuda and Haltiwanger, 2017*; *Taylor et al., 2014*; *Yang et al., 2005*). In addition, Lfng potentiated trans-activation for Delta-Notch combinations, but attenuated Jagged-Notch signaling. It also had diverse receptor- and ligand-specific effects on cis-activation, which in general differed from those on trans-activation for Notch2 but not for Notch1. Together, this

map of relative cis and trans receptor-ligand-Fringe interactions should be useful to explain Notch signaling behaviors in diverse developmental and physiological contexts, and guide more rational, targeted perturbation of Notch signaling activity.

## Results

### Engineered 'receiver' and 'sender' cell lines enable quantitative comparison of receptor-ligand-Fringe interactions

To systematically analyze pairwise cis and trans receptor-ligand interactions, and their dependence on Fringe enzyme expression, we engineered a set of over 50 different stable cell lines (Key resources table). Collectively, they provide quantitative readouts of Notch signaling activity, receptor level, and ligand level. They also enable precise modulation of ligand expression. As a base cell line, we used CHO-K1 cells, which exhibit negligible endogenous expression levels of Notch receptors and ligands, and no endogenous Notch signaling activity (*Supplementary file 1*; *Singh et al., 2018*; *Sprinzak et al., 2010*). In this background, we constructed three types of cell lines, described below.

First, to read out Notch signaling, we created monoclonal 'receiver' cells, similar to those described previously (*LeBon et al., 2014*; *Nandagopal et al., 2019*; *Sprinzak et al., 2010*; *Figure 1A*). These cells expressed a chimeric human Notch receptor, whose intracellular domain was replaced with a minimal Gal4 transcription factor ('Gal4esn', henceforth denoted Gal4). This coding sequence was followed by H2B-mTurq2, with an intervening ribosomal skipping T2A sequence, for cotranslational readout of receptor expression. This construct was stably integrated in the host cell genome using piggyBac transposition (Methods). Expression of the cotranslational mTurq2 reporter was broadly comparable between different receiver clones (*Figure 1—figure supplement 1A*), and correlated with surface receptor expression (*Figure 1—figure supplement 1B*). The cells also contained an insulated UAS promoter driving expression of H2B-mCitrine, such that mCitrine production reflects Notch activity.

Second, to analyze same-cell (cis) ligand-receptor interactions, we used lentivirus to stably integrate each of the four activating human Notch ligands (Dll1, Dll4, Jag1, Jag2) into the receiver cells (*Figure 1A*). All ligand constructs contained a T2A cotranslational H2B-mCherry reporter for readout of expression. Ligand expression was controlled by the Tet-OFF system, allowing the use of the doxycycline analog 4-epi-tetracycline (4-epi-Tc) to titrate ligand levels. In these lines, varying the concentration of 4-epi-Tc tuned expression unimodally across two orders of magnitude (*Figure 1B*, *Figure 1—figure supplement 2*). To control for non-specific effects of ligand overexpression, we also constructed parallel negative control cell lines expressing H2B-mCherry or human nerve growth factor receptor (NGFR)-T2A-H2B-mCherry constructs in place of ligands. NGFR has been used in other Notch studies as a surface-detectable coexpression reporter or marker (*Del Real and Rothenberg, 2013*; *Romero-Wolf et al., 2020*; *Sakata-Yanagimoto et al., 2008*; *Taghon et al., 2009*). These cell lines enabled control and readout of cis-ligand levels.

Third, we engineered a repertoire of 'sender' cell lines (*Figure 1A*). One set of these cell lines enabled inducible expression of each of the four ligands under control of the Tet-OFF system (*Figure 1—figure supplement 3*, Methods). These lines allowed control of ligand expression in cells without Notch receptors. A second set provided constitutive expression of each ligand at a variety of different levels (*Figure 1C*).

Because cotranslational protein abundance is limited as a proxy for surface protein, this study focuses primarily on the relationship between signaling activity and the total amount of translated receptor or ligand. However, we note that high signaling activity need not, in general, directly correlate with surface expression levels; e.g., factors such as vimentin decrease basal surface of Jag1 but increase its activity (*Antfolk et al., 2017*).

To limit combinatorial complexity, we focused on two essential receptors (Notch1 and Notch2), four canonical activating ligands (Dll1, Dll4, Jag1, and Jag2), and one Fringe (Lfng). We omitted Notch3, since Notch3 knockout mice are viable with minor vascular defects (*Kitamoto et al., 2005*; *Krebs et al., 2003*), and because Notch3 receptors are hypersensitive to activation during cell passaging, yielding elevated background signaling. We also omitted Notch4, which has no knockout phenotype in mice (*Krebs et al., 2000*) does not appear to activate in coculture with ligand-expressing cells (*Groot et al., 2014*; *James et al., 2014*; *Lafkas et al., 2015*), and has been suggested to inhibit

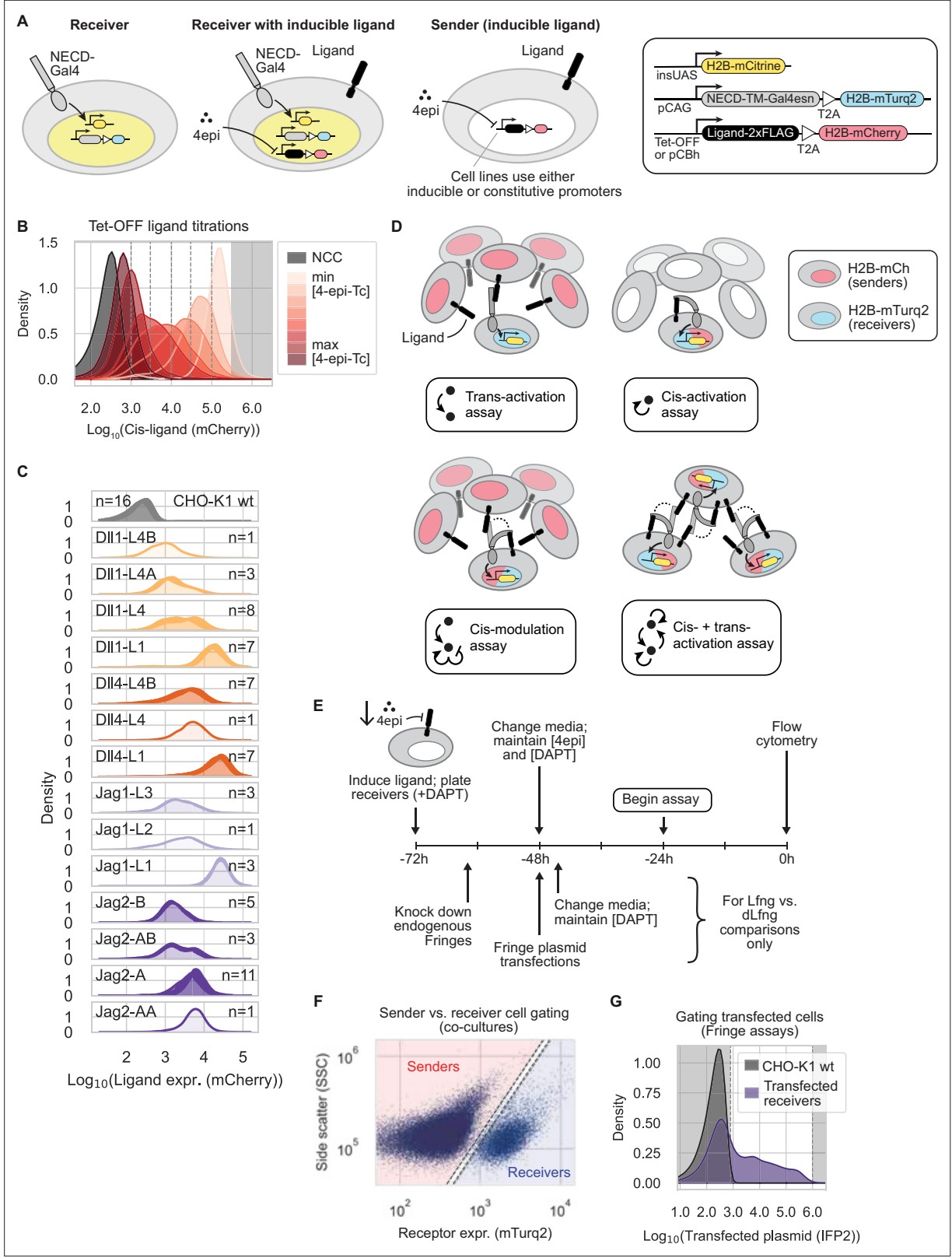

**Figure 1.** Engineered 'receiver' and 'sender' cells enable quantitative comparison of receptor-ligand interactions. (**A**) Engineered cell lines enable systematic analysis of trans-activation, cis-activation, and cis-inhibition. Receiver cells contain chimeric Notch receptors (gray) composed of Notch extracellular and transmembrane domains fused with a minimal Gal4 transcription factor (Gal4) in place of the endogenous intracellular domain, co-transcribed with a T2A-H2B-mTurq2 cassette for quantitative readout of receptor expression. Receptor activation releases Gal4, which activates

*Figure 1 continued on next page*

*Figure 1 continued*

expression of a stably integrated UAS-H2B-mCitrine reporter (yellow). Some strains also contained a stably integrated ligand and reporter (second cartoon). These sender cell lines were constructed by integrating plasmids containing each of the four activating ligands fused to 2xFLAG, followed by T2A-H2B-mCherry for cotranslational readout of ligand expression, either under Tet-OFF control (to allow induction by 4-epiTc) or expressed from the constitutive CBh promoter. (**B**) Example distributions of Tet-OFF inducible ligand expression in CHO-K1 cells, read out by fluorescence of a cotranslational H2B-mCherry (A.U.). Black histograms are CHO-K1 wild-type cells (NCC = no color control). Dotted gray vertical lines mark binning windows used in some analyses (see also Methods). Data points in the gray-shaded region were omitted to avoid overexpression artifacts (see Methods). (**C**) Single-cell histograms (kernel density estimates) of stably expressed ligand levels (read out by cotranslational H2B-mCherry, A.U.) in the CHO-K1 sender populations used for trans-activation assay experiments. n denotes number of replicates per plot. Sender populations are named with the ligand expressed and a population identifier (e.g. 'L4B'). (**D**) Schematics of the main assays used in this work. Each panel shows the cocultured cell types and their relative population sizes (majority or minority). In the cis-+trans-activation assay, dotted lines indicate alternative receptor interactions. See also Methods. Dot-and-arrow icons are used to identify assays in subsequent figures. (**E**) Experimental workflow for cell culture experiments with flow cytometry readout. Ligand expression is preinduced in either sender or receiver cells by reducing the 4-epi-Tc concentration in the culture medium to induce ligand expression to a given level. Receivers are incubated in the Notch signaling inhibitor DAPT to prevent reporter activation during this preinduction phase. Prepared cells, which may also undergo siRNA knockdown and/or plasmid transfection during the ligand preinduction phase, are replated without DAPT according to the chosen experimental scheme in (**D**) and allowed to signal for 22–24 hr before cells are detached and analyzed by flow cytometry. (**F–G**) Data plots show example flow cytometry data processing. Gates used in data processing are shown as dashed lines. (**F**) Senders and receivers are separated computationally in the flow cytometry data in a 2D plane of cell size (side scatter, A.U.) vs. cotranslational receptor expression (mTurquoise2, A.U.). (**G**) Plasmid-transfected cells are gated on fluorescence levels of a cotransfected infrared fluorescent protein (IFP2, A.U.). Data points in gray-shaded regions were discarded (Methods).

The online version of this article includes the following figure supplement(s) for figure 1:

**Figure supplement 1.** Receptor expression in CHO-K1 receiver clones.

**Figure supplement 2.** The Tet-OFF system enables unimodal titration of ligand levels in receiver cells.

**Figure supplement 3.** The Tet-OFF system enables stable ligand expression in sender cells.

**Figure supplement 4.** Knockdown of endogenous Lfng and Rfng in CHO-K1 cells.

**Figure supplement 5.** Normalizing reporter activity to receptor expression addresses spurious dependence of signal on cis-ligand controls.

Notch1 activation in cis (*James et al., 2014*). Among the best-studied ligands, we omitted Dll3, which is essential and believed to be purely cis-inhibitory (*Bochter et al., 2022*). Finally, we focused on Lunatic Fringe (Lfng), which we perturbed using siRNAs and transient plasmid transfection, because it is the only Fringe enzyme that is essential in mice (*Evrard et al., 1998*; *Moran et al., 2009*; *Zhang et al., 2002*), and its effects dominate over those of Radical and Manic Fringe (Rfng and Mfng, respectively) when coexpressed (*Pennarubia et al., 2021*). Together, these cell lines enabled systematic analysis of trans-activation, cis-activation, and cis-inhibition (*Figure 1D*, Methods).

## Trans-activation strength depends on ligand and receptor identity, and is modulated by Lfng

A fundamental question about the Notch system is how signaling strength depends on the identity of the interacting ligand and receptor, and expression of Fringe enzymes. Previous investigations reached conflicting conclusions about how strongly and even in what direction Fringes affect different ligand-receptor interactions (*Hicks et al., 2000*; *Kakuda et al., 2020*; *Shimizu et al., 2001*; *Yang et al., 2005*). They also focused on only a subset of essential ligand-receptor combinations, omitted analysis of Fringe dependence (*Tveriakhina et al., 2018*), and in some cases used plate-bound ligands rather than ligands expressed by cells (*Kakuda et al., 2020*).

To address these gaps, we used a trans-activation assay (*Figure 1D*) to systematically measure trans-signaling across all eight receptor-ligand combinations, under controlled Fringe expression conditions (Methods). We first suppressed endogenous Rfng and Lfng expression via siRNA knockdown in receiver cells (*Figure 1E*; *Figure 1—figure supplement 4A*; *Supplementary file 1*). Approximately 16 hr later, we removed siRNAs and transfected plasmids encoding wild-type (wt) mouse Lfng or, as a negative control, a catalytically inactive mutant Lfng (D289E, denoted 'dLfng') (*Luther et al., 2009*). We also cotransfected a plasmid expressing infrared fluorescent protein (IFP2) as a transfection marker. After recovering for 16–20 hr post-transfection, we cocultured these receiver cells with an excess of sender cells, previously sorted into bins of stable ligand expression (*Figure 1C*). Finally, after 22–24 hr of coculture, we measured Notch activity in receivers by flow cytometry, gating on mTurq2,

the reporter of Notch expression, to separate senders and receivers (*Figure 1F*), and gating on IFP2 to enrich for Fringe plasmid-transfected cells (Methods, *Figure 1G*, *Figure 1—figure supplement 4B*).

To quantify signaling, we first defined signaling activity as reporter fluorescence (mCitrine) normalized by receptor expression, as read out by mTurq2 fluorescence (*Figure 1A*, *Figure 1—figure supplement 5*). To control for variation in ligand expression across sender populations (*Figure 1C*), we further normalized this signaling activity by ligand expression, read out by a distinct cotranslational fluorescent protein (mCherry) (*Figure 2—figure supplement 1A*), similar to an approach used previously (*Tveriakhina et al., 2018*). The resulting receptor- and ligand-normalized signaling strengths varied widely across the 16 Notch-ligand-Fringe combinations (*Figure 2A*, *Figure 2—figure supplement 1B*).

Comparing signaling between the two receptors revealed two key features of trans-signaling: First, almost all ligands signaled more strongly to Notch2 than to Notch1, regardless of Fringe expression (*Figure 2A*). The exception was Dll4, which signaled more strongly to Notch1 than Notch2 in the dLfng condition, but activated Notch1 and Notch2 to similar levels with Lfng. Second, Notch1 and Notch2 responded in a qualitatively similar way to Lfng, as can be seen by plotting signaling strengths for the Lfng condition vs. those for dLfng (*Figure 2B and C*). Lfng significantly enhanced Notch1 and Notch2 trans-activation by both Dll1 and Dll4. Its greatest effect was on Dll1-Notch1 (3-fold increase) followed by Dll4-Notch2 (2.5-fold increase). Lfng significantly decreased trans-activation of both receptors by Jag1 (>2.5-fold) and, to a lesser extent, Jag2 (~1.4-fold). (Note that while Jag1-Notch1 signaling was low, it was still possible to detect further reductions (*Figure 2C*).) Thus, the two receptors differed in their responses to the four ligands but responded similarly to Lfng, which strengthened Delta-mediated trans-activation and weakened Jagged-mediated trans-activation for both receptors.

Next, we focused on the difference in trans-activation by Dll1 and Dll4 in the two Fringe conditions. In the absence of Fringe expression (i.e. with dLfng), Dll1-Notch1 signaling strength was only slightly above background (*Figure 2A*). By contrast, Dll4 activated Notch1 9-fold more strongly than Dll1 in the dLfng condition, consistent with previous results from E14TG2a mouse embryonic stem cells (*Tveriakhina et al., 2018*), possibly reflecting the ~10-fold greater binding affinity of Notch1 to the Dll4 extracellular domain (ECD), compared to the Dll1 ECD (*Andrawes et al., 2013*). Lfng expression increased Dll1 signaling more than Dll4, reducing the difference in signaling strengths between the ligands to 2.7-fold (*Figure 2A*). With Notch2, Dll1 signaling exceeded Dll4 signaling, with or without Lfng.

Similar to the Delta ligands, the Jagged ligands also showed diverse signaling activities. Strikingly, Jag1 activated Notch1 poorly in both the dLfng and Lfng conditions (*Figure 2A*). This lack of signaling was not due to a defect in the ligand, which was properly trafficked to the cell surface (*Figure 2—figure supplement 2*) and which signaled to Notch2 at levels comparable to those of Dll1 in the dLfng condition (*Figure 2A*).

In contrast to Jag1, Jag2 was the strongest trans-activating ligand for both receptors with dLfng, and remained among the strongest with Lfng. Taken together, these results establish unique patterns of activity for the four ligands across different receptor and Fringe contexts.

In addition to signaling strength, another key feature of signaling is ultrasensitivity, which plays a pivotal role in developmental patterning circuits (*Gozlan and Sprinzak, 2023*; *Sprinzak et al., 2011*; *Yasugi and Sato, 2022*). Ultrasensitivity could in principle emerge from clustering of Notch ligands and receptors (*Bardot et al., 2005*; *Cattoni et al., 2015*; *Duke and Graham, 2009*; *Gopalakrishnan et al., 2005*; *Narui and Salaita, 2013*; *Radhakrishnan et al., 2012*; *Tetzlaff et al., 2018*). To quantify ultrasensitivity, we titrated ligand expression in 4-epi-Tc-inducible sender cells (*Figure 1—figure supplement 3A*) and measured signaling activity in a cocultured minority of CHO-K1 receiver cells (*Figure 2D*). Signaling levels increased monotonically to levels that varied by ~1.5-fold across most ligands (*Figure 2E*). However, there were two exceptions: First, Dll1-Notch signaling was biphasic, declining at high trans-ligand expression levels (*Figure 2D*). Second, Jag1-Notch1 signaling was much weaker than other signaling interactions, as expected, preventing analysis of ultrasensitivity in this case. Most dose-response curves showed ultrasensitive responses in which signaling activity increased approximately as the square of the ligand concentration (*Figure 2F*, *Figure 2—figure supplement 3*). The exception was Jag1-Notch2 signaling, with a logarithmic sensitivity of ~1.5. An independent analysis based on fitting Hill functions to the subset of dose-response curves that reached saturation similarly produced Hill coefficients of ~2 for Dll4 with both receptors and Jag1-Notch2 (*Figure 2G*).

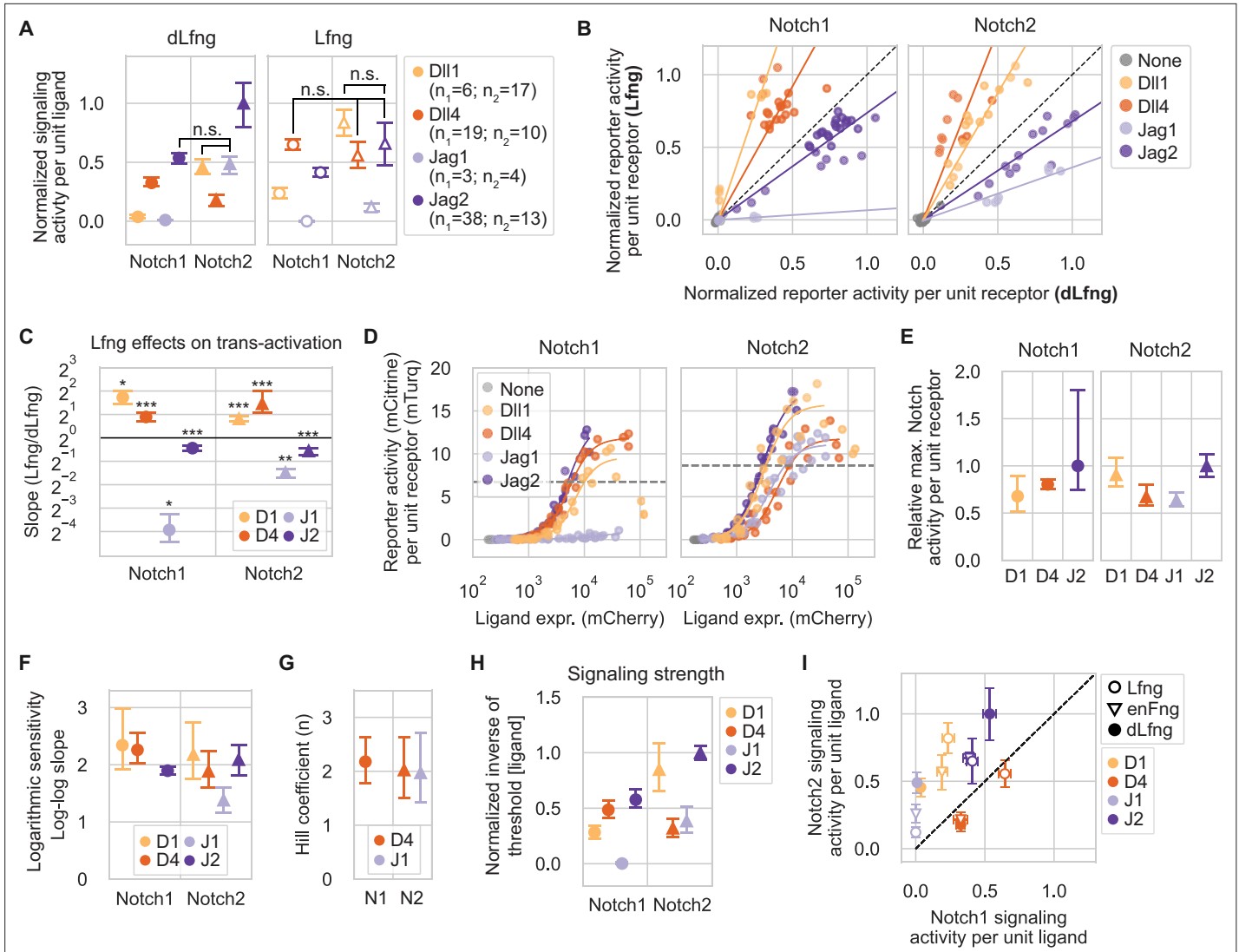

**Figure 2.** Trans-activation properties depend on ligand and receptor identity, and are modulated by Lfng. (**A**) Mean trans-activation signaling strength for different ligand-receptor-Lfng combinations. In each case, expression of the mCitrine reporter was normalized by the cotranslational Notch reporter (mTurq2) fluorescence. These values were averaged across all single cells in each sample, then background subtracted and further normalized by the strongest signaling activity measured for each receiver clone in the experiment. Normalized signaling activities for sub-saturated data points (Methods) were further normalized to the mean expression of the sender population (**Figure 1C**) in each coculture (Methods), and values are relative to the overall maximum. Data are from five different Notch1 receiver clones and three different Notch2 clones, with at least three biological replicates per clone (**Supplementary file 3**). Cis-ligand was suppressed with high 4-epi-Tc concentrations in receivers with integrated ligands. The significance of pairwise differences in normalized signaling activity was evaluated through permutation testing (see Methods). Except where labeled with 'n.s.', all signaling strengths within each subplot are significantly different (p-val<0.05) from all other receptor-ligand combinations within the same Lfng or dLfng condition. See **Figure 2—figure supplement 1B** for more granular analysis of p-values. Colors indicate the identity of the trans-ligand expressed by cocultured sender cells. Receptor identity is indicated below plot and by marker type (Notch1=circles, Notch2=triangles). Here and in subsequent panels, error bars denote bootstrapped 95% confidence intervals (Methods), in this case sampled from the number of bioreplicates given in the legend—$n_1$ (for Notch1) or $n_2$ (for Notch2). (**B**) Effects of Lfng on Notch1 (left) and Notch2 (right) signaling. Non-ligand-normalized signaling activities were re-plotted in dLfng (x-axis) vs. Lfng (y-axis) conditions. Saturated data points, defined here as those with normalized signaling activity over 0.75 in both dLfng and Lfng conditions, were excluded. Colored lines are least-squares linear fits. Black dashed line indicates no effect of Lfng expression. (**C**) Plotting the mean slope in (**B**) from bootstrap analysis reveals the effects of Lfng on trans-signaling for the indicated ligand-receptor combinations (Methods). Asterisks denote the p-value of the test statistic from a one-sided Wilcoxon signed-rank test (*p-val<0.05; **p-val<0.01; ***p-val<0.001), reflecting whether the slope is greater or less than 1. Receiver identity is indicated by x-axis labels and marker type. Ligand identity is indicated by color and the following abbreviations, which are used here and in subsequent figures: D1=Dll1, D4=Dll4, J1=Jag1, J2=Jag2. (**D**) Trans-activation dose-response curves for CHO-K1 Notch1 or Notch2 receivers, expressing endogenous Fringes activated by a cocultured majority of the Tet-OFF inducible sender cell lines shown in **Figure 1—figure supplement 3A**. Colors indicate ligand identity as in (**C**). X-axis values are the mean of the mCherry fluorescence

*Figure 2 continued on next page*

*Figure 2 continued*

in senders used for each coculture sample data point. Y-axis signaling activity values are the mean of the distribution in mCitrine (reporter activity, A.U.) divided by mTurq2 (cotranslational receptor expression, A.U.). Solid lines are fits of the increasing phase of each dataset to activating Hill functions (Methods). Dotted gray horizontal lines represent half-maximal signaling activity for each receptor across all ligand inputs. (E) Mean saturating signaling activities for indicated ligand-receptor combination, estimated by bootstrapping the Hill fits in (D), and normalized by the response to Jag2 for the same receptor. Although the decreasing phases of the Dll1 curves were excluded, saturating activity estimates may be influenced by incomplete saturation and biphasic behavior. (F) Mean logarithmic sensitivities of signaling response to ligand expression from bootstrapped linear regressions to log-log transformed, sub-saturating data points (see also *Figure 2—figure supplement 3*). (G) Mean Hill coefficients (**n**) estimated from bootstrap analysis of apparently saturating ligand-receptor responses in (D). (H) Relative signaling strength (ligand potency) was computed by estimating the ligand expression level sufficient to reach threshold activity level (where Hill fits crossed dotted lines in 2D) for each ligand-receptor combination, inverting it, and further normalizing all values by the value obtained for Jag2-Notch2. Mean values and 95% confidence intervals were computed from bootstrapped Hill function fitting (Methods). (I) Comparison of signaling strengths for Notch2 vs. Notch1 receivers in the dLfng, Lfng, and endogenous Fringe (enFng) backgrounds. dLfng and Lfng data are identical to values plotted in (A), and enFng values are identical to values plotted in (H), but scaled such that Dll4-Notch1 signaling strengths match in dLfng and enFng (based on *Figure 2—figure supplement 4B*).

The online version of this article includes the following source data and figure supplement(s) for figure 2:

**Figure supplement 1.** Statistical analysis of differences in trans-activation strength for all receptor-ligand-Fringe combinations.

**Figure supplement 2.** Quantification of surface ligand in CHO-K1 sender clones.

**Figure supplement 2—source data 1.** Original files for western blots displayed in *Figure 2—figure supplement 2B*.

**Figure supplement 2—source data 2.** PDF file containing original western blots for *Figure 2—figure supplement 2B*, indicating the relevant bands and sizes.

**Figure supplement 3.** Notch trans-activation in coculture is modestly ultrasensitive.

**Figure supplement 4.** Endogenous Lfng activity dominates over Rfng in CHO-K1 cells.

---

The dose-response curves also allowed analysis of the relative activation strength of different ligands, defined by the ligand concentration required to signal above a threshold (Methods). Signaling strengths determined this way interpolated between the values described above in Lfng and dLfng conditions (*Figure 2I*), likely reflecting the endogenous Fringe profile of CHO-K1 cells, which express Rfng as well as low levels of Lfng, a profile whose effects are largely consistent with low levels of Lfng (*Figure 2—figure supplement 4*, Methods).

Together, these results show that Notch signaling is modestly ultrasensitive for most ligand-receptor pairs in this cell context and, more broadly, reveal quantitative receptor-ligand preferences and their dependence on Lfng expression.

## Plated Jag1 ligands activate Notch1 more efficiently than expressed Jag1 ligands

Plate-bound recombinant ligands provide a convenient method to assay Notch signaling (*Kakuda et al., 2020*), but differ from cell-expressed ligands in their Notch activation mechanism. We therefore sought to determine whether the method of ligand presentation (plate-bound or expressed by cells) affects ligands' signaling properties, such as signaling strength and ultrasensitivity. We analyzed CHO-K1 receiver cells' responses to titrated concentrations of plated recombinant ligand ECD, tagged at the C-terminus by a human Fc domain ('ligand-ext-Fc'), in a plated ligand assay (*Figure 3A and B*). As with the expressed ligand dose-response analysis, receiver cells expressed endogenous CHO-K1 Fringes.

For all ligand-receptor combinations, signaling increased monotonically with plated ligand levels across the utilized concentration range. Because factors such as partial denaturation of recombinant proteins preclude absolute measurements of effective ligand concentrations, we compared the relative activity of individual ligands across different receptors (*Figure 3C*), but did not compare activation of the same receptor by different ligands. We again defined the relative activation strengths by the ligand concentration required to signal above a threshold (Methods). Receptor preferences were qualitatively, and sometimes quantitatively, consistent between the plated ligand assay and trans-activation coculture assay (*Figure 3D*). For example, Jag2 and Dll1 exhibited ~2- and ~3-fold stronger activation, respectively, for Notch2 than Notch1 in both assays. Similarly, Dll4 showed a slight preference for Notch1 over Notch2 in both assays, although the fold difference was larger in the plated ligand assay. Strikingly, while Jag1 sender cells failed to activate Notch1 receivers above background

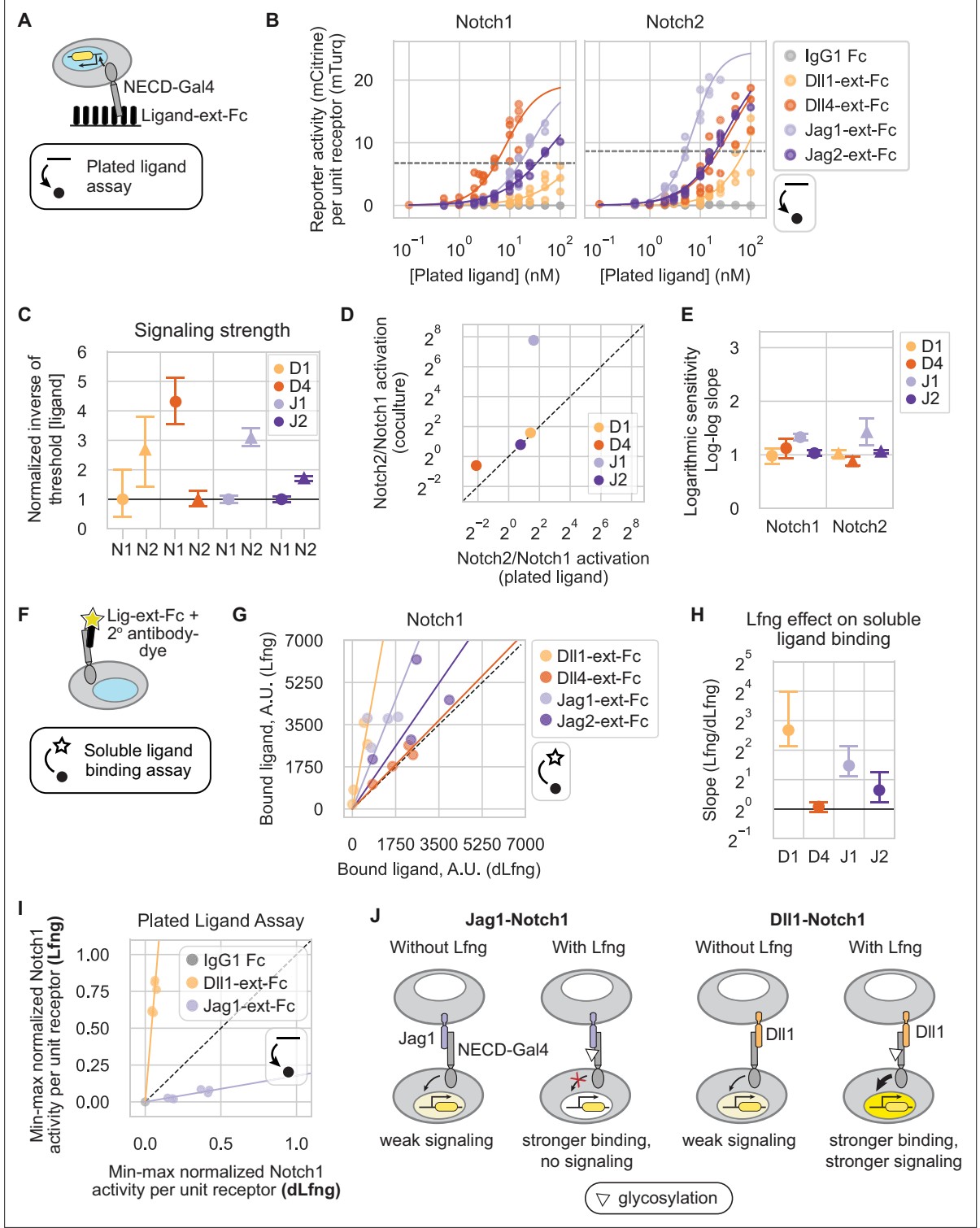

**Figure 3.** Signaling properties of recombinant ligands. (**A**) Plated ligand assay records Notch receivers' responses to plated recombinant human C-terminal Fc-tagged ligand extracellular domains ('ligand-ext-Fc') (schematic). Blue in the cell nucleus represents H2B-mTurq2 fluorescence (readout of receptor expression), and the yellow construct represents the mCitrine reporter promoter. Line-dot-and-arrow icon refers to this assay. (**B**) CHO-K1 Notch1 (left) and Notch2 (right) receivers expressing endogenous Fringes were cocultured on plated recombinant ligand-ext-Fc proteins at the concentrations indicated on the x-axis. Y-axis signaling activity values are the mean of mCitrine reporter distributions (reporter activity, A.U.) divided by mTurq2 (cotranslational receptor expression, A.U.). Solid lines are activating Hill function fits, with free Hill coefficient (**n**) and EC50 parameters. Saturating activities were fixed since the curves did not fully saturate in these ligand concentration regimes (Methods). Dotted gray horizontal lines

*Figure 3 continued on next page*

*Figure 3 continued*

represent the signaling thresholds defined in *Figure 2D*. (**C**) Mean signaling strengths, based on bootstrap analysis of the responses in (**D**) (Methods). Each signaling strength is defined as the inverse of the ligand concentration sufficient to reach threshold activity level (dotted lines in (**B**)), normalized to show receptors' relative activities. Colors and labels indicate ligand identity, as in *Figure 2*. X-axis labels are receivers; 'N' = 'Notch.' Here and in subsequent panels, error bars denote bootstrap 95% confidence intervals (Methods). (**D**) Comparison of mean Notch2/Notch1 signaling strength ratios with canonical trans-activation in sender-receiver cell cocultures (y-axis, values from *Figure 2H*) vs. the plated ligand assay (x-axis, values from (**C**)). (**E**) Mean logarithmic sensitivities computed from the slope of linear regressions to 10,000 bootstrap replicates of log-log sub-saturating signaling activities vs. ligand concentrations in *Figure 3—figure supplement 1*. (**F**) Soluble ligand binding assay (schematic, see also Methods), which enables the quantification of the strength of receptor binding to ligand-ext-Fc pre-clustered with secondary antibody. Blue in the cell nucleus represents H2B-mTurq2 fluorescence (readout of receptor expression). Star-dot-and-arrow icon refers to this specific assay. (**G**) Scatterplot of averaged single-cell data from the soluble ligand binding assay with Lfng (y-axis) vs. dLfng (x-axis) expression (Methods). Fluorescence background, determined as ligand bound to parental reporter cells with no ectopic Notch receptors, was subtracted, and negative values were set to zero. Solid lines are least-squares best fits, and the black dashed line is y=x. (**H**) Mean fold difference in the amount of each ligand bound to Notch1 with Lfng vs. dLfng from slopes of linear regressions in (**G**), based on bootstrap analysis of n=4 biological replicates per ligand. X-axis labels are recombinant ligands. (**I**) Normalized Notch signaling strength in CHO-K1 Notch1 receivers expressing Lfng or dLfng, plated on Dll1-ext-Fc (yellow) or Jag1-ext-Fc (purple) in a plated ligand assay (Methods). X- and y-axis values represent mean signaling activity (reporter activity, mCitrine, divided by cotranslational receptor expression, mTurq2), background subtracted and normalized to the maximum signaling activity—the average signal in the same receiver (+Lfng) plated on a high concentration of Dll4-ext-Fc. Solid lines are the least-squares best fits to six data points, representing three biological replicates for each of two plated ligand concentrations. Both slopes were significantly greater than or less than 1 according to a one-sided Wilcoxon signed-rank test (p-value = 0.015 for both lines). The black dashed line is y=x. Bracketed numbers are bootstrap 95% confidence intervals. (**J**) Cell schematic depicting the effects of Lfng expression on Jag1 and Dll1 interactions with the Notch1 receptor. White triangles represent glycosylation modifications added by Lfng, and yellow saturation level in cell nuclei represents signaling activity.

The online version of this article includes the following figure supplement(s) for figure 3:

**Figure supplement 1.** Notch activation by plated ligands is not ultrasensitive.

(*Figure 2D*), plate-bound Jag1-ext-Fc activated Notch1 only ~3-fold less efficiently than it activated Notch2 (*Figure 3B–D*). This suggests that the natural endocytic activation mechanism, or potential differences in tertiary structure between the expressed and recombinant Jag1 ECD, could play roles in preventing Jag1-Notch1 signaling in coculture.

Finally, in contrast to the ultrasensitive responses to expressed ligands, dose-response curves in the plated ligand assay were approximately linear for all ligand-receptor combinations except Jag1, which showed logarithmic sensitivities closer to ~1.5 for both receptors (*Figure 3E*, *Figure 3—figure supplement 1*). These differences in ultrasensitivity may reflect a difference in clustering behaviors between plated ligand-ext-Fc ligands and cell-expressed ligands (*Cattoni et al., 2015*). Taken together, these results indicate that plate-bound ligands differ in their Notch activation compared to ligands expressed on living cells.

## Ligand binding is sensitive to Fringe expression

Despite their different signaling properties, recombinant ligands provide a convenient system to measure receptor-ligand binding interactions. Previous work has shown that the effects of Lfng on binding and signaling are not directly correlated. Lfng increased recombinant Jag1 binding to Notch1, but reduced its signaling (*Hicks et al., 2000*; *Kakuda and Haltiwanger, 2017*; *Taylor et al., 2014*; *Yang et al., 2005*). On the other hand, Lfng strengthened Dll1-Notch1 binding and increased signaling. In another study, Lfng reduced Jag1-Notch2 binding (*Shimizu et al., 2001*). However, Lfng effects on the binding strength of most receptor-ligand combinations remain unknown.

Here, we focused on Lfng effects on relative ligand-receptor binding strengths for the four Notch1-ligand combinations. We incubated CHO-K1 receiver cells, or parental reporter cells expressing only endogenous receptors, with soluble ligand-ext-Fc fragments pre-clustered with a dye-conjugated antibody, as previously described (*Kakuda and Haltiwanger, 2017*; *Varshney and Stanley, 2017*; *Figure 3F*). We then used flow cytometry to measure the amount of ligand bound to cells, transiently expressing either Lfng or dLfng.

In agreement with the previous work, Lfng strengthened Notch1 binding to Dll1 and Jag1 by 6.4-fold and 2.8-fold on average, respectively (*Figure 3G and H*). Lfng also modestly strengthened Notch1 binding to Jag2. However, Lfng had relatively little effect on Dll4-Notch1 binding. These data do not rule out the possibility that other ligand concentration regimes might be more sensitive to Lfng.

In parallel with binding, we also analyzed the effect of Lfng on signaling by plate-bound ligands (*Figure 3A*). We incubated dLfng- or Lfng-transfected Notch1 receivers on plated Dll1-ext-Fc and Jag1-ext-Fc. Lfng strongly increased Dll1-Notch1 signaling and weakened Jag1-Notch1 signaling in the plated ligand assay (*Figure 3I*), similar to results with expressed ligands (*Figure 2B and C*). Thus, Lfng had opposite effects on binding and activation of Notch1 by the same Jag1-ext-Fc fragment. By strengthening Jag1-Notch1 binding while decreasing activation, Lfng may allow Jag1 to competitively trans-inhibit Notch1 activation by other ligands (*Figure 3J*; *Benedito et al., 2009*; *Golson et al., 2009*; *Pedrosa et al., 2015*).

While they do not rule out additional effects of Lfng on binding affinity for Dll4-Notch1, these results together suggest that with Dll1, Lfng strengthens both binding and activation of Notch1, while with Jagged ligands it strengthens binding but instead weakens activation.

## Cis-interaction outcomes depend on receptor and ligand identity

Activating ligands are frequently coexpressed with Notch1 and/or Notch2, provoking the question of how ligands and receptors interact in the same cell (in cis), and more specifically whether they activate (cis-activation) or inhibit (cis-inhibition) signaling. Cis-interactions are difficult to investigate in vivo (*Henrique and Schweisguth, 2019*). However, in vitro studies in mammalian cells have demonstrated that Dll1 and Dll4 can cis-activate both Notch1 and Notch2 in CHO-K1 cells (*Nandagopal et al., 2019*), and that Dll1, Dll4, and Jag1 can cis-inhibit Notch1 activation by other ligands (*LeBon et al., 2014*; *Preuße et al., 2015*; *Sprinzak et al., 2010*; *Thambyrajah et al., 2024*). Nevertheless, it has remained unclear whether other receptor-ligand combinations also engage in cis-activation and/or cis-inhibition. We therefore sought to analyze cis-interactions more comprehensively.

In the cis-activation assay (*Figure 1D*), we first preinduced expression of cis-ligands in receivers for 48 hr by titrating 4-epi-Tc to the desired level in the presence of the γ-secretase inhibitor DAPT, as described previously (*Nandagopal et al., 2019*; *Figure 1E*, Methods). To focus on cis-interactions alone, we then cultured receiver cells at low density, amid an excess of wt CHO-K1 cells, and allowed them to signal for 22–24 hr before flow cytometry analysis (*Figure 1E and F*). At these cell densities, trans-interactions among the minority cells should be minimal, activating to no more than 5% of maximum reporter activity (*Figure 4A*). As described below, trans-interactions between sister cells following cell division during the assay could in principle contribute to the observed cis-activation signal, but are insufficient to account for strong cis-activation.

We found that cis-activation was both ligand and receptor specific (*Figure 4B*, left column). Notch1 exhibited no cis-activation for any of the ligands, except for a modest response to Dll4, of no more than 20% of maximal trans-activation (*Figure 4B*, upper left), consistent with previous observations in wt CHO-K1 cells (*Nandagopal et al., 2019*). By contrast, Notch2 was cis-activated by both Dll1 and Dll4, to levels exceeding those produced by trans-activation by high-Dll1 senders (*Figure 4B*, lower left, compare with trans-activation in *Figure 4B*, lower right). (Thus, the Notch2 cis-activation observed here is too strong to be explained by potential trans-interactions between sister cells following cell division in the cis-activation assay.) Jagged ligands, on the other hand, did not cis-activate Notch2 in this assay (*Figure 4B*, lower left).

In contrast to cis-activation, the reciprocal phenomenon of cis-inhibition can only be detected in the context of basal Notch activation by other ligands. We therefore established a parallel 'cis-modulation' assay, which probes the combined effect of cis- and trans-interactions (*Figure 1D*, Methods). In this assay, a minority of preinduced receiver cells are cocultured with an excess of 'high-Dll1' sender cells that constitutively express Dll1 at levels sufficient to strongly activate receiver cells not expressing cis-ligands (*Figure 4—figure supplement 1*).

Cis-inhibition, like cis-activation, was found to depend on both receptor and cis-ligand identity. All ligands cis-inhibited intercellular Dll1-Notch1 signaling, achieving 75–100% inhibition in the highest cis-ligand expression bin (*Figure 4B*, upper right). By contrast, intercellular Dll1-Notch2 signaling was cis-inhibited by Jag1 and Jag2, but not by Dll1 or Dll4, whose cis-activation further increased Notch2 signaling (*Figure 4B*, lower right). These results were not due to non-specific effects of ectopic cis protein expression, as Notch1 and Notch2 reporter activities showed no dependence on expression of negative control proteins H2B-mCherry or NGFR-T2A-H2B-mCherry, in either the cis-activation or cis-modulation assay (*Figure 4B*).

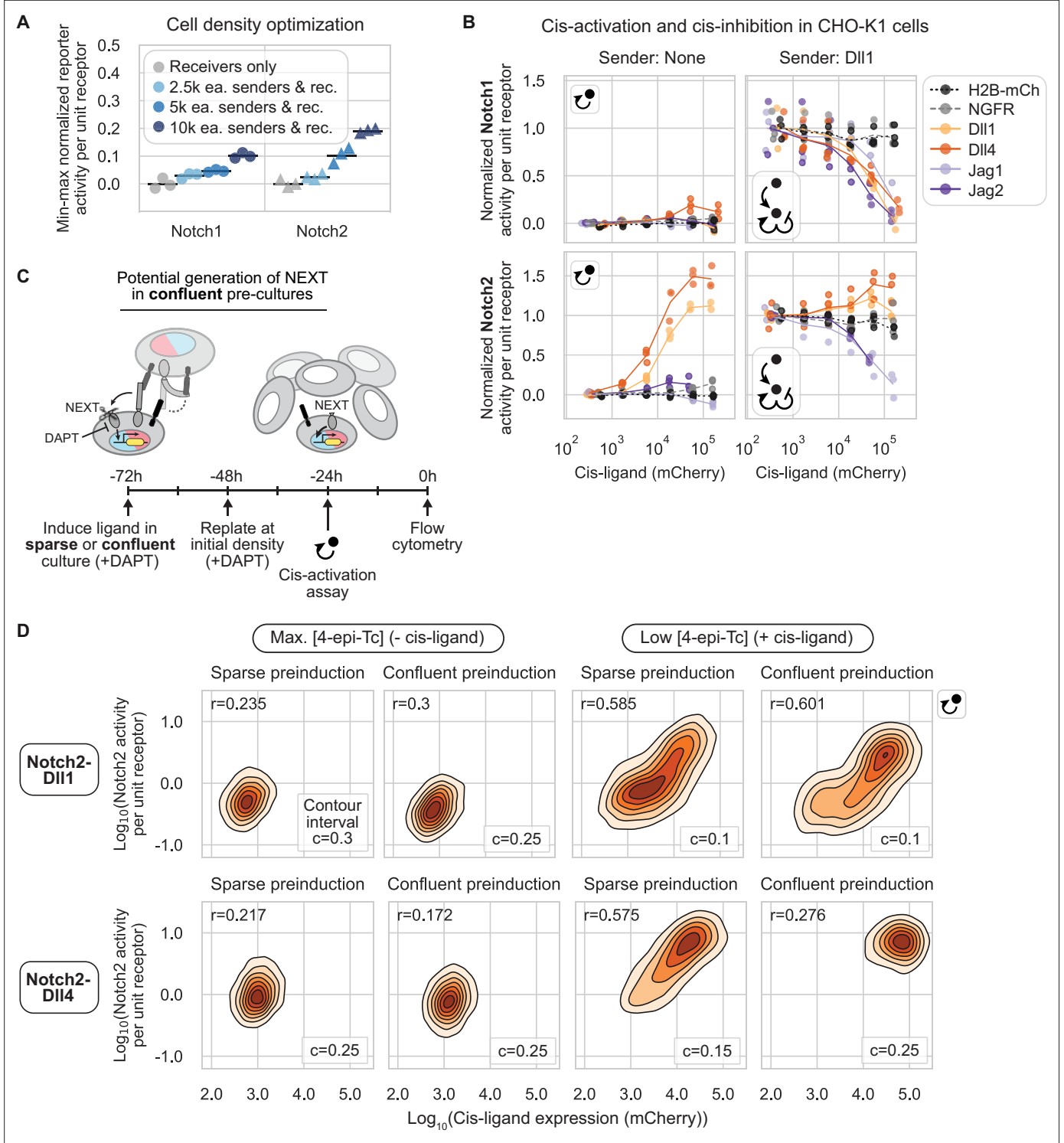

**Figure 4.** Cis-interaction outcomes depend on receptor and ligand identity. (**A**) The cell density used in the cis-activation assay prevents intercellular signaling (Methods). The legend indicates the number of pure sender and pure receiver cells plated along with an excess of wild-type CHO-K1 cells in each condition tested (e.g. '2.5k ea. senders & rec.'=2.5k senders+2.5k receivers). k indicates a multiplier of 1000. Normalized signaling activity (mCitrine/mTurq2) was averaged across all cells in each sample and then min-max normalized (Methods). Black bars represent the mean of three biological repeats. (**B**) Data from cis-activation (left column) and cis-modulation (right column) assays for CHO-K1 Notch1 (top row) and Notch2 (bottom row) receivers coexpressing a Notch ligand or control protein at different levels, corresponding to a range of 4-epi-Tc concentrations (see distributions in *Figure 1—figure supplement 2*). Individual receivers were sorted into discrete bins of mCherry (*Figure 1B*, Methods). The signaling activity (y-axis, defined in (**A**)) and cotranslational ligand expression (x-axis, mCherry (A.U.)) signals were averaged across all cells in each mCherry bin (*Figure 1B*). For

*Figure 4 continued on next page*

*Figure 4 continued*

each receptor, y-axis signaling activity was min-max normalized using the trans-signaling response to high-Dll1 senders in the cis-modulation assay (Methods, *Supplementary file 3*). Lines are interpolated through the mean of three biological replicates (individual data points). (**C**) Experimental workflow to assess the contribution of intercellular signaling during the 48 hr preinduction phase (from –72 to –24 hr) to the overall signal measured in the cis-activation assay at 0 hr (schematic). Cells were preinduced and cultured either sparsely or densely before setting up a cis-activation assay (Methods). NEXT denotes the Notch extracellular truncation, generated during the preinduction phase if cell density enables intercellular ligand-receptor interactions. Red and blue colors in cell nuclei represent H2B-mCherry and H2B-mTurq2, cotranslational reporters of cis-ligand (black) and receptor (gray) expression, respectively. Gamma-secretase is represented by the scissors. (**D**) 2D fluorescence distributions of signaling activity (mCitrine/mTurq2) vs. cis-ligand expression (mCherry) in single-cell fluorescence distributions measured by flow cytometry according to the experiment in (**C**) performed with Notch2-Dll1 and -Dll4 receiver cells. c denotes the interval between successive cell density contours, and cell density in each discrete contour interval is indicated by color. The density interval below c is not shown. Pearson's correlation coefficient r is shown in the top left of each plot, and all p-values were <<0.001.

The online version of this article includes the following figure supplement(s) for figure 4:

**Figure supplement 1.** CHO-K1 receiver clones' reporter dynamic ranges with minimum cis-ligand expression.

**Figure supplement 2.** Cell density can affect expression from the Tet-OFF promoter.

Together, these results reveal three striking differences in cis-interactions among receptor-ligand combinations: (1) Delta ligands cis-activate Notch2 much more strongly than Notch1, (2) all four ligands cis-inhibit Notch1, and (3) Jagged, but not Delta, ligands cis-inhibit Notch2.

## Cis-activation does not arise from prior trans-activation

Since cis-activation in Delta-Notch2 receivers appeared to rival the strength of trans-activation (*Figure 4B*) and is blocked by γ-secretase inhibitors, similar to canonical trans-activation (*Nandagopal et al., 2019*), we considered the possibility that the cis-activation signal could be an artifact of inter-cellular signaling occurring prior to the start of the assay. During the ligand preinduction phase, the γ-secretase inhibitor DAPT prevents S3, but not S2, receptor cleavage. Thus, intercellular contacts between receivers during preinduction culture could in principle generate an S2-cleaved receptor, also known as the Notch extracellular truncation (NEXT). NEXT could then undergo S3 cleavage after DAPT removal, contributing to the total observed signal in the cis-activation assay (*Figure 4C*).

To test this possibility, we carried out a control cis-activation assay in which receivers were cultured at both sparse and confluent densities during the 48 hr cis-ligand preinduction phase, and then replated sparsely or densely after 24 hr of preinduction to maintain the initial low or high cell density conditions (Methods). If cis-activation signal resulted from intercellular signaling during preinduction, only cells preinduced in confluent culture (able to make intercellular contacts) would show a 'cis-activation' signal 24 hr after DAPT removal. For Notch2-Dll1 and Notch2-Dll4, single-cell reporter activities correlated with cis-ligand expression, regardless of whether cells were preinduced at a high or low culture density (*Figure 4D*). While cells pre-cultured in the confluent condition shifted the signaling distributions to higher values relative to cells in sparse pre-cultures, the correlation between cis-ligand expression and reporter activity was maintained. This signaling shift is consistent with elevated expression of Tet-OFF-controlled cis-ligands at high cell density (*Figure 4—figure supplement 2*). Together, this analysis rules out the possibility that Delta-Notch2 cis-activation signal arises from S3-cleavage of NEXT generated by intercellular signaling during the ligand preinduction phase.

## Coexpression of ligands and receptors can produce cis- or trans-signaling in confluent monoculture

In a population of cells coexpressing both receptors and ligands, signaling could occur through cis-activation, trans-activation, or both. How do these two modes of signaling combine when both can occur simultaneously? To disentangle the contributions of these two modes of signaling, we compared cis-activation alone (sparsely cultured receivers, 'cis-activation assay') to signaling occurring through both cis- and trans-interactions (confluent receivers, cis-+trans-activation assay') (*Figure 1D*). We repeated this analysis for each receptor-ligand pair, and for both dLfng and Lfng conditions, across a broad range of ligand expression (*Figure 5—figure supplement 1*).

For ligand-Notch1 interactions (cis+trans), trans-activation significantly contributed to the overall signaling strength observed in our experiments (*Figure 5A*, *Figure 5—figure supplement 2*). With Lfng, the combination of cis- and trans-interactions produced over twice the signaling activity from

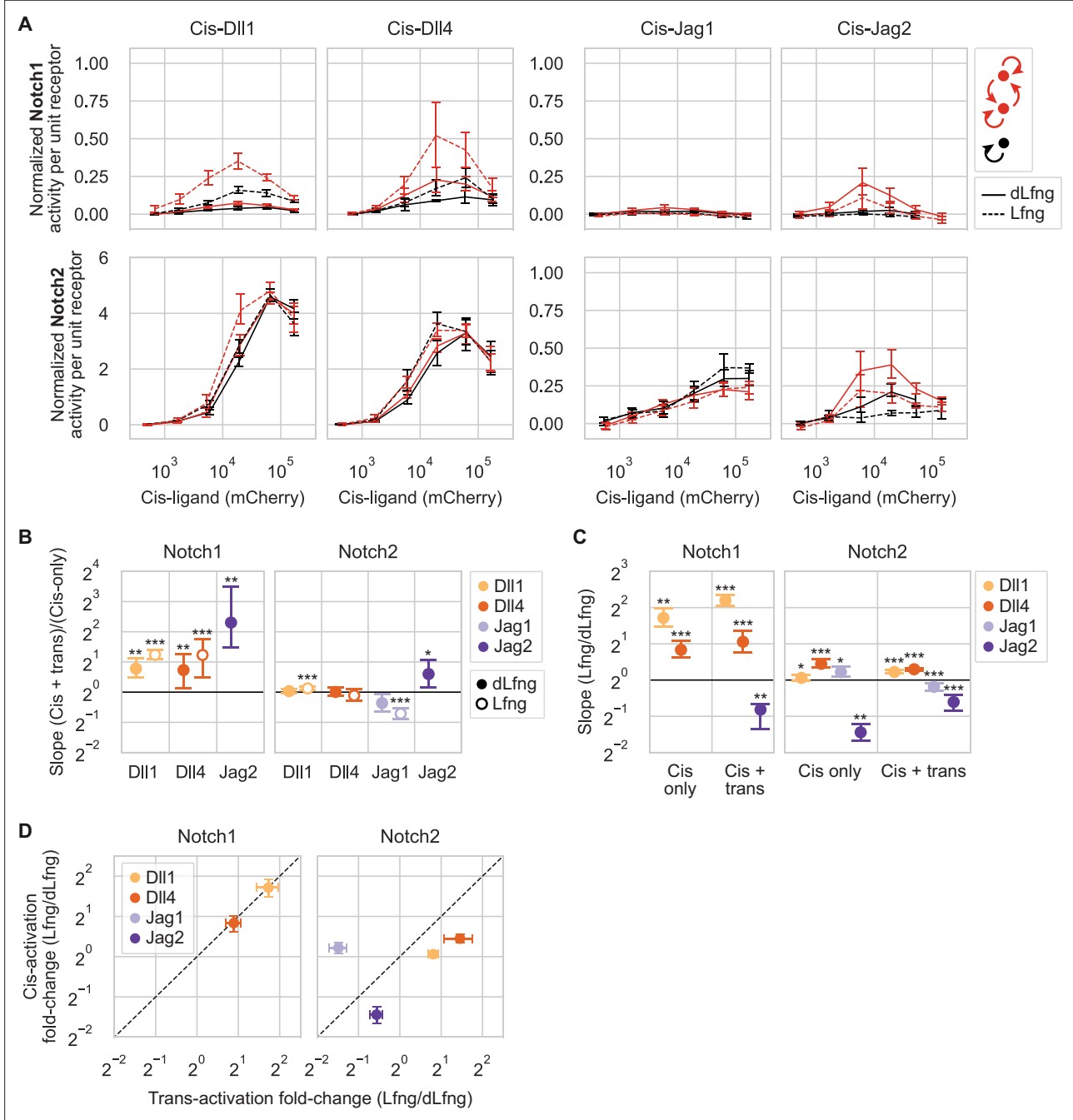

**Figure 5.** Coexpression of ligands and receptors can produce cis- or trans-signaling in confluent monoculture. (**A**) Results of cis-activation (black) and cis-+trans-activation (red) assays for CHO-K1 Notch1 (top row) and Notch2 (bottom row) receivers coexpressing a Notch ligand at different levels. Ligand expression was varied by titrating 4-epi-Tc concentration (see distributions in *Figure 5—figure supplement 1*). X- and y-axis values respectively represent cis-ligand expression (mCherry, A.U.) and min-max normalized, mean Notch signaling activity (mCitrine/mTurq2) of all cells in a given mCherry bin (Methods, *Supplementary file 3*). Y-axis values thus reflect signaling strengths relative to strong trans-signaling. Lines are interpolated through the mean of seven biological replicates in each mCherry bin. Error bars are bootstrapped 95% confidence intervals. (**B–C**) Fold differences in signaling between the cis-+trans-activation assay vs. the cis-activation assay (**B**) and between Lfng and dLfng conditions (**C**). Values are means and 95% confidence intervals of the slopes computed from linear regressions to 10,000 bootstrap replicates of paired signaling activities from the two assay types (see scatterplots with linear regressions for (**B**) in *Figure 5—figure supplement 3* and for (**C**) in *Figure 5—figure supplement 5*). Asterisks denote the p-value of the test statistic from a one-sided Wilcoxon signed-rank test (*p-val<0.05; **p-val<0.01; ***p-val<0.001), reflecting whether the slope is greater or lesser than 1. Fold differences could not be computed for ligand-receptor combinations with background-level signaling in one or both axis coordinates. (**D**) Comparison of Fringe effects on cis-activation (y-axis, mean values, and confidence intervals from (**C**)) vs. trans-activation (x-axis, mean values and confidence intervals from *Figure 2C*).

*Figure 5 continued on next page*

*Figure 5 continued*

The online version of this article includes the following figure supplement(s) for figure 5:

**Figure supplement 1.** CHO-K1 receiver clones' cis-ligand expression distributions for the cis-activation assay and cis-+trans-activation assay.

**Figure supplement 2.** Flow cytometry data analysis pipeline without mCherry binning yields similar results.

**Figure supplement 3.** Allowing intercellular contacts between cells coexpressing ligands and receptors alters signaling activity.

**Figure supplement 4.** CHO-K1 receiver clones' reporter dynamic ranges with minimum cis-ligand expression.

**Figure supplement 5.** Lfng effects on cis-activation and cis-+trans-activation.

Dll1 or Dll4 compared to cis-interactions alone in the increasing phase of the dose-response curves (*Figure 5B*). This result indicates that trans-activation can increase Notch1 activity beyond the level produced by cis-activation alone. For Jag2-Notch1, in either dLfng or Lfng conditions, signaling was only observed with trans-interactions (*Figure 5A and B*). For Jag1-Notch1, no activation was seen in the cis-activation or the cis-+trans-activation modes, consistent with earlier results (*Figure 2A, Figure 4B*).

Ligand-Notch2 interactions showed a qualitatively different profile of behaviors compared to Notch1. Of the four ligands, trans Jag2 modestly increased Notch2 signaling beyond the level achieved by cis-activation alone, by about 1.5-fold (*Figure 5B*). In all other ligand-Notch2 conditions, trans-ligands failed to increase signaling, suggesting that cis-activation alone is sufficient to explain the observed level of signaling in these cis+trans conditions. With Jag1-Notch2, combined cis- and trans-signaling was slightly lower than cis-only signaling with Lfng (*Figure 5B*), suggesting that trans-interactions may limit signaling from cis-activation in this case. (Note that this assay used a Jag1-Notch2 clone with lower background signaling than in *Figure 4B*, enabling detection of cis-activation (*Figure 4—figure supplement 1*; *Figure 5—figure supplement 4*).)

Together, these results indicate that cis- and trans-signaling combine differently across the ligand-receptor combinations. Some combinations showed increased activity with cis- and trans-interactions compared to cis-interactions alone, while other combinations showed either a preference for a particular signaling mode or appeared to signal equally well through both modes.

## Lfng modulates cis-activation strengths

Cis-activation and trans-activation share several features. They both require receptor-ligand binding at the cell surface and depend on γ-secretase cleavage (*Nandagopal et al., 2019*). Additionally, they share similar ligand-dependent responses to Rfng, which increases both cis-activation and trans-activation by Dll1 but not Dll4 (*Kakuda et al., 2020*; *Nandagopal et al., 2019*). However, it was not clear whether Lfng would exhibit similar effects on cis-activation and trans-activation, and how any similarities or differences between activation modes might vary across receptor-ligand combinations.

To address these questions, we analyzed the results from the cis-activation assay with or without Lfng (*Figure 5A*). Effects of Lfng on Notch1 cis-activation (*Figure 5C*, *Figure 5—figure supplement 5*) quantitatively resembled those on trans-activation (*Figure 5D*). Lfng significantly increased Dll1-Notch1 and Dll4-Notch1 signaling by 3.3-fold and 1.8-fold, respectively, in both assays. By contrast, Lfng had no effect on cis-activation of Notch1 by Jag1 and Jag2, which remained negligible with or without Lfng.

With Notch2, Lfng's effects on cis- and trans-interactions diverged (*Figure 5D*). Despite its ability to increase Dll1 and Dll4 trans-activation of Notch2, Lfng did not strongly affect cis-activation. While Lfng sharply reduced Jag1-Notch2 trans-activation (*Figure 2C*), it had negligible effects on cis-activation by the same components (*Figure 5C*). Conversely, for Jag2-Notch2, Lfng had weak effects on trans-activation, but reduced cis-activation by more than 2-fold (*Figure 5D*). Taken together, these results suggest that cis- and trans-activation mechanisms respond to Lfng similarly for Notch1, but differently for Notch2.

## Three-component interactions modulate cis-inhibition

Notch trans-activation and cis-inhibition involve oligomeric binding and clustering (*Chen et al., 2023*; *Narui and Salaita, 2013*), provoking the question of whether higher order (beyond pairwise) signaling interactions could influence signaling. For example, the identity of a trans-ligand could in principle impact the effective strength of cis-interactions between a receptor and cis-ligand. To examine this

possibility, we used the cis-modulation assay (*Figure 1D*) to determine how cis-inhibition strength depends on the identity of the trans-activating ligand for each cis-ligand-receptor pair. We cocultured each of the eight cis-ligand containing receiver cell lines (expressing endogenous CHO-K1 Fringes and preinduced to a broad range of cis-ligand levels) with an excess of each of four constitutive sender cell lines (*Figure 6—figure supplement 1A and B*), for a total of 32 coculture combinations (*Figure 6A*).

The effects of cis-interactions were qualitatively similar across different trans-activating ligands (*Figure 6A*). In all cases, cis-Jagged ligands inhibited Notch1 and Notch2, while cis-Delta ligands inhibited Notch1 and activated Notch2 (*Figures 6A and 4B*). (Note that cis-inhibition of Notch1 could not be analyzed when Jag1 sender cells were used due to a lack of trans-activation by Jag1 (*Figure 6—figure supplement 1C*).)

Quantitatively, cis-inhibition strength depended on both cis-ligand and trans-ligand identity. For example, Jag2 was the most effective Notch1 cis-inhibitor across all trans-ligands (*Figure 6B*). Ligands' relative cis-inhibition strengths did not correlate well with trans-activation strengths (*Figure 6C*). In general, combinations with modest or strong cis-activating potentials, such as Dll4-Notch1, Dll1-Notch2, and Dll4-Notch2 (*Figure 5A*), showed relatively poor cis-inhibition efficiencies relative to trans-activation (*Figure 6C*). The converse was also true for Notch1: the non-cis-activating ligands, Jag1 and Jag2 (*Figure 5A*), showed strong cis-inhibition efficiencies relative to trans-activation (*Figure 6C*). For Notch2, no ligands favored cis-inhibition over trans-activation, consistent with the observation that all ligands can cis-activate Notch2 to some extent (*Figure 5A*).

To quantitatively compare cis-inhibition efficiencies for different trans-ligands, we computed trans-adjusted cis-inhibition efficiencies by dividing values from *Figure 6B* by the strength of trans-activation mediated by each sender population (*Figure 6—figure supplement 1C*). For Notch1, ligands' trans-adjusted cis-inhibition efficiencies showed diverse dependencies on the identity of the trans-ligand (*Figure 6D*). Notably, each cis-ligand inhibited signaling from the trans-ligand of the same identity with an equal or greater efficiency than it inhibited other trans-ligands. For example, Dll1 cis-inhibited trans-Dll1 signaling more efficiently than either trans-Jag2 or -Dll4. Similarly, Jag2 cis-inhibited trans-Jag2 signaling more efficiently than trans-Dll1 signaling, and Dll4 cis-inhibited trans-Dll4 signaling more efficiently than trans-Jag2 signaling. In contrast to the variability of Notch1 cis-inhibition, Jag-Notch2 cis-inhibition was relatively uniform across different cis- and trans-ligand combinations, varying by no more than 2-fold between the strongest and weakest cis-inhibiting combinations. Together, these results suggest that cis-inhibition strengths for Notch1, and to a lesser extent, Notch2, depend on the interplay of trans-ligand, cis-ligand, and receptor.

Finally, we estimated the ultrasensitivity of cis-inhibition by fitting Hill functions to most of the dose-response curves in *Figure 6A*. Hill coefficients ranged from ~0.75 to 1.5 (*Figure 6E*), below the values obtained for trans-activation by the same components (*Figure 2F and G*). These results suggest that cis-inhibition is a less ultrasensitive, or potentially more stoichiometric, process compared to trans-activation. Taken together, these data suggest a complex interplay among ligands and receptors in cis and trans.

## Similar ligand-receptor signaling features occur in a distinct cell line

The studies above were conducted in CHO-K1 cells, but interactions between Notch receptors and ligands could in principle depend on cell type or context. To test this possibility, we constructed receiver and sender cell lines in an unrelated C2C12 mouse myoblast background, which is distinct from CHO-K1 cells, allows control of Notch component expression without altering cell fate or morphology, and has routinely been used to investigate the role of Notch signaling in muscle cell differentiation (*Dahlqvist et al., 2003*; *Gioftsidi et al., 2022*; *Nofziger et al., 1999*; *Shawber et al., 1996*). Because C2C12 cells express Notch receptors, ligands, and Fringes (*Liang et al., 2021*; *Sassoli et al., 2012*; *Shimizu et al., 2001*; *Figure 7—figure supplement 1A*), we first constructed a C2C12 base cell line, dubbed C2C12-Nkd, for 'Notch knockdown'. We used two rounds of CRISPR/Cas9 editing to target Notch2 and Jag1, the most abundant receptor and ligand, for deletion (Methods). The resulting clone (C2C12-Nkd) showed a loss of Notch2 protein by western blot (*Figure 7—figure supplement 1B*) and a loss of Jag1 mRNA by RT-PCR (*Figure 7—figure supplement 1C*). Additionally, it exhibited an impaired ability to upregulate Notch target genes, including Notch1 and Notch3 (*Castel et al., 2013*), relative to wt C2C12 (*Figure 7—figure supplement 1A*, right). We then engineered sender

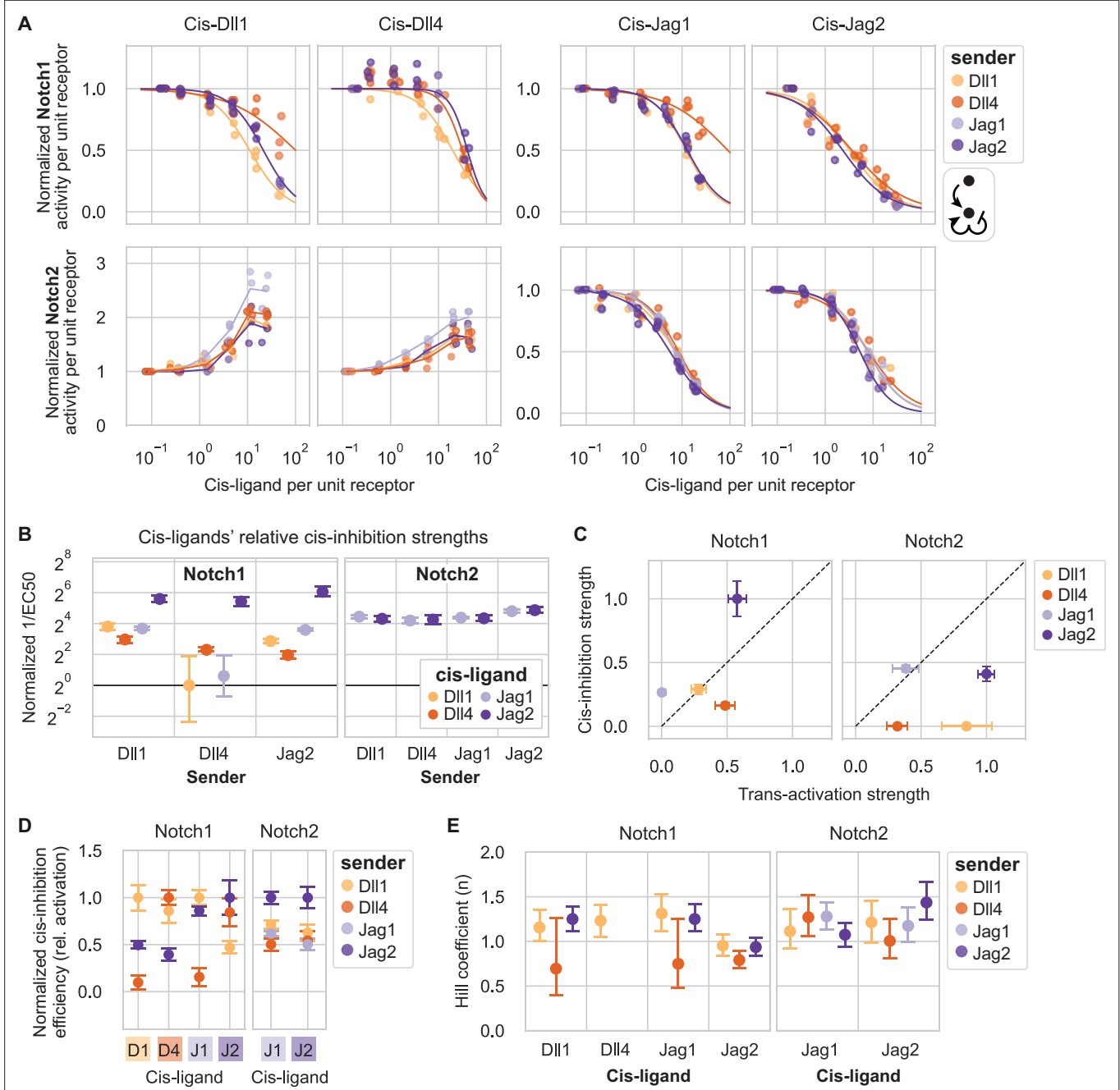

**Figure 6.** Cis-inhibition strengths depend on the identities of the receptor, the cis-ligand, and the trans-ligand. (**A**) Results of cis-modulation assays for CHO-K1 Notch1 (top row) and Notch2 (bottom row) receivers with endogenous Fringes, coexpressing cis-ligands at different levels corresponding to a range of 4-epi-Tc concentrations (*Supplementary file 2*). Columns are cis-ligands, and colors indicate the ligand expressed by the cocultured sender cells (*Figure 6—figure supplement 1A and B*; *Supplementary file 3*). X- and y-axis values are flow cytometry fluorescence values averaged across all cells in a given mCherry bin (*Figure 1B*). The x-axis represents cis-ligand expression (mCherry, A.U.) relative to cotranslational receptor expression (mTurq2, A.U.); see also Methods. Y-axis values are Notch signaling activity (reporter activity [mCitrine, A.U.] divided by cotranslational receptor expression [mTurq2, A.U.]). For each bioreplicate curve, signaling activity was normalized to maximum trans-signaling, at the y-value with minimum cis-ligand (*Figure 6—figure supplement 1C*). Cis-inhibition of Jag1-Notch1 signaling could not be analyzed because trans-activation was too weak. For Notch2-Delta receivers, lines connect means of three biological replicates in each mCherry bin along the x-axis. For other receivers, curves are fits of three biological replicates to a repressive Hill function with maximum y=1 (Methods). (**B**) Mean cis-inhibition strengths based on bootstrap analysis of repressive Hill fits in (**A**). Cis-inhibition strength is defined as the inverse of the fit EC50 parameter, relative to the minimum value across all combinations. (**C**) Comparison of cis-inhibition strengths (y-axis, mean values and confidence intervals from (**B**) with Dll1 senders rescaled such that the maximum value over all ligand-receptor combinations equals 1) vs. trans-activation strengths with endogenous Fringes (x-axis, mean values and confidence intervals

*Figure 6 continued on next page*

*Figure 6 continued*

from *Figure 2H*). Cis-inhibition strength was set to zero for Delta-Notch2 combinations. (**D**) Cis-inhibition efficiencies and confidence intervals from (**B**) adjusted for the strength of trans-signaling induced by the indicated sender cells (*Figure 6—figure supplement 1C*), and normalized such that the maximum cis-inhibition strength is equal to 1 for each cis-ligand-receptor combination. Error bars reflect uncertainty on cis-inhibition efficiency only (trans-activation error bars were not propagated). (**E**) Hill coefficients (**n**) computed from fits of data in (**A**) to repressive Hill functions. Hill coefficients could not be computed for Notch1-Dll4 receivers with Dll4 or Jag2 senders because of the modest cis-activation observed at intermediate cis-ligand levels for those combinations. Values are means and 95% confidence intervals from bootstrap analysis.

The online version of this article includes the following figure supplement(s) for figure 6:

**Figure supplement 1.** Ligands' trans-activation strengths show greater diversity for Notch1 than Notch2.

and receiver cells in the C2C12-Nkd background with the same ligand and receptor constructs used previously with CHO-K1 cells.

To compare C2C12-Nkd signaling responses to those of CHO-K1 cells, we repeated the trans-activation assay described previously (*Figure 1D and E*) in the C2C12-Nkd background. In these assays, to minimize residual expression of endogenous Notch components, we also inserted a transient knockdown step, in which we used siRNAs to suppress basal expression of endogenous Notch receptors, and, in some cases, Rfng (*Figure 7—figure supplement 1D*, Methods). We focused on one ligand of each Delta and Jagged class, Dll1 and Jag1, to test whether these ligands would exhibit similar relative abilities to trans-activate Notch1 and Notch2 in the C2C12-Nkd background vs. the CHO-K1 cells.

Jag1 senders activated Notch1 C2C12-Nkd receivers poorly, despite activating Notch2 receivers well (*Figure 7A*), just as we observed in CHO-K1 (*Figure 2A*). Dll1 senders activated Notch1 much more efficiently than Jag1 senders did, but Notch2 receivers were activated to a similar extent by Jag1 and Dll1 senders (<1.5-fold difference), also consistent with our results in CHO-K1.

Next, to compare cis-interactions between cell lines, we performed cis-activation and cis-modulation assays (*Figure 1D*) in the C2C12-Nkd background, adjusting assay parameters such as cell density to compensate for differences in cell properties (*Figure 7B*, Methods). We focused on Dll4 and Jag2, for one ligand each of the Delta and Jagged classes. With Notch2, we observed strong cis-activation by Dll4 and cis-inhibition by Jag2 (*Figure 7C*), similar to results in the CHO-K1 background (*Figure 7D*). Also similar to their behavior in CHO-K1 cells, Dll4 and Jag2 cis-inhibited Notch1 reporter activity in C2C12-Nkd cells relative to the NGFR cis-ligand negative control. Furthermore, Jag2 did not cis-activate Notch1 in this background. However, in contrast to CHO-K1, where Dll4 weakly cis-activated Notch1 even without Fringe expression (*Figure 5A*), we observed no cis-activation by Dll4 in C2C12-Nkd.

Together, these results indicate that key signaling behaviors are similar between the CHO-K1 and C2C12 backgrounds. In particular, weak trans-activation of Notch1 by Jag1, and strong cis-activation of Notch2 by Dll1 and Dll4, are likely to be intrinsic properties of these ligands and receptors. Future studies will be necessary to extend these comparisons to a broader range of cell contexts.

## Discussion

It has been known for decades that the mammalian Notch signaling pathway contains multiple interacting ligand, receptor, and Fringe variants. However, the functional consequences of this component diversity have remained difficult to understand. This is due in part to a lack of comprehensive measurements of signaling outcomes across ligand-receptor pairs, signaling modes (cis vs. trans), and Fringe modification states. Here, systematic mapping of interactions revealed that each ligand and receptor is unique in terms of its profile of cis- and trans-activation and cis-inhibition activities, and the dependence of these activities on Lfng (*Figure 8A*).

Canonical trans-signaling exhibited a range of signaling activities across ligand-receptor-Fringe combinations. Ligand class (Delta vs. Jagged) did not predict the signaling level with either receptor. However, it did correlate strongly with the sign of response to Lfng, which respectively increased or decreased signaling by Delta or Jagged ligands to both receptors (*Figure 2C*). Amid variability in trans-signaling efficiencies, there was striking uniformity in the ultrasensitivity of Notch responses, which was roughly constant across ligand-receptor-Fringe combinations (*Figure 2F and G*).

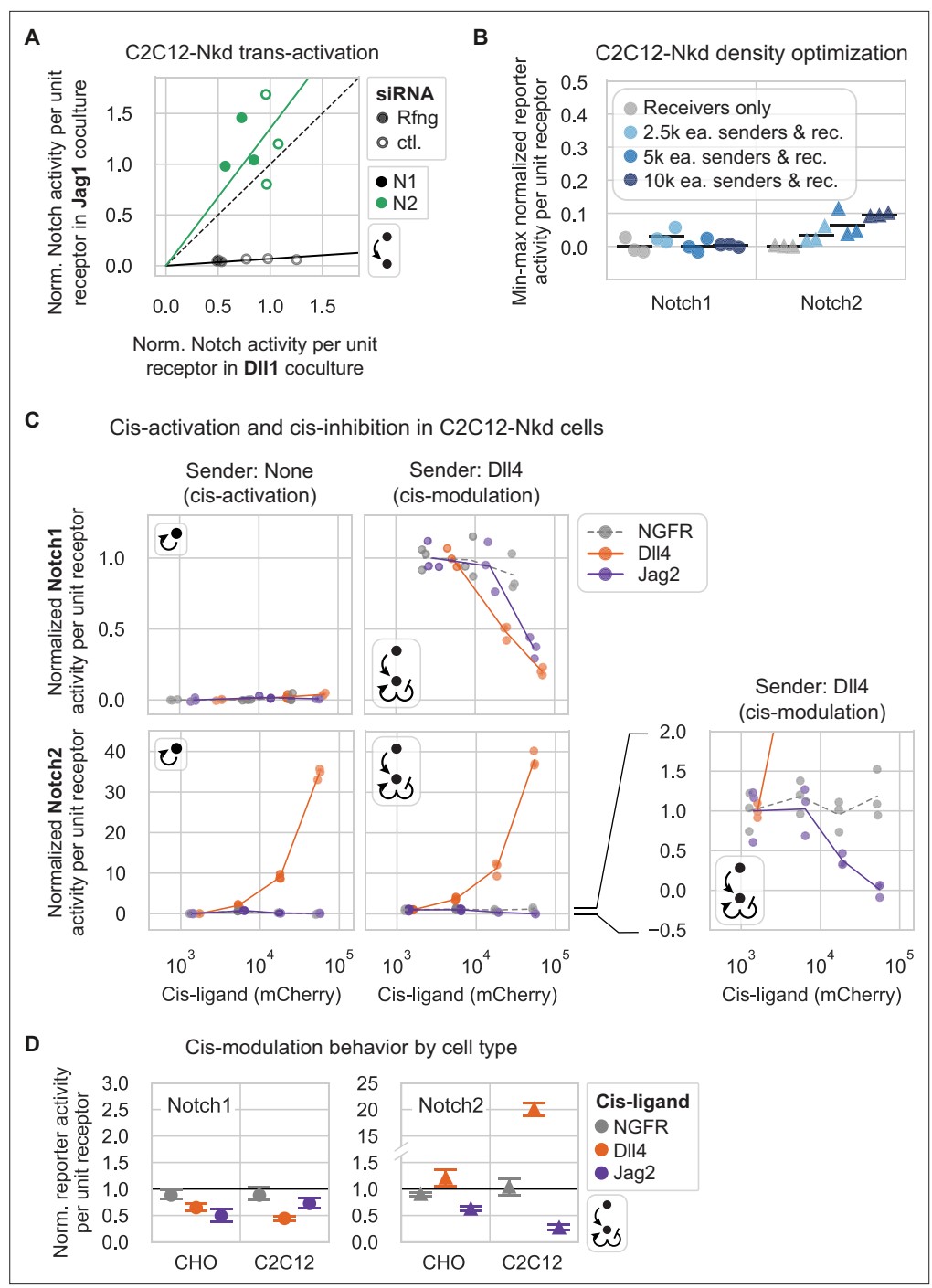

**Figure 7.** Similar ligand-receptor signaling features occur in CHO-K1 and C2C12-Nkd cells. (**A**) Normalized Notch signaling strength in C2C12-Nkd Notch1 (black) or Notch2 (green) receivers cultured with Jag1 (y-axis) vs. Dll1 (x-axis) senders in a trans-activation assay. Receivers were treated with negative control (unfilled markers) or mouse Rfng (filled markers) siRNAs prior to the assay. X and y-axis values are mean signaling activity (reporter activity, mCitrine, divided by cotranslational receptor expression, mTurq2), background subtracted and normalized to the average signal in Dll1 coculture with negative control siRNA treatment for each receiver. Solid lines are the least-squares best fit through all points for a given receiver (pooling control and Rfng siRNA-treated samples, see Methods). Both slopes were significantly greater or less than 1 according to one-sided Wilcoxon signed-rank tests (p-val<0.05). The black dashed line is y=x. (**B**) The cell density used in the C2C12-Nkd cis-activation assay prevents intercellular signaling. After siRNA treatment to knock down residual endogenous Notch components (Methods), the assay was performed similarly to the assay used for CHO-K1 cells (***Figure 4A***), except for use of a

*Figure 7 continued on next page*

*Figure 7 continued*

12-well plate. Here, y-axis values are min-max normalized, mean Notch signaling activities. Black bars are the mean of three biological repeats. (**C**) Results of cis-activation (left column) and cis-modulation (right column) assays for C2C12-Nkd Notch1 (top row) and Notch2 (bottom row) receivers coexpressing a Notch ligand or control protein. Fluorescence values were averaged differently for Notch1 vs. Notch2 based on responses to the nerve growth factor receptor (NGFR) control (see Methods). For both receptors, signaling activity defined in (**A**) was min-max normalized using the maximal trans-signaling in receivers cultured with the high-Dll4 senders (Dll4-2H10) used in the cis-modulation assay (**Supplementary file 3**). Lines connect means of three biological replicates (individual data points) in each mCherry bin. The bottom right plot is a zoomed-in view of the data for Notch2 receivers cultured with high-Dll4 senders, showing Jag2 cis-inhibition of Notch2 activation by Dll4 senders. (**D**) Comparison of cis-ligand effects in the cis-modulation assay for CHO-K1 ('CHO') vs. C2C12-Nkd ('C2C12') cell types. Y-axis units are normalized signaling activities as defined in **Figure 4B** for CHO-K1 cells and in (**C**) for C2C12-Nkd cells. (Note, maximal trans-signaling activities used in normalization differed greatly for CHO-K1 and C2C12-Nkd, so responses should be compared qualitatively, but not quantitatively (Methods).) Here, y-values are signaling activities corresponding to a cis-ligand expression level where x-axis cotranslational H2B-mCherry fluorescence equals $3 \times 10^4$ A.U., calculated by fitting a line between the x-axis mCherry bins flanking $x = 3 \times 10^4$ A.U. Y-values are the mean, and error bars are 95% confidence intervals, from bootstrap analysis.

The online version of this article includes the following source data and figure supplement(s) for figure 7:

**Figure supplement 1.** CRISPR/Cas9 editing of C2C12 cells yields Notch-depleted clone C2C12-Nkd.

**Figure supplement 1—source data 1.** Original files for western blots displayed in **Figure 7—figure supplement 1B**.

**Figure supplement 1—source data 2.** PDF file containing original western blots for **Figure 2—figure supplement 2B**, indicating the relevant bands and sizes.

**Figure supplement 1—source data 3.** Original file for RT-PCR gel image displayed in **Figure 7—figure supplement 1C**.

**Figure supplement 1—source data 4.** PDF file containing original RT-PCR gel image for **Figure 7—figure supplement 1C**, indicating the relevant bands and sizes.

---

Including cis-interactions revealed additional differences among ligand and receptor variants. All ligands interacted with both Notch1 and Notch2 in cis, but did so with effects that varied across ligand-receptor-Fringe combinations. Broadly, Delta ligands cis-inhibited only Notch1, and cis-activated Notch2 at levels exceeding the maximal Notch2 trans-signaling observed here, while Jagged ligands cis-inhibited both receptors (**Figure 4B**). Overall, the effects of cis-ligands were complex, with signaling activity often exhibiting Fringe dependence (**Figure 5C**), a biphasic dependence on ligand level (**Figure 5A**), and, in some cases, sensitivity to trans-activating ligands (**Figure 6D**). Notably, Lfng had different impacts on cis- and trans-interactions for some ligand-receptor combinations such as Jag1-Notch2 and Jag2-Notch2 (**Figure 5D**), suggesting the possibility that cis- and trans-activation may involve different molecular mechanisms.

Lfng-dependent trans-inhibition of Notch1 by Jag1 has been shown to play a key role in some contexts such as angiogenesis, where it inhibits Dll4-Notch1 signaling (**Benedito et al., 2009**; **Pedrosa et al., 2015**) and the embryonic pancreas, where it inhibits Dll1-Notch1 signaling (**Golson et al., 2009**). Our results explain how trans-inhibition could result from the combination of strong binding with weak activation, which was most pronounced for Jag1-Notch1 (**Figure 2A**, **Figure 3B**, and **Figure 6C**). Further, this effect was enhanced by Lfng, which increased binding strength in the ligand binding assay (**Figure 3H**), while decreasing signaling in both the coculture and plated ligand assays (**Figures 2C and 3I**), consistent with previous observations (**Hicks et al., 2000**; **Kakuda and Haltiwanger, 2017**; **Taylor et al., 2014**; **Yang et al., 2005**). No other ligand exhibited this combination of strong binding and weak activation. These results suggest that Notch1 trans-inhibition may be a core function of Jag1, and could help explain the sometimes divergent behavior of Jag1 mutants compared with other ligands (**Benedito et al., 2009**; **Chrysostomou et al., 2020**). However, it is possible that positive Jag1-Notch1 signaling is also functionally important in some contexts, since weak signaling from Jag1-expressing OP9 cells prevents T-cell progenitors from differentiating into the B-cell lineage, in contrast with no-ligand controls (**Lehar et al., 2005**).

A major question is how Notch receptors integrate information from cis- and trans-ligands. Previous work introduced a general model in which cis- and trans-ligands compete to form different

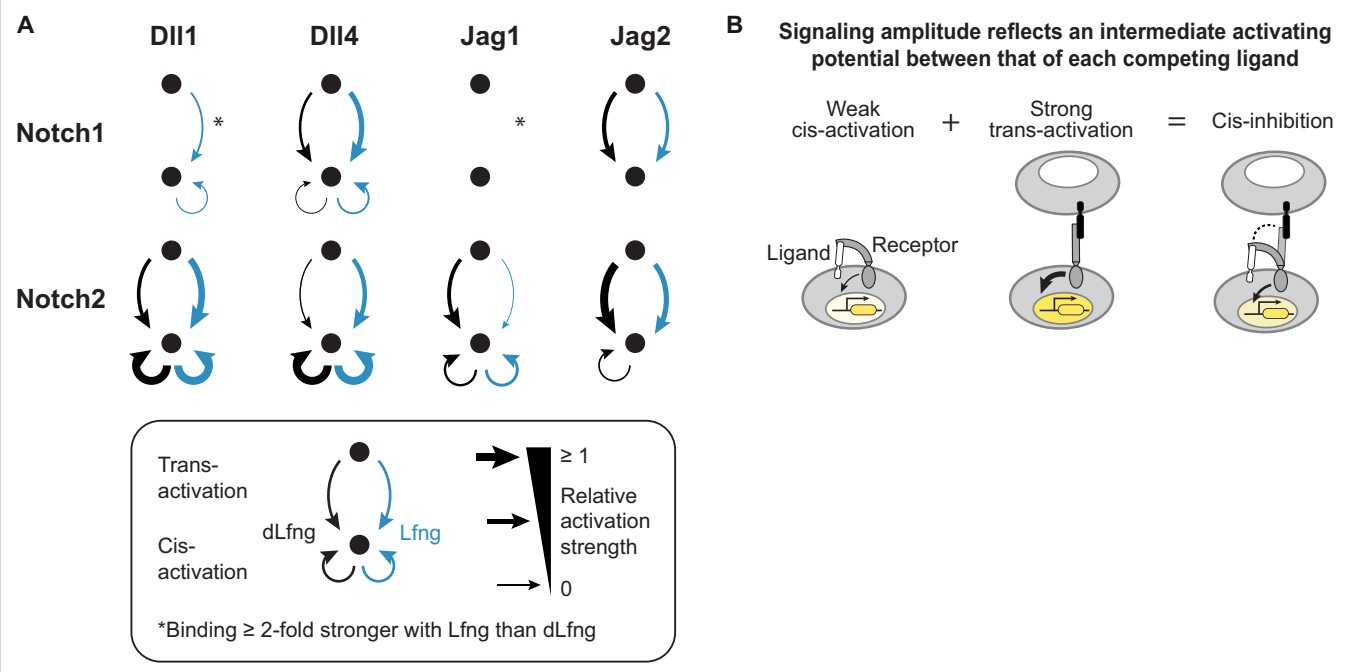

**Figure 8.** Each receptor-ligand combination has a unique activity profile. (**A**) Schematic summary of productive signaling interactions observed here. Min-max normalized trans- and cis-activation strengths above a minimum threshold of 0.1 (Methods) are shown for all eight receptor-ligand pairs both with dLfng (black) and with Lfng (blue) expression, based on data from **Figure 2A and Figure 5A** (peak cis-activation). Combinations in which binding was strengthened at least 2-fold by Lfng (**Figure 3H**) are indicated with an asterisk. Absence of an asterisk should be interpreted as an unknown Lfng effect, rather than a lack of effect. (**B**) Cis-activation and cis-inhibition exhibited an inverse relationship (schematic, based on **Figure 6C**). Black ligands represent strong activators and white ligands represent weak activators, which are still able to bind receptors. Yellow intensity in cell nuclei represents signaling activity.

ligand-receptor complexes. In this model, each complex may have a distinct binding strength and a distinct signaling efficiency (**Formosa-Jordan and Ibañes, 2014**; **Sprinzak et al., 2011**; **Sprinzak et al., 2010**). The competitive binding model is indirectly supported by previous observations that the same conserved domain of the Serrate ligand is required for both cis-inhibitory and trans-activating Notch interactions in *Drosophila* (**Cordle et al., 2008**), that interdomain flexibility in receptors and ligands could facilitate antiparallel binding in both cis and trans (**Luca et al., 2015**), and that synthetic synNotch ligands can cis-inhibit their receptors when expressed in the same cell (**Morsut et al., 2016**). However, it has not been tested systematically.

Our results are broadly consistent with the competitive binding model (**Figure 8B**). For example, when strongly cis-activating ligands compete with strongly trans-activating ligands, they cause no net decrease in signaling, as observed when cis-Delta ligands interact with Notch2 (**Figure 4B**, **Figure 6A**). By contrast, when a strongly binding cis-ligand produces a weak or inactive complex, it can cis-inhibit. In general, combinations that did not cis-activate (Jagged-Notch1) showed relatively stronger cis-inhibition than combinations that cis-activated weakly (Delta-Notch1, Jag1-Notch2) (**Figure 6C**, **Figure 5A**). Cis-Jag2-Notch1 was the strongest cis-inhibiting combination overall. Because of its unique attributes of strong binding and weak cis-activation, cis-Jag2-Notch1 could operate in a 'mutual inactivation' regime in which a cis-ligand titrates the receptor levels available to potential trans-activating ligands (**LeBon et al., 2014**; **Sprinzak et al., 2010**; **Xu et al., 2023**). Linear titration of this type is consistent with the roughly linear logarithmic sensitivity observed in dose-response experiments for Notch1 with cis-Jag2 (**Figure 6E**).

Our results revealed additional complexities in cis-inhibition. Existing models describe competitive inhibition of Notch signaling in terms of pairwise cis and trans receptor-ligand binding affinities (**del Álamo et al., 2011**; **Formosa-Jordan and Ibañes, 2014**; **LeBon et al., 2014**; **Luna-Escalante et al., 2018**; **Sprinzak et al., 2010**). However, in our data, cis-inhibition strength depended on the identity of the receptor, the activating trans-ligand, and the cis-ligand. Strikingly, for Notch1, each type of

ligand had a comparative advantage in its ability to cis-inhibit trans-activation by itself compared to other ligands (*Figure 6D*). For example, Dll1 cis-inhibited trans-Dll1 signaling more efficiently than trans-Jag2, while Jag2 cis-inhibited trans-Jag2 signaling more efficiently than trans-Dll1. These results hint at the possibility of multi-way interactions, possibly involving oligomeric structures.

Overall, our results were broadly consistent with earlier work for previously analyzed receptor-ligand pairs (*Benedito et al., 2009*; *Hicks et al., 2000*; *Kakuda et al., 2020*; *Song et al., 2016*; *Stanley and Guidos, 2009*; *Tveriakhina et al., 2018*). However, there were some notable discrepancies. While we observed a negative effect of Lfng on trans Jag1-Notch2 signaling (*Figure 2C*), consistent with one previous study (*Shimizu et al., 2001*), other studies showed either positive (*Hicks et al., 2000*) or no effect (*Kakuda et al., 2020*) with this combination. A second notable discrepancy was our observation that Delta-like ligands cis-activate but cannot cis-inhibit Notch2 (*Figure 6A and Figure 7C*). This conflicts with a report that wt Dll4 cis-inhibited Notch2 in U2OS osteosarcoma cells (*Chen et al., 2023*). It is possible that cell type-specific factors prevent Dll4 cis-activation of Notch2 in U2OS cells or that Dll4 cis-inhibition requires some interaction between Dll4 and the N2ICD (absent from our N2ECD-Gal4 receptors). Third, we found that Dll1 trans-activated Notch2 more efficiently than Notch1 (*Figure 2A*), consistent with results from a previous plated ligand assay (*Kakuda et al., 2020*), but conflicting with a coculture assay showing equivalent activation of both receptors by Dll1 (*Tveriakhina et al., 2018*).

Looking ahead, a major question is how to understand and predict the behavior of complex pathway profiles involving multiple receptors, ligands, and Fringe proteins. For example, one of the most prevalent Notch component expression profiles, found in cell types within the heart, trachea, forelimb, and other tissues, exhibits high levels of Notch1 and Notch2 as well as moderate levels of all four activating ligands (*Granados et al., 2022*). What signaling properties does this combination of components produce? Which cell types can it signal to or receive signals from? Expanding the approach developed here to allow simultaneous analysis of multiple receptor activities could help to address these questions. On the other hand, directly measuring the signaling properties of all prevalent profiles may be infeasible with current techniques. Therefore, it will be critical to develop quantitative models that predict signaling behaviors among cells based on their pathway expression profiles. The data and analysis provided here should help to create such models, and thereby improve our ability to understand, predict, and control Notch signaling in diverse contexts.

## Methods

### Plasmid construction

The pEV-2xHS4-UAS-H2B-Citrine-2xHS4 reporter construct is the same base construct used in *Nandagopal et al., 2019*, with the modification of the 2xHS4 insulating elements flanking each side of the expression cassette. All of the Notch receptor piggyBac constructs were derived from the vector PB-CMV-MCS-EF1-Puro (System Biosciences), with changes made to the promoter (CMV changed to PGK or CAG) and selection marker (Puromycin to Neomycin), as well as insertion of the NotchECD-Gal4esn-T2A-H2B-mTurq2 sequence into the MCS. The Lfng and Lfng (D289E) sequences were cloned into the MCS of the original PB-CMV-MCS-EF1-Puro plasmid to create the Fringe constructs, while the IFP2.0 sequence was cloned into the same base vector as the L-Fringes, but with the Puro resistance gene replaced by the Neomycin resistance gene. Tet-OFF constructs were designed from the original pCW57.1-MAT2A plasmid obtained by Addgene. The sequence from the end of the TRE-tight promoter to the beginning of the Blast resistance gene promoter was removed and replaced with each of the activating Notch ligand sequences fused to a T2A-H2B-mCherry sequence or with an H2B-mCherry or NGFR-T2A-H2B-mCherry sequence in the case of the control plasmids. For the constitutively expressing ligand constructs, the TRE-tight promoter was also removed and replaced with the CBh promoter. The plasmids used to PCR the gRNA for the CRISPR-Cas9-mediated knockout of endogenous Notch2 and Jagged1 in C2C12 cells (see CRISPR section below) were the same plasmids used in *Nandagopal et al., 2019*.

### Cell culture and plasmid transfections

CHO-K1 cells (ATCC) were cultured in alpha MEM Earle's Salts (FUJIFILM Irvine Scientific) supplemented with 10% FBS (Avantor, VWR), and 1X Pen/Strep/L-glutamine (Thermo Fisher Scientific)

as previously described (*Elowitz et al., 2018*). Transfection of CHO-K1 cells was performed using Polyplus-transfection jetOPTIMUS DNA Transfection Reagent (Genesee Scientific) according to the manufacturer's instructions. Briefly, for cells plated in 24 wells the night before (to reach ~80% confluency at time of transfection), 500 ng of non-piggyBac DNA was used along with 0.5 µL of transfection reagent. For piggyBac constructs, 500 ng of DNA+100 ng of the Super piggyBac transposase (System Biosciences) was used. For generation of stable cell lines, cells were incubated in 0.5 mL media with DNA+transfection reagent overnight at 37°C, 5% $CO_2$ before changing media the next day. For transient transfections, cells were incubated with DNA+transfection reagent for only 4–6 hr before media change.

C2C12 cells (ATCC) were cultured in DMEM with high glucose, no glutamine (Thermo Fisher Scientific) supplemented with 20% FBS, 1X Pen/Strep/L-glutamine, and 1X sodium pyruvate (Thermo Fisher Scientific). Cells were split to ensure that stock cell cultures never reached more than 80–90% confluency. Plasmid transfection of C2C12 cells was performed using the same protocol as used for CHO cells mentioned above.

All cell lines were originally authenticated by the manufacturer, ATCC, through STR profiling. ATCC also certified the cells to be free from mycoplasma contamination. Subsequent testing for mycoplasma contamination was also performed by our lab using the InvivoGen MycoStrip test kit and protocol. All cell lines were found to be free of mycoplasma contamination at the time of experiments.

## Lentivirus production and infection

Lentivirus was produced using the ViraPower Lentiviral Expression System (Thermo Fisher Scientific). Briefly, 293FT producer cells in a T-25 flask were transfected with a pCW57.1 expression construct (1.3 µg DNA) along with a packaging plasmid mix consisting of pVSV-G, pLP1, and pLP2 in a 2:1:1 ratio (3.9 µg DNA total). 24 hr after transfection, cell media was changed with 4 mL of fresh media. 48 hr post-transfection, virus containing cell media was collected and centrifuged at 3k rpm for 15 min at 4°C to remove cell debris and filtered through a 0.45 µm PVDF filter (EMD Millipore). 200 µL unconcentrated viral supernatant was added to cells plated at 20,000/24 well the day before, in a total volume of 300 µL (100 µL media+200 µL virus) and incubated at 37°C, 5% $CO_2$. 24 hr post-infection, virus containing media was removed and replaced with fresh media. At 48 hr post-infection, cells were placed under selection with media containing 10 µg/mL blasticidin (Invivogen). After two-cell passages in selection media, cells with high mCherry expression were sorted (see Cell line construction section) and used to screen for clones.

## Cell line construction

CHO-K1 cells were engineered to produce Notch receiver cells, receiver cells with inducible ligand, and Notch ligand sender cells. Receiver cells were created by initial transfection of the 2xHS4-UAS-H2B-mCitrine-2xHS4 plasmid. After selection in 400 µg/mL Zeocin (Thermo Fisher Scientific), cells were placed into limiting dilution, and a single reporter clone was identified that activated well in response to transient expression of Gal4, and continued with. The reporter clone was then transfected with a chimeric Notch1 or Notch2 receptor, whose intracellular domain was replaced with a Gal4esn-T2A-H2B-mTurq2 sequence (Gal4esn = minimal Gal4 transcription factor). Transfected reporter cells were selected in 600 µg/mL Geneticin (Thermo Fisher Scientific) and placed into limiting dilution. A Notch1 or Notch2 receiver cell clone was chosen by its ability to demonstrate high mCitrine expression when plated on plate-bound ligands in culture. For receiver cells with inducible ligand, Tet-OFF-ligand-T2A-H2B-mCherry was added to the receiver cells by lentiviral infection (see Lentiviral production and infection section above). Cells were selected in 10 µg/mL blasticidin (Invivogen), and sorted (Sony MA900 Multi-Application Cell Sorter) for expression of Notch (mTurq2) and ligand (mCherry) in the absence of 4-epi-Tc. Sorted cells were placed into limiting dilution, and clones were selected that demonstrated good ligand induction range (mCherry levels) when treated with various amounts of 4-epi-Tc as well as expressed good levels of mTurq2 (Notch receptor). Sender cell lines, stably expressing each of the four ligands under control of the Tet-OFF system or constitutively activated by the CBh promoter (*Figure 1A*), were created by infection of CHO-K1 cells with Tet-OFF-ligand-T2A-H2B-mCherry lentivirus or CBh-ligand-T2A-H2B-mCherry lentivirus, respectively. After selection in 10 µg/mL blasticidin, sorting for high mCherry levels and placement in limiting dilution,

cell clones were chosen by their tunability of the Tet-OFF ligand with 4-epi-Tc or by the level of ligand expression in the case of constitutively expressed ligand.

C2C12 receivers, receivers with inducible ligand, and sender cell lines were constructed in the same manner as the CHO-K1 cells with one major difference. C2C12 wt cells were first depleted of endogenous Notch2 and Jagged1 by CRISPR-Cas9 knockout before any cell lines were made (see CRISPR section below).

## CRISPR-Cas9 knockout of endogenous C2C12 Notch2 and Jagged1

Endogenous Notch2 and Jagged1 genes were knocked out in C2C12 mouse myoblast cells using two rounds of transfection with RNPs (ribonucleoproteins) consisting of gRNAs targeting Notch2 and Jagged1 complexed with Cas9 protein, along with an empty plasmid containing the blasticidin resistance gene added to the transfection mixture. To make the RNPs, gRNA sequences were first placed into the pX330 CRISPR-Cas9 plasmid (*Cong et al., 2013*) as described in *Nandagopal et al., 2019*. The gRNA sequences were then PCR amplified with a forward primer containing a T7 promoter and a reverse primer containing a short pA tail:

> Notch2 gRNA primer sequences: (5' to 3')
> T7 mN2C2 F
> TCTACCTAATACGACTCACTATAGGGTGGTACTTGTGTGCC (T7 promoter)
> mN2C2 R
> AAAAGCACCGACTCGGTG
> Jagged1 gRNA primer sequences: (5' to 3')
> T7 mJ1C1 F
> TCTACCTAATACGACTCACTATAGCGGGTGCACTTGCG (T7 promoter)
> mN2C2 R
> AAAAGCACCGACTCGGTG

Resulting PCR products were transcribed using the Megashortscript T7 transcription kit (Thermo Fisher Scientific), following the manufacturer's directions. 125 ng of each gRNA was separately incubated with 250 ng of Cas9 protein (PNA Bio Inc) for 10 min at room temperature, after which, the two gRNA/Cas9 mixtures were combined and used to transfect cells along with plasmid containing blasticidin resistance. Transfected cells were incubated for 24 hr before transfection media was replaced with media containing 10 µg/mL blasticidin (Invivogen). Cells were incubated for 30 hr, after which, selection media was removed and replaced with fresh media without selection. After the selected cell population was grown to ~80% confluency, cells were placed into limiting dilution in 96-well plates, calculating for 1 cell/well. Single-cell clones were identified and grown for ~10–12 days before expanding. Screening of C2C12 clones for reduced surface expression of Notch2 and Jag1 enabled identification of clone 'Nkd' that showed a loss of Notch signaling with impaired upregulation of Notch target genes Hey1, HeyL, Notch1, Notch3, and Jag2, when cells were cultured on plated recombinant Dll1 (Dll1-ext-Fc) and analyzed by RNA-sequencing (*Figure 7—figure supplement 1A*). The C2C12-Nkd clone also showed loss of Notch2 protein by western blot (*Figure 7—figure supplement 1B*) and a loss of Jag1 mRNA by RT-PCR (*Figure 7—figure supplement 1C*). Western blot was performed as described in *Nandagopal et al., 2019*. For RT-PCR analysis, RNA from cell lines was extracted by using the RNeasy Mini Kit (QIAGEN) with the cell-lysate first being homogenized through a QIAshredder column (QIAGEN), per the manufacturer's directions. RNA was then reverse-transcribed with the iScript cDNA Synthesis Kit (Bio-Rad), and PCR was carried out using the Jagged1-specific primers (5' to 3'):

> mJ1 C1 F
> CCAAAGCCTCTCAACTTAGTGC
> mJ1 C1 R
> CTTAGTTTTCCCGCACTTGTGTTT

## RNA-sequencing data collection and analysis of C2C12 wt and Nkd clone

C2C12-Nkd or C2C12 wt cells were plated at 80,000 cells/well in a 12-well plates treated with recombinant ligand (2 µg/mL Dll1-ext-Fc in PBS) or PBS only (negative control, C2C12-Nkd only) in 10 µM DAPT (Notch signaling inhibitor) overnight in two biological replicates. The next morning, DAPT was washed out and cells were allowed to signal for 6 hr before cells were harvested for RNA extraction using QIAshredder columns (QIAGEN) and the RNeasy Mini Kit per the manufacturer's instructions (see next section for further details of the plated ligand assay). cDNA libraries were prepared according to standard Illumina protocols at the Millard and Muriel Jacobs Genetics and Genomics Laboratory at Caltech. SR50 sequencing (10 libraries/lane) with a sequencing depth of 20–30 million reads was performed on a HiSeq2500. TrimGalore was used to run Cutadapt to trim low-quality ends (with -q 28), to remove adapters, and to clip 3 bp from the 3' ends of reads after adapter/quality trimming, as well as to run FastQC for quality assessment. Data were then uploaded to Galaxy for subsequent processing. Reads were aligned to the Mouse Dec. 2011 (GRCm38/mm10) genome using HISAT2 with default parameters and transcript abundances were computed with StringTie using the GENCODE annotation of the mouse genome (GRCm38), version M22 (Ensembl 97), downloaded June 27, 2019.

## Analysis of publicly available, pre-processed CHO-K1 transcriptomic data from RNA-sequencing

We computed mRNA expression levels for Notch genes in CHO-K1 cells (*Supplementary file 1*) using processed RNA-sequencing data from CHO-K1 cells downloaded from the CHO gene expression visualization application (CGEVA) located at https://anksi.shinyapps.io/biosciences/ (*Singh et al., 2018*). Gene expression was given by the CGEVA application in units of normalized Log2-transformed counts per million reads mapped (CPM), with normalization procedures previously described by the authors (*Singh et al., 2018*). For Notch genes of interest, we averaged expression measured in two different samples from CHO-K1 wt cells, and converted Log2-transformed CPM values to CPM units by the transformation $CPM = 2^{\wedge}(Log_2(CPM))$.

## Characterization of Tet-OFF promoter behavior

To optimize pre-culture conditions in experiments using cell lines with Tet-OFF inducible ligands, we analyzed the dynamics and density dependence of transcription. First, we analyzed transcription dynamics with a qRT-PCR time course. CHO-K1 senders with integrated Dll1 or Dll4 ligands driven by the Tet-OFF promoter were seeded at a density that would reach confluence in a 24-well plate at the indicated time of collection. Seeded senders were induced to express ligand by reducing 4-epi-Tc concentration in the culture medium from 500 ng/mL to 5 or 10 ng/mL for Dll4 and Dll1 senders, respectively. Media was changed every 24 hr to replenish the proper 4-epi-Tc concentration, and cells were passaged every 72 hr, where applicable. At collection time, cells were spun down and RNA was extracted, followed by cDNA synthesis and qRT-PCR analysis. Ligand mRNA expression reached steady-state levels within 24 hr of 4-epi-Tc addition (*Figure 1—figure supplement 3B*).

Second, we analyzed the Tet-OFF promoter's dependence on cell density. 25,000 CHO-K1 Dll1 or Dll4 sender cells were seeded in a 24-well, 12-well, or 6-well plate (for high, medium, and low density, respectively) and induced to maximal ligand expression by removing 4-epi-Tc from the culture medium. Cells were collected 72 hr later and RNA expression was analyzed by qRT-PCR. Ligand expression from the Tet-OFF promoter showed a modest (<2-fold) dependence on cell culture density in the absence of 4-epi-Tc (*Figure 4—figure supplement 2*).

## Surface ligand isolation and quantification

We measured surface ligand levels in several CHO-K1 sender cell lines with the Pierce Cell Surface Protein Biotinylation and Isolation Kit following the manufacturer's protocol, followed by western blot analysis of the purified surface proteins. Western blot was performed as described in *Nandagopal et al., 2019*. Notch ligand protein was detected with anti-FLAG tag staining, while GAPDH and NAK ATPase staining served as negative and positive controls, respectively, for detection of surface protein isolation. Dll4 surface levels were undetectable despite cotranslational expression levels similar to a Dll1 cell line with positive signal (*Figure 2—figure supplement 2*), as others have observed in OP9 cells (*Shah et al., 2012*). However, this disparity could also reflect a potential dependence of surface

protein pulldown efficiency on the number of primary amines (lysines) available for biotinylation in the ECD (Dll1 has 17, while Dll4 has only 9). Similarly, Jag1's high extracellular lysine count (44) could explain, at least in part, why its surface abundance exceeded that of the other ligands. Note that although surface protein isolation successfully depleted the cytoplasmic control and enriched it for the surface protein control, it is possible that some portion of the detected Notch ligands is residual cytoplasmic protein.

## Signaling assays with flow cytometry readout

Various cellular assays enabled quantitative analysis of relative signaling activity mediated by different receptor-ligand pairs expressed in cis and/or trans (see *Figures 1D, 3A and F*). The general workflow for all assays was as follows (see also *Figure 1E*): Ligand expression was induced in either sender or receiver cells by reducing the 4-epi-Tc concentration in the culture medium to the desired level (*Supplementary file 2*), and receivers were incubated in the Notch signaling inhibitor DAPT to prevent reporter activation during this 48 hr preinduction phase. Media was changed after 24 hr to maintain the desired 4-epi-Tc concentration and 1 or 2 µM DAPT for CHO-K1 and C2C12 receivers, respectively. For all assays with siRNA knockdown, siRNAs were applied 7–10 hr after cell seeding such that cells were 30–50% confluent at time of transfection (see section 'siRNA transfections') and siRNAs were removed after an ~16 hr incubation. Plasmid transfections (when applicable) were performed immediately after siRNA media change (24 hr before starting an assay) when cells were ~80% confluent. Per 0.5 mL of medium in a 24-well, 500 ng of total plasmid was transfected including IFP2 plasmid only or 350 ng of an IFP2 plasmid cotransfected with 150 ng of a plasmid containing wt mouse Lfng or the Lfng D289E mutant (mutated catalytic aspartate, 'dLfng'). (Note that surface Notch1 levels differed by less than 25% between the dLfng- and Lfng-transfected samples (*Figure 1—figure supplement 4B*).) Media was changed (replacing proper 4-epi-Tc and DAPT concentrations) 4–6 hr after transfection. To start the signaling assay, receivers (and sender cells, if applicable) were trypsinized using 0.25% Trypsin without EDTA and plated at a total cell density to reach confluence after 24 hr, either in a monoculture, coculture, or on plated ligand (see assay subtype details below). Senders used for each coculture assay experiment are listed in *Supplementary file 3*. Cells were incubated for 22–24 hr in the proper 4-epi-Tc concentration (if applicable) but without DAPT to allow signaling, and with the IFP2 cofactor biliverdin for samples transfected with IFP2 plasmid. Cells were harvested by trypsinization and Notch activation was analyzed by flow cytometry. All assays were performed with at least three biological replicates. Procedural details specific to each assay subtype, including cell coculture ratios, are described below.

### Plated ligand assay

For assays with CHO-K1 cells, wells of a 48-well plate were coated with recombinant ligand (Bio-Techne/R&D Systems) diluted in PBS to a volume of 150 µL/well and incubated at room temperature for 1 hr with rocking. For experiments with C2C12 cells, 24-well plates were used and volumes were scaled up accordingly. Either PBS alone or IgG1 Fc was plated as a negative control. The ligand or control solution was removed, and receiver cells were trypsinized using 0.25% Trypsin without EDTA and plated at 50,000 cells/well to start the assay. Plated ligand concentrations used for each experiment are given in *Supplementary file 4*.

### Trans-activation assay

Excess sender cells (stable or preinduced Tet-OFF senders) or negative control senders expressing no ligand (CHO-K1 wt or C2C12-Nkd parental cells, corresponding to the receiver cell type) were cocultured with receiver cells (trypsinized in 0.25% Trypsin without EDTA) in 48-well, 24-well, or 12-well plates with a sender:receiver cell ratio of 5:1 or greater and a total cell density equivalent to 150,000 total cells/well in a 24-well plate. When receiver clones with Tet-OFF ligands were used, cis-ligand expression was suppressed with maximal [4-epi-Tc] (500–800 ng/mL). Signaling activity was analyzed by flow cytometry after 22–24 hr of signaling.

### Cis-activation assay

Ligand expression was induced in receiver cells as described above. 150,000 cells expressing no ligand (CHO-K1 wt or C2C12-Nkd parental cells, corresponding to the receiver cell type) were cocultured

with 5000 receiver cells (trypsinized in 0.25% Trypsin without EDTA) in 24-well plates or 12-well plates for CHO-K1 and C2C12-Nkd cells, respectively. Signaling activity was analyzed by flow cytometry after 22–24 hr of signaling.

### Cis-modulation assay

Ligand expression was induced in receiver cells as described above. 150,000 sender cells (stable or preinduced Tet-OFF senders) or negative control senders expressing no ligand (CHO-K1 wt or C2C12-Nkd parental cells, corresponding to the receiver cell type) were cocultured with 5000 receiver cells (trypsinized in 0.25% Trypsin without EDTA) in 24-well plates or 12-well plates for CHO-K1 and C2C12-Nkd cells, respectively. Signaling activity was analyzed by flow cytometry after 22–24 hr of signaling.

For *Figure 6* experiments only (the analysis of cis-inhibition dependence on trans-ligand identity), a slightly different culture format was used (10,000 receiver cells with 100,000 sender cells in a 48-well plate). The sender populations selected for this assay differed in their ligand expression levels (*Figure 6—figure supplement 1A and B*) and trans-activation efficiencies (*Figure 6—figure supplement 1C*). High-Dll1 and -Dll4 senders ('Dll1-L2' and 'Dll4-L2') showed similar cotranslational mCherry fluorescence. mCherry expression in high-Jag1 ('Jag1-L1') and medium-Jag2 ('Jag2-A') senders was 25% and 60% lower than in high-Dll1 senders, respectively. In the absence of cis-ligand expression, these senders activated Notch1 and Notch2 receivers with ≤2-fold differences in overall signaling activity, except for the Jag1-Notch1 combination, which failed to signal (*Figure 6—figure supplement 1C*), as expected (*Figure 2H*).

### Cis-+trans-activation assay

Ligand expression was induced in CHO-K1 receiver cells as described above. 150,000 receiver cells were trypsinized in 0.25% Trypsin without EDTA and plated in a 24-well plate. Receiver cells treated with siRNA knockdown and plasmid transfection were plated at a higher density of 300,000 per well to reach confluence during the course of the assay despite reduced cell viability following the transfections. Signaling activity was analyzed by flow cytometry after 22–24 hr of signaling.

### Cis-activation assay—ruling out preinduction signaling

To determine to what extent S2 cleavage from trans-signaling during the ligand preinduction phase contributes to Notch activity measured in the cis-activation assay, CHO-K1 Notch2-Dll1 and Notch2-Dll4 receivers cells were plated either at 29k in a 96-well plate (for dense conditions) or 25,000 in a six-well plate (for sparse conditions). All cells were plated in 1 µM DAPT with varying amounts of 4-epi-Tc (*Supplementary file 2*). 24 hr post-incubation, cells were trypsinized with 0.25% Trypsin without EDTA. Cells were then counted and replated at the same cell numbers, using the same plating conditions used to initially seed the cells. After another 24 hr of incubation, the CHO-K1 cells were trypsinized, counted, and plated along with CHO-K1 cells as described above. 22 hr post-incubation, cells were run on the flow cytometer to measure Notch activation. Positive controls were set up using 5000 receiver cells (treated with 500 ng/mL 4-epi-Tc) cultured with 150,000 Dll1-L1 sender cells. All conditions were performed and run in triplicate. See also *Figure 4C and D*.

## Soluble ligand binding assay and surface receptor quantification

CHO-K1 reporter cells and reporter cells with integrated Notch1 ('receiver' cells) were prepared with endogenous Fringe knockdown and Lfng or dLfng transfection (with IFP2 as a cotransfection marker) as described in the section above ('Signaling assays with flow cytometry readout'). Instead of beginning a signaling assay, cells were replated on non-tissue-culture-treated polystyrene plates from Cell-Star to reduce cell attachment to the bottom of the well, enabling detachment without trypsin. After a 24 hr incubation in medium supplemented with the IFP2 cofactor biliverdin, the ligand binding assay was carried out with slight modifications from protocols described previously (*Kakuda and Haltiwanger, 2017*; *Varshney and Stanley, 2017*). Cells were detached by pipetting and spun down for 5 min at 400×*g* at room temperature (same parameters used for all centrifugation steps), then blocked with blocking buffer (2% BSA+100 µg/mL CaCl$_2$ in 1X DPBS) for 15 min. Ligands were prepared by pre-clustering ligand-ext-Fc fragments (at 2× concentrations relative to the final concentrations given in *Supplementary file 4*) with Alexa Fluor 594-conjugated anti-human secondary antibodies (1:1000)

in blocking buffer for 1 hr in the dark at 4°C. After blocking, cells were spun down again, resuspended in 50 µL blocking buffer, and mixed with 50 µL 2× pre-clustered ligands, then incubated in the dark for 1 hr at 4°C. To evaluate Fringe effects on surface Notch levels, cells were incubated with 400 ng/mL PE-conjugated anti-Notch1 antibodies (instead of ligands) in blocking buffer at room temperature in the dark for 30 min. Cells were washed once by adding 1 mL blocking buffer before spinning down, aspirating buffer, and resuspending cells in 200 µL FACS Buffer for analysis by flow cytometry (see 'Flow cytometry analysis' section for FACS buffer composition).

These receptor staining data also enabled a comparison of receptor surface levels to cotranslational expression (H2B-mTurq2 fluorescence); see *Figure 1—figure supplement 1B*. Although receptor antibody binding strengths could differ, it appears unlikely that higher surface levels could explain most ligands' preferential activation of Notch2 over Notch1 (*Figure 2H*), since Notch2 levels were lower than Notch1 levels in both surface expression and cotranslational expression (*Figure 1—figure supplement 1B*).

## Density optimization for the cis-activation assay

For CHO cells, Notch1 or Notch2 receiver cells were trypsinized using 0.25% Trypsin without EDTA (Thermo Fisher Scientific). Receiver cells were plated sparsely at 2500, 5000, or 10,000 in a 24-well plate along with an equal number of CHO-K1 'high-Dll4' (Dll4-L1) sender cells for Notch1 or 'high-Dll1' (Dll1-L1) sender cells for Notch2, and surrounded by 150,000 CHO-K1 wt cells. Cell cocultures were incubated for ~22 hr and subsequently analyzed by flow cytometry to determine Notch activation (mCitrine levels) in the receiver cells. For positive controls, 5000 receiver cells were plated with 150,000 sender cells (Dll4 with Notch1 cells, and Dll1 with Notch2 cells) in order to achieve maximal activation of the receivers (used for signal normalization). 5000 receiver cells were plated with 150,000 CHO-K1 wt cells along with the gamma-secretase inhibitor DAPT (1 µM, Sigma) for use as negative controls. All experiments were performed in triplicate. See results in *Figure 4A*.

For C2C12 cells, the assay was performed similarly to the CHO-K1 cell assay with a few key changes. C2C12 Notch1 and Notch2 receiver cells were first plated at 140,000 cells per 12-well. After ~6 hr post-plating, the cells were transfected with siRNAs targeting residual endogenous mouse N1, N2, and N3 (*Figure 7—figure supplement 1D*), using Lipofectamine RNAiMAX Transfection Reagent (Thermo Fisher Scientific), following the manufacturer's instructions. Each siRNA was used at a final amount of 4 pmol, except for N2 siRNA which was used at 20 pmol. Cell media was changed 24 hr after siRNA transfection. 48 hr post-transfection, receiver cells were cocultured in a 12-well plate at 2500, 5000, or 10,000 with an equal number of C2C12 Dll1 sender c19 cells as well as with 150,000 C2C12-Nkd cells. Positive controls were set up with 5000 receiver cells+150,000 Dll1 sender c19 cells, and negative controls consisted of 5000 receiver cells+150,000 C2C12-Nkd cells along with 2 µM DAPT. See results in *Figure 7B*.

## siRNA transfections and qRT-PCR analysis

Cells were seeded in 24-well or 12-well plates at a density to reach 30–50% confluence at the time of transfection—either 7–10 hr or 24 hr after plating, depending on the experiment. For Fringe knockdown in CHO-K1 cells, 2 pmol each of siRNA targeting the endogenous Rfng and Lfng transcripts, or 4 pmol total negative control siRNA (Allstars negative control, QIAGEN), were transfected using the Lipofectamine RNAiMAX Transfection Reagent (Thermo Fisher Scientific) according to the manufacturer's instructions. For all assays with C2C12 cells, the same reagent was used to transfect cells with either negative control siRNA (32 pmol), N1+N2+N3 siRNA (4 pmol+20 pmol+4 pmol, respectively, plus 4 pmol control siRNA) or N1+N2+N3+Rfng siRNA (4 pmol+20 pmol+4 pmol+4 pmol, respectively) in a 12-well plate. C2C12-Nkd cells already exhibit reduced levels of Notch1 and Notch3 (*Figure 7—figure supplement 1A*); however, Notch1 and Notch3 knockdowns were performed to prevent their subsequent upregulation by signaling during the assay. The Notch2 siRNA knockdown was not essential for these experiments, but performed despite very low levels of Notch2 to begin with.

In all assays, siRNAs were incubated with cells for 16–24 hr before media change, and transfected cells were harvested for knockdown quantification by qRT-PCR or for use in signaling assays 24–36 hr after initial transfection. For analysis by qRT-PCR, cells were spun down by centrifugation at 1400 rpm for 3 min at room temperature. After supernatant removal, the cell pellets were stored at –80°C for

later RNA extraction using the RNeasy Mini Kit (QIAGEN) with the cell-lysate first being homogenized through a QIAshredder column (QIAGEN), per the manufacturer's directions, followed by cDNA synthesis with the iScript cDNA Synthesis Kit (Bio-Rad), and finally analyzed by qPCR using the qPCR primers in *Supplementary file 5*. qPCR was performed on a CFX96 Touch Real-Time PCR Detection System (Bio-Rad).

## Analysis of endogenous Fringe effects on signaling in CHO-K1 cells

CHO-K1 cells endogenously express low levels of Lfng and ~20-fold higher levels of Rfng (*Supplementary file 1*; *Singh et al., 2018*). To assess the effects of endogenous Fringes on receptor-ligand interactions, we expressed dLfng or Lfng while knocking down endogenous Rfng and Lfng with siRNA in CHO-K1 receiver cells (as described above), and compared these 'No Fringe' and 'high Lfng' conditions to wt levels of Fringe in a trans-activation assay with flow cytometry readout (*Figure 2—figure supplement 4*).

As a general trend, the magnitude of signaling in the presence of endogenous Fringes fell between the dLfng and Lfng signaling magnitudes. Jag1-Notch1 signaling was not potentiated by endogenous Fringes (as would be expected in the case of Rfng dominance), but showed 1.7-fold weaker with endogenous CHO-K1 Fringes than with dLfng, consistent with Lfng dominance (*Kakuda et al., 2020*; *Pennarubia et al., 2021*; *Yang et al., 2005*), despite the much lower expression of Lfng relative to Rfng (*Supplementary file 1*). (Note: Near signal saturating conditions, we observed very small (<1.3-fold) decreases in Delta-Notch signaling with endogenous Fringes compared with dLfng, suggesting the possibility of mild, non-specific effects of the siRNA or plasmid treatment in the two conditions.) Although endogenous Fringes showed overall weaker effects on signaling activity than transfected Lfng did, our results suggest that endogenous Lfng modestly weakens Jagged signaling and strengthens Dll1-Notch1 signaling. See also *Figure 2I*.

## Flow cytometry analysis pipeline

For analysis of cells by flow cytometry, cells were trypsinized in 0.05% or 0.25% Trypsin-EDTA (Thermo Fisher Scientific). Cells were resuspended in 1X FACS buffer: 1X Hanks Balanced Salt Solution (Thermo Fisher Scientific) supplemented with 2.5 mg/mL bovine serum albumin (Sigma-Aldrich) and 200 U/mL DNAse I. Resuspended cells were filtered through 40 μm cell strainers (Corning Inc, Corning, NY, USA) into U-bottom 96-well tissue culture-treated plates. Cells were analyzed on a Beckman Coulter Life Sciences CytoFLEX benchtop flow cytometer. Data were analyzed in Python using custom software according to the following workflow:

1. Cells were gated in a 2D plane of forward scatter (FSC) and side scatter (SSC) to select intact, singlet cells.
2. Cells were gated in a 2D plane of mTurq2 (PB450, A.U.) vs. SSC to separate out the +mTurq2 receiver cells from -mTurq2 senders or 'blank' parental cells (*Figure 1F*).
3. Plasmid-transfected cells were gated in the APC700 channel to select cells expressing the cotransfection marker IFP2 above background levels (*Figure 1G*). High IFP2 levels were excluded to avoid overexpression artifacts.
4. Receiver cells coexpressing ligand were gated into six logarithmically spaced bins of arbitrary mCherry (ECD) fluorescence units (*Figure 1B*). Expression levels above the highest mCherry bin were excluded due to overexpression artifacts observed with the control proteins H2B-mCherry and NGFR.
5. Compensation was applied to subtract mTurq2 signal leaking into the FITC channel.
6. If applicable, reporter activity, mCitrine (FITC, A.U.) fluorescence was normalized to cotranslational receptor expression by dividing mCitrine by the mTurq2 signal (PB450, A.U.). The resulting mCitrine/mTurq2 ratio is the 'signaling activity' (reporter activity per unit receptor). Signaling activity defined this way controls for variations in receptor expression across receiver clones and eliminates artifacts in mCitrine and mTurq2 fluorescence quantification that occur when binning cells on high cis-ligand expression (*Figure 1—figure supplement 5*).
7. If applicable, cotranslational cis-ligand expression was normalized to cotranslational receptor expression by dividing mCherry by the mTurq2 signal (PB450, A.U.). The resulting mCherry/mTurq2 ratio (cis-ligand expression per unit receptor) controls for slight variations in receptor expression when quantitatively comparing ligands' cis-inhibition efficiencies (*Figure 6A*).
8. Average bulk measurements for each sample were obtained by computing the mean signal across single-cell data for a given sample (and mCherry bin, if applicable). Cells treated

with different 4-epi-Tc levels were pooled as technical replicates after mCherry binning. A minimum of 100 cells were required during averaging; mCherry bins with too few cells did not generate a bulk data point.

Notes: (1) Despite sometimes heterogeneous cis-ligand expression, reporter activity vs. cis-ligand expression curves showed similar trends in CHO-K1 cells whether fluorescence values were averaged across cells from different wells sorted into the same bin of cis-ligand (mCherry) expression (*Figure 5A*) or across cells from the same well and 4-epi-Tc concentration, without mCherry binning (*Figure 5—figure supplement 2*). (2) For C2C12-Nkd cells, the fluorescence averaging method sometimes showed differences when values were averaged across cells from different wells sorted into the same bin of cis-ligand (mCherry) expression or across cells from the same well and 4-epi-Tc concentration, without mCherry binning. We selected averaging methods based on responses to expression of the NGFR control (*Figure 7C*). For Notch2, individual receivers were sorted into discrete bins of mCherry (A.U.) and the mCitrine/mTurq2 (signaling activity) and mCherry (cotranslational cis-ligand expression) fluorescence values were averaged across all cells in that bin. For Notch1, single-cell data were not sorted into mCherry bins, but averaged by 4-epi-Tc concentration, because mCherry binning yielded anomalous reporter activities with the NGFR control, possibly due to selection of abnormally large cells when sorting into the high mCherry bins.

9. Background subtraction was performed by subtracting 'leaky' reporter activity of the receiver (using high 4-epi-Tc concentrations to minimize cis-ligand expression, where applicable) in coculture with 'blank' senders (CHO-K1 wt or C2C12-Nkd parental cells, according to the receiver cell type).

10. Y-axis normalization was performed as described in each Figure caption. The term, 'max normalized', refers to signaling activities that were divided by trans-signaling in coculture with excess senders expressing relatively high ligand levels (with cis-ligand expression suppressed by high 4-epi-Tc, where applicable.) When signaling activities were background subtracted first, they are referred to as 'min-max normalized'. Note: the maximum trans-signaling activity used in these normalizations may not reflect the true reporter activity maximum, especially in cases where cis-activation exceeds trans-activation strength. Additionally, in *Figure 7D*, maximal trans-signaling used in normalization was relatively weak for Notch2 (only 1.3- to 2.2-fold above background).

11. In some cases, average signaling activity (from steps #6 and #8 above) was further normalized to average cotranslational ligand expression (mCherry, A.U.) to define a signaling strength metric (*Figure 2A*, *Figure 2—figure supplement 4B*). In such cases, saturated data points, defined as those with normalized signaling activity over 0.75 in both dLfng and Lfng conditions, were excluded. Only senders with mCherry levels <75% of the maximum sender expression were used, due to nonlinearities in normalized signaling activity with extremely high ligand expression. This simple normalization method reduced variance in normalized signaling activity, but does not reflect the independently observed ultrasensitivity with respect to ligand expression (*Figure 2F and G*).

Note: all uses of 'histogram' to describe single-cell fluorescence distributions refer to kernel density estimates.

## Statistical analysis

At least three biological replicates were used for each flow cytometry and qRT-PCR experiment, where biological replicates are distinct samples prepared separately, sometimes in parallel on the same day and sometimes on different days. Two biological replicates were used for RNA-sequencing. Least squares regressions to fit data to lines or Hill functions were computed using scipy.optimize. leastsq. Activating and repressing Hill functions were defined, respectively, as $y = bx^n / (K^n + x^n)$ and $y = bK^n / (K^n + x^n)$, where K denotes the EC50 and b denotes the curve maximum. We note that fits of activating responses to Hill functions could exhibit systematic errors in estimation of maximal signaling activity where observed responses are biphasic (e.g. Dll1 curves in *Figure 2D*) or do not fully saturate (e.g. Jag2 curves in *Figure 2D*). For the biphasic Dll1 curves, only data points in the increasing phase were fit to Hill functions.

Signaling strength metrics were based on the ligand concentration at which the fitted Hill function crossed a y-axis threshold (*Figure 2D and Figure 3B*). These thresholds were defined as the half-maximum signaling measured for each receptor (separately, due to differences in reporter dynamic

range) from *Figure 2D* trans-activation data with Jag2, which signaled most strongly to both receptors. However, Jag2 curves were not fully saturated for either receptor, and errors in the estimation of relative maximum signaling for the two receptors (used to define half-maximal thresholds) could potentially skew relative signaling strengths computed for Notch1 vs. Notch2.

For linear regressions to compute a fold-change between two conditions, y-intercepts were fixed to (0,0) (*Figure 2B*, *Figure 3G* , *Figure 3I*, *Figure 5—figure supplement 3*, *Figure 5—figure supplement 5*, *Figure 7A*). For statistically robust comparison of the relative trans-signaling strengths of Dll1 and Jag1 in C2C12-Nkd cells (*Figure 7A*), we pooled data from control and Rfng siRNA treatment conditions (Rfng is reported to potentiate Notch1 signaling but have no effect on Notch2 signaling for both Dll1 and Jag1; *Kakuda et al., 2020*).

All mean values and 95% confidence intervals were computed based on 10,000 bootstrap replicates with at least n=3 biological replicates. When bootstrapping confidence intervals on parameter estimates from fitting ligand dose-response curves to lines (*Figure 2—figure supplement 3*, *Figure 3—figure supplement 1*, *Figure 5—figure supplement 3*, *Figure 5—figure supplement 5*) or Hill functions (*Figure 2D*, *Figure 3B*, *Figure 6A*), bootstrapped datasets were constructed by sampling separately from biological replicates within each x-axis 'bin', defined as the mCherry fluorescence window, the plated ligand concentration, or the 4-epi-Tc concentration, as relevant.

p-Values were computed in two ways. To evaluate whether the correlation between two paired variables had a slope greater or less than 1, we used one-sided Wilcoxon signed-rank tests implemented in scipy—scipy.stats.wilcoxon. To test whether two sets of data points came from distributions with the same means, we used permutation testing following these steps:

1. Compute the true difference in means between the control and test datasets (of length n and m, respectively).
2. Pool the two datasets and scramble the order of data points.
3. Define the permutation sample by labeling the first n points from the scrambled data as 'control' data and the next m points as 'test' data.
4. Compute the difference in means between the control and test distributions from the permutation replicate vs. the original dataset.
5. Repeat many times. The p-value is the fraction of permutation replicates with a difference in means greater than the true difference in means.

p-Values were computed using at least 10,000 permutation replicates with one-sided differences in mean.

## Acknowledgements

RK, LS, and MBE conceived and designed the experiments RK and LS generated the cell lines and performed the experiments. RK and LS analyzed the experimental data. RK, LS, and MBE wrote the paper. We thank Irwin Bernstein for generously sharing the recombinant Dll1-ext-Fc ligand, and Igor Antoshechkin in the Millard and Muriel Jacobs Genetics and Genomics Laboratory for assistance with RNA-sequencing. We are grateful to Xun Wang and the Rothenberg Lab for providing the NGFR construct. We would also like to thank Ellen Rothenberg, David Sprinzak, Stephen Blacklow, Sandy Nandagopal, James Linton, Martin Tran, Jan Gregrowicz, Ronghui Zhu, Felix Horns, and Jacob Parres-Gold for discussions about this work and for critical feedback on this manuscript. This work was supported by the National Institutes of Health (NIH) (grant R01 HD7335C). RK was supported by an NIH Ruth L Kirschstein NRSA predoctoral fellowship (F31 HD100185). MBE is a Howard Hughes Medical Institute Investigator. This article is subject to HHMI's Open Access to Publications policy. HHMI lab heads have previously granted a nonexclusive CC BY 4.0 license to the public and a sublicensable license to HHMI in their research articles. Pursuant to those licenses, the author-accepted manuscript of this article can be made freely available under a CC BY 4.0 license immediately upon publication.

## Additional information

### Funding

| Funder | Grant reference number | Author |
|---|---|---|
| National Institutes of Health | R01 HD7335C | Rachael Kuintzle<br>Leah A Santat<br>Michael B Elowitz |
| National Institutes of Health | F31 HD100185 | Rachael Kuintzle |
| Howard Hughes Medical Institute | | Leah A Santat<br>Michael B Elowitz |

The funders had no role in study design, data collection and interpretation, or the decision to submit the work for publication.

### Author contributions

Rachael Kuintzle, Conceptualization, Data curation, Software, Formal analysis, Funding acquisition, Validation, Investigation, Visualization, Methodology, Writing - original draft, Project administration, Writing - review and editing; Leah A Santat, Resources, Validation, Investigation, Methodology, Project administration; Michael B Elowitz, Conceptualization, Supervision, Funding acquisition, Writing - original draft, Writing - review and editing

### Author ORCIDs

Rachael Kuintzle ⓘ https://orcid.org/0000-0002-1035-4983
Leah A Santat ⓘ https://orcid.org/0000-0003-0511-9740
Michael B Elowitz ⓘ https://orcid.org/0000-0002-1221-0967

Reviewer #1 (Public review): https://doi.org/10.7554/eLife.91422.3.sa1
Reviewer #2 (Public review): https://doi.org/10.7554/eLife.91422.3.sa2
Author response https://doi.org/10.7554/eLife.91422.3.sa3

# Additional files

### Supplementary files

Supplementary file 1. Notch gene expression in wild-type CHO-K1 cells. These expression values in counts per million reads mapped (CPM) were obtained from the CHO gene expression visualization application (CGEVA) database of CHO RNA-sequencing (RNA-seq) data, including from wild-type CHO-K1 cells, located at https://anksi.shinyapps.io/biosciences/ (*Singh et al., 2018*). Dll1, Dll4, Jag2, and Mfng were not available in the database. See also Methods.

Supplementary file 2. 4-epi-Tc concentrations used to induce ligand expression. Where Tet-OFF inducible sender or receiver cells were used but don't appear in the table above, 4-epi-Tc was used at a concentration of 500–800 ng/mL to fully suppress expression.

Supplementary file 3. Sender-receiver cell pairs used for coculture assays. Where multiple receivers are listed, they were cultured with the same senders but in separate wells (receivers were not mixed together). All experiments with CHO-K1 cells used wild-type cells as negative sending controls and experiments with C2C12-Nkd receivers used the blank C2C12-Nkd parental line as negative controls. Clone identifiers are in parentheses; polyclonal cell lines lack parentheses. See senders' ligand expression distributions in *Figure 1C*, *Figure 1—figure supplement 3A*, and *Figure 6—figure supplement 1*.

Supplementary file 4. Recombinant Notch ligands and concentrations.

Supplementary file 5. Primers for RT-PCR and qRT-PCR.

MDAR checklist

## Data availability

RNA sequencing data are available at NCBI via the GEO accession GSE233573. All raw and processed datasets (RNA-seq, qRT-PCR, and flow cytometry) and the custom Python scripts used to make figures and support conclusions in this paper are available at data.caltech.edu: https://doi.org/10.22002/gjjkn-wrj28.

The following datasets were generated:

| Author(s) | Year | Dataset title | Dataset URL | Database and Identifier |
|---|---|---|---|---|
| Kuintzle R, Elowitz MB, Santat L | 2024 | Diversity in Notch ligand-receptor signaling interactions: Data and Analysis Pipeline | https://doi.org/10.22002/gjjkn-wrj28 | CaltechDATA, 10.22002/gjjkn-wrj28 |
| Kuintzle R, Elowitz MB, Santat L | 2023 | Effect of depletion of Notch signaling genes in C2C12 cells on the activation of Notch pathway activity in response to stimulation by plated ligand | https://www.ncbi.nlm.nih.gov/geo/query/acc.cgi?acc=GSE233573 | NCBI Gene Expression Omnibus, GSE233573 |

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

# Appendix 1

## Appendix 1—key resources table

| Reagent type (species) or resource | Designation | Source or reference | Identifiers | Additional information |
|---|---|---|---|---|
| Gene (*Homo sapiens*) | Notch1 | GenBank | NCBI ID: 4851 | |
| Gene (*Homo sapiens*) | Notch2 | GenBank | NCBI ID: 4853 | |
| Gene (*Homo sapiens*) | Dll1 | GenBank | NCBI ID: 28514 | |
| Gene (*Homo sapiens*) | Dll4 | GenBank | NCBI ID: 54567 | |
| Gene (*Homo sapiens*) | Jag1 | GenBank | NCBI ID: 182 | |
| Gene (*Homo sapiens*) | Jag2 | GenBank | NCBI ID: 3714 | |
| Gene (*Homo sapiens*) | NGFR | GenBank | NCBI ID: 4804 | |
| Gene (*Mus musculus*) | Rfng | GenBank | NCBI ID: 19719 | |
| Gene (*Mus musculus*) | Lfng | GenBank | NCBI ID: 16848 | |
| Cell line (*Cricetulus griseus*) | CHO-K1 | ATCC | Cat# CCL-61 | |
| Cell line (*Cricetulus griseus*) | CHO Dll1 sender L1 | Derived from CHO-K1 | CHO-K1+CBh-Dll1-2xFLAG-T2A-H2B-mCherry - Sort level 1 (high mCherry) | |
| Cell line (*Cricetulus griseus*) | CHO Dll1 sender L4B | Derived from CHO-K1 | CHO-K1+CBh-Dll1-2xFLAG-T2A-H2B-mCherry - Sort level 4B (low mCherry) | |
| Cell line (*Cricetulus griseus*) | CHO Dll1 sender L4A | Derived from CHO-K1 | CHO-K1+CBh-Dll1-2xFLAG-T2A-H2B-mCherry - Sort level 4A (low mCherry) | |
| Cell line (*Cricetulus griseus*) | CHO Dll1 sender L4 | Derived from CHO-K1 | CHO-K1+CBh-Dll1-2xFLAG-T2A-H2B-mCherry - Sort level 4 (medium mCherry) | |
| Cell line (*Cricetulus griseus*) | CHO Dll1 sender L2 | Derived from CHO-K1 | CHO-K1+CBh-Dll1-2xFLAG-T2A-H2B-mCherry - Sort level L2 (high mCherry) | |
| Cell line (*Cricetulus griseus*) | CHO Dll4 sender L1 | Derived from CHO-K1 | CHO-K1+CBh-Dll4-2xFLAG-T2A-H2B-mCherry - Sort level 1 (high mCherry) | |
| Cell line (*Cricetulus griseus*) | CHO Dll4 sender L4B | Derived from CHO-K1 | CHO-K1+CBh-Dll4-2xFLAG-T2A-H2B-mCherry - Sort level 4B (medium mCherry) | |
| Cell line (*Cricetulus griseus*) | CHO Dll4 sender L4 | Derived from CHO-K1 | CHO-K1+CBh-Dll4-2xFLAG-T2A-H2B-mCherry - Sort level 4 (medium mCherry) | |
| Cell line (*Cricetulus griseus*) | CHO Dll4 sender L2 | Derived from CHO-K1 | CHO-K1+CBh-Dll4-2xFLAG-T2A-H2B-mCherry - Sort level 2 (medium mCherry) | |
| Cell line (*Cricetulus griseus*) | CHO Jag1 sender L1 | Derived from CHO-K1 | CHO-K1+CBh-Jag1-2xFLAG-T2A-H2B-mCherry - Sort level 1 (high mCherry) | |

*Appendix 1 Continued on next page*

*Appendix 1 Continued*

| Reagent type (species) or resource | Designation | Source or reference | Identifiers | Additional information |
|---|---|---|---|---|
| Cell line (*Cricetulus griseus*) | CHO Jag1 sender L2 | Derived from CHO-K1 | CHO-K1+CBh-Jag1-2xFLAG-T2A-H2B-mCherry - Sort level 2 (medium mCherry) | |
| Cell line (*Cricetulus griseus*) | CHO Jag1 sender L3 | Derived from CHO-K1 | CHO-K1+CBh-Jag1-2xFLAG-T2A-H2B-mCherry - Sort level 3 (low mCherry) | |
| Cell line (*Cricetulus griseus*) | CHO Jag2 sender A | Derived from CHO-K1 | CHO-K1+CBh-Jag2-2xFLAG-T2A-H2B-mCherry - Sort level A (high mCherry) | |
| Cell line (*Cricetulus griseus*) | CHO Jag2 sender AB | Derived from CHO-K1 | CHO-K1+CBh-Jag2-2xFLAG-T2A-H2B-mCherry - Sort level AB (medium mCherry) | |
| Cell line (*Cricetulus griseus*) | CHO Jag2 sender AA | Derived from CHO-K1 | CHO-K1+CBh-Jag2-2xFLAG-T2A-H2B-mCherry - Sort level AA (high mCherry) | |
| Cell line (*Cricetulus griseus*) | CHO Jag2 sender B | Derived from CHO-K1 | CHO-K1+CBh-Jag2-2xFLAG-T2A-H2B-mCherry - Sort level B (low mCherry) | |
| Cell line (*Cricetulus griseus*) | CHO Tet-OFF Dll1 sender G1 | Derived from CHO-K1 | CHO-K1+Tet-OFF Dll1-2xFLAG-T2A-H2B- mCherry G1 | |
| Cell line (*Cricetulus griseus*) | CHO Tet-OFF Dll4 sender 2D6 | Derived from CHO-K1 | CHO-K1+Tet-OFF Dll4-2xFLAG-T2A-H2B- mCherry 2D6 | |
| Cell line (*Cricetulus griseus*) | CHO Tet-OFF Jag1 sender 1G3 | Derived from CHO-K1 | CHO-K1+Tet-OFF Jag1-2xFLAG-T2A-H2B- mCherry 1G3 | |
| Cell line (*Cricetulus griseus*) | CHO Tet-OFF Jag2 sender 1H6 | Derived from CHO-K1 | CHO-K1+Tet-OFF Jag2-2xFLAG-T2A-H2B- mCherry 1H6 | |
| Cell line (*Cricetulus griseus*) | CHO reporter | Derived from CHO-K1 | CHO-K1+2xHS4-UAS-H2B-Citrine-2xHS4 H1 | |
| Cell line (*Cricetulus griseus*) | CHO N1 receiver 2D1 | Derived from CHO reporter | CHO-K1+2xHS4-UAS-H2B-Citrine-2xHS4+PB-PGK-N1ECD-Gal4esn- T2A-H2B-mTurq2 2-D1 | |
| Cell line (*Cricetulus griseus*) | CHO N2 receiver 2A4 | Derived from CHO reporter | CHO-K1+2xHS4-UAS-H2B-Citrine-2xHS4+PB-PGK-N2ECD-Gal4esn- T2A-H2B-mTurq2 2-A4 | |
| Cell line (*Cricetulus griseus*) | CHO N1-Tet-OFF Dll1 2G5 | Derived from CHO reporter | CHO-K1+2xHS4-UAS-H2B-Citrine- 2xHS4+Tet-OFF Dll1-2xFLAG-T2A-H2B-mCherry+PB-CAG-N1ECD-Gal4esn- T2A-H2B-mTurq2 2-G5 | |
| Cell line (*Cricetulus griseus*) | CHO N1-Tet-OFF Dll4 1G11 | Derived from CHO reporter | CHO-K1+2xHS4-UAS-H2B-Citrine- 2xHS4+Tet-OFF Dll4-2xFLAG-T2A-H2B-mCherry+PB-CAG-N1ECD-Gal4esn- T2A-H2B-mTurq2 1-G11 | |
| Cell line (*Cricetulus griseus*) | CHO N1-Tet-OFF Jag1 1E2 | Derived from CHO reporter | CHO-K1+2xHS4-UAS-H2B-Citrine- 2xHS4+Tet-OFF Jag1-2xFLAG-T2A-H2B-mCherry+PB-CAG-N1ECD-Gal4esn-T2A-H2B-mTurq2 1-E2 | |

*Appendix 1 Continued on next page*

*Appendix 1 Continued*

| Reagent type (species) or resource | Designation | Source or reference | Identifiers | Additional information |
|---|---|---|---|---|
| Cell line (*Cricetulus griseus*) | CHO N1-Tet-OFF Jag2 2E10 | Derived from CHO reporter | CHO-K1+2xHS4-UAS-H2B-Citrine- 2xHS4+Tet-OFF Jag2-2xFLAG-T2A-H2B-mCherry+PB-CAG-N1ECD-Gal4esn- T2A-H2B-mTurq2 2-E10 | |
| Cell line (*Cricetulus griseus*) | CHO N2-Tet-OFF Dll1 1C4 | Derived from CHO reporter | CHO-K1+2xHS4-UAS-H2B-Citrine- 2xHS4+Tet-OFF Dll1-2xFLAG-T2A-H2B-mCherry+PB-CAG-N2ECD-Gal4esn- T2A-H2B-mTurq2 1-C4 | |
| Cell line (*Cricetulus griseus*) | CHO N2-Tet-OFF Dll4 1B8 | Derived from CHO reporter | CHO-K1+2xHS4-UAS-H2B-Citrine- 2xHS4+Tet-OFF Dll4-2xFLAG-T2A-H2B-mCherry+PB-CAG-N2ECD-Gal4esn- T2A-H2B-mTurq2 1-B8 | |
| Cell line (*Cricetulus griseus*) | CHO N2-Tet-OFF Jag1 2B8 | Derived from CHO reporter | CHO-K1+2xHS4-UAS-H2B-Citrine- 2xHS4+Tet-OFF Jag1-2xFLAG-T2A-H2B-mCherry+PB-CAG-N2ECD-Gal4esn- T2A-H2B-mTurq2 2-B8 | |
| Cell line (*Cricetulus griseus*) | CHO N2-Tet-OFF Jag1 2B7 | Derived from CHO reporter | CHO-K1+2xHS4-UAS-H2B-Citrine- 2xHS4+Tet-OFF Jag1-2xFLAG-T2A-H2B-mCherry+PB-CAG-N2ECD-Gal4esn- T2A-H2B-mTurq2 2-B7 | |
| Cell line (*Cricetulus griseus*) | CHO N2-Tet-OFF Jag2 1A1 | Derived from CHO reporter | CHO-K1+2xHS4-UAS-H2B-Citrine- 2xHS4+Tet-OFF Jag2-2xFLAG-T2A-H2B-mCherry+PB-CAG-N2ECD-Gal4esn- T2A-H2B-mTurq2 1-A1 | |
| Cell line (*Cricetulus griseus*) | CHO N1-Tet-OFF H2B-mCherry | Derived from CHO reporter | CHO-K1+2xHS4-UAS-H2B-Citrine- 2xHS4+Tet-OFF H2B-mCherry+PB-CAG-N1ECD-Gal4esn- T2A-H2B-mTurq2 F4 | |
| Cell line (*Cricetulus griseus*) | CHO N2-Tet-OFF H2B-mCherry | Derived from CHO reporter | CHO-K1+2xHS4-UAS-H2B-Citrine- 2xHS4+Tet-OFF H2B-mCherry+PB-CAG-N2ECD-Gal4esn- T2A-H2B-mTurq2 G2 | |
| Cell line (*Cricetulus griseus*) | CHO N1-Tet-OFF NGFR-T2A- H2B-mCherry | Derived from CHO reporter | CHO-K1+2xHS4-UAS-H2B-Citrine- 2xHS4+Tet-OFF NGFR-T2A-H2B-mCherry+PB-CAG-N1ECD-Gal4esn- T2A-H2B-mTurq2 2-A6 | |
| Cell line (*Cricetulus griseus*) | CHO N2-Tet-OFF NGFR-T2A- H2B-mCherry | Derived from CHO reporter | CHO-K1+2xHS4-UAS-H2B-Citrine- 2xHS4+Tet-OFF NGFR-T2A-H2B-mCherry+PB-CAG-N2ECD-Gal4esn- T2A-H2B-mTurq2 1-H4 | |
| Cell line (*Cricetulus griseus*) | CHO Dll1 sender 2F10 | Derived from CHO-K1 | CHO-K1+PGK-Dll1-2xFLAG-T2A-H2B-mCherry 2F10 | Used for surface ligand quantification only |
| Cell line (*Cricetulus griseus*) | CHO Dll1 sender 2D12 | Derived from CHO-K1 | CHO-K1+PGK-Dll1-2xFLAG-T2A-H2B-mCherry 2D12 | Used for surface ligand quantification only |
| Cell line (*Cricetulus griseus*) | CHO Dll1 sender 1D12 | Derived from CHO-K1 | CHO-K1+PGK-Dll1-2xFLAG-T2A-H2B-mCherry 1D12 | Used for surface ligand quantification only |
| Cell line (*Cricetulus griseus*) | CHO Dll1 sender 1G1 | Derived from CHO-K1 | CHO-K1+PGK-Dll1-2xFLAG-T2A-H2B-mCherry 1G1 | Used for surface ligand quantification only |
| Cell line (*Cricetulus griseus*) | CHO Dll1 sender 2A10 | Derived from CHO-K1 | CHO-K1+PGK-Dll1-2xFLAG-T2A-H2B-mCherry 2A10 | Used for surface ligand quantification only |

*Appendix 1 Continued on next page*

*Appendix 1 Continued*

| Reagent type (species) or resource | Designation | Source or reference | Identifiers | Additional information |
|---|---|---|---|---|
| Cell line (*Cricetulus griseus*) | CHO Dll4 sender 1B9 | Derived from CHO-K1 | CHO-K1+PGK-Dll4-2xFLAG-T2A-H2B-mCherry 1B9 | Used for surface ligand quantification only |
| Cell line (*Cricetulus griseus*) | CHO Dll4 sender 1D12 | Derived from CHO-K1 | CHO-K1+PGK-Dll4-2xFLAG-T2A-H2B-mCherry 1D12 | Used for surface ligand quantification only |
| Cell line (*Cricetulus griseus*) | CHO Dll4 sender 2E5 | Derived from CHO-K1 | CHO-K1+PGK-Dll4-2xFLAG-T2A-H2B-mCherry 22E5F10 | Used for surface ligand quantification only |
| Cell line (*Cricetulus griseus*) | CHO Dll4 sender 1B7 | Derived from CHO-K1 | CHO-K1+PGK-Dll4-2xFLAG-T2A-H2B-mCherry 1B7 | Used for surface ligand quantification only |
| Cell line (*Cricetulus griseus*) | CHO Jag1 sender 2E3 | Derived from CHO-K1 | CHO-K1+PGK-Jag1-2xFLAG-T2A-H2B-mCherry 2E3 | Used for surface ligand quantification only |
| Cell line (*Cricetulus griseus*) | CHO Jag1 sender 1H11 | Derived from CHO-K1 | CHO-K1+PGK-Jag1-2xFLAG-T2A-H2B-mCherry 1H11 | Used for surface ligand quantification only |
| Cell line (*Cricetulus griseus*) | CHO Jag1 sender 1A6 | Derived from CHO-K1 | CHO-K1+PGK-Jag1-2xFLAG-T2A-H2B-mCherry 1A6 | Used for surface ligand quantification only |
| Cell line (*Mus musculus*) | C2C12 | ATCC | Cat# CRL-1772 | |
| Cell line (*Mus musculus*) | C2C12-Nkd | Derived from C2C12 | C2C12 with N2 and Jag1 CRISPR knockout | |
| Cell line (*Mus musculus*) | C2C12 reporter | Derived from C2C12-Nkd | C2C12-Nkd+2xHS4-UAS-H2B-Citrine- 2xHS4 B5 | |
| Cell line (*Mus musculus*) | C2C12 N1 receiver | Derived from C2C12 reporter | C2C12-Nkd+2xHS4-UAS-H2B-Citrine –2xHS4+PB-CAG-N1ECD-Gal4esn- T2A-H2B-mTurq2 1-A5 | |
| Cell line (*Mus musculus*) | C2C12 N2 receiver | Derived from C2C12 reporter | C2C12-Nkd+2xHS4-UAS-H2B-Citrine –2xHS4+PB-CAG-N2ECD-Gal4esn- T2A-H2B-mTurq2 c24 | |
| Cell line (*Mus musculus*) | C2C12 Dll1 sender c19 | Derived from C2C12-Nkd | C2C12-Nkd+CBh-Dll1-2xFLAG-T2A- H2B-mCherry c19 (high mCherry) | |
| Cell line (*Mus musculus*) | C2C12 Dll4 sender 2-H10 | Derived from C2C12-Nkd | C2C12-Nkd+CBh-Dll4-2xFLAG-T2A- H2B-mCherry 2-H10 (high mCherry) | |
| Cell line (*Mus musculus*) | C2C12 Jag1 sender c11 | Derived from C2C12-Nkd | C2C12-Nkd+CBh-Jag1-2xFLAG-T2A- H2B-mCherry c11 (high mCherry) | |
| Cell line (*Mus musculus*) | C2C12 N1-Tet-OFF NGFR | Derived from C2C12 reporter | C2C12+2xHS4-UAS-H2B-Citrine- 2xHS4+Tet-OFF NGFR-2xFLAG-T2A-H2B-mCherry+PB-CAG-N1ECD-Gal4esn- T2A-H2B-mTurq2 1-H6 | |
| Cell line (*Mus musculus*) | C2C12 N2-Tet-OFF NGFR | Derived from C2C12 reporter | C2C12+2xHS4-UAS-H2B-Citrine- 2xHS4+Tet-OFF NGFR-2xFLAG-T2A-H2B-mCherry+PB-CAG-N2ECD-Gal4esn- T2A-H2B-mTurq2 1-E9 | |
| Cell line (*Mus musculus*) | C2C12 N1-Tet-OFF Dll4 | Derived from C2C12 reporter | C2C12+2xHS4-UAS-H2B-Citrine- 2xHS4+Tet-OFF Dll4-2xFLAG-T2A-H2B-mCherry+PB-CAG-N1ECD-Gal4esn- T2A-H2B-mTurq2 2-D1 | |

*Appendix 1 Continued on next page*

*Appendix 1 Continued*

| Reagent type (species) or resource | Designation | Source or reference | Identifiers | Additional information |
|---|---|---|---|---|
| Cell line (*Mus musculus*) | C2C12 N2-Tet-OFF Dll4 | Derived from C2C12 reporter | C2C12+2xHS4-UAS-H2B-Citrine- 2xHS4+Tet-OFF Dll4-2xFLAG-T2A-H2B-mCherry+PB-CAG-N2ECD-Gal4esn- T2A-H2B-mTurq2 3-B12 | |
| Cell line (*Mus musculus*) | C2C12 N1-Tet-OFF Jag2 | Derived from C2C12 reporter | C2C12+2xHS4-UAS-H2B-Citrine- 2xHS4+Tet-OFF Jag2-2xFLAG-T2A-H2B-mCherry+PB-CAG-N1ECD-Gal4esn- T2A-H2B-mTurq2 1-E7 | |
| Cell line (*Mus musculus*) | C2C12 N2-Tet-OFF Jag2 | Derived from C2C12 reporter | C2C12+2xHS4-UAS-H2B-Citrine- 2xHS4+Tet-OFF Jag2-2xFLAG-T2A-H2B-mCherry+PB-CAG-N2ECD-Gal4esn- T2A-H2B-mTurq2 1-B12 | |
| Transfected construct (*Cricetulus griseus* and *Mus musculus*) | Super PiggyBac Transposase Expression Vector | System Biosciences | Cat# PB210PA-1 | Transposase used in all transfections with PiggyBac vectors |
| Transfected construct (*Cricetulus griseus* and *Mus musculus*) | pEV-2xHS4-UAS-H2B-Citrine-2xHS4-SV40-Zeocin | This paper | | Reporter for hNotch-Gal4 receptors in cell lines |
| Transfected construct (*Cricetulus griseus* and *Mus musculus*) | PB-CMV-MCS-EF1-Puro (PiggyBac vector) | System Biosciences | Cat# PB510B-1 | Base vector used to derive all PiggyBac constructs |
| Transfected construct (*Cricetulus griseus*) | PB-PGK-N1ECD-Gal4esn-T2A-H2B-mTurq2-SV40-Neomycin | This paper | | Notch1ECD- Gal4 synthetic receptor in CHO receiver cells |
| Transfected construct (*Cricetulus griseus*) | PB-PGK-N2ECD- Gal4esn-T2A-H2B-mTurq2-SV40-Neomycin | This paper | | Notch2ECD- Gal4 synthetic receptor in CHO receiver cells |
| Transfected construct (*Cricetulus griseus* and *Mus musculus*) | PB-CAG-N1ECD- Gal4esn-T2A-H2B-mTurq2-SV40-Neomycin | This paper | | Notch1ECD- Gal4 synthetic receptor in Tet-OFF cell lines and C2C12 receiver cells |
| Transfected construct (*Cricetulus griseus* and *Mus musculus*) | PB-CAG-N2ECD- Gal4esn-T2A-H2B-mTurq2-SV40-Neomycin | This paper | | Notch2ECD- Gal4 synthetic receptor in Tet-OFF cell lines and C2C12 receiver cells |
| Transfected construct (*Cricetulus griseus* and *Mus musculus*) | pCW57.1-MAT2A | Addgene | Cat# 100521 | Base vector used to derive all Tet-OFF constructs and constitutive ligand constructs |
| Transfected construct (*Cricetulus griseus*) | pCW57.1 Tet-OFF Dll1-2xFLAG-T2A-H2B-mCherry-PGK Blasticidin | This paper | | Repressible Dll1 ligand in cell lines. Used to make lentivirus. |
| Transfected construct (*Cricetulus griseus* and *Mus musculus*) | pCW57.1 Tet-OFF Dll4-2xFLAG-T2A-H2B-mCherry-PGK Blasticidin | This paper | | Repressible Dll4 ligand in cell lines. Used to make lentivirus. |
| Transfected construct (*Cricetulus griseus*) | pCW57.1 Tet-OFF Jag1-2xFLAG-T2A-H2B-mCherry-PGK Blasticidin | This paper | | Repressible Jag1 ligand in cell lines. Used to make lentivirus. |
| Transfected construct (*Cricetulus griseus* and *Mus musculus*) | pCW57.1 Tet-OFF Jag2-2xFLAG-T2A-H2B-mCherry-PGK Blasticidin | This paper | | Repressible Jag2 ligand in cell lines. Used to make lentivirus. |
| Transfected construct (*Cricetulus griseus*) | pCW57.1 Tet-OFF H2B-mCherry-PGK-Blasticidin | This paper | | Repressible H2B-mCherry used as negative control in cell lines. Used to make lentivirus. |

*Appendix 1 Continued on next page*

*Appendix 1 Continued*

| Reagent type (species) or resource | Designation | Source or reference | Identifiers | Additional information |
|---|---|---|---|---|
| Transfected construct (*Cricetulus griseus* and *Mus musculus*) | pCW57.1 Tet-OFF NGFR-T2A-H2B- mCherry-PGK -Blasticidin | This paper | | Repressible NGFR ligand used as negative control in cell lines. Used to make lentivirus. |
| Transfected construct (*Cricetulus griseus* and *Mus musculus*) | pCW57.1 CBh-Dll1-2xFLAG-T2A-H2B-mCherry-PGK-Blasticidin | This paper | | Constitutively expressed Dll1 ligand in cell lines. Used to make lentivirus. |
| Transfected construct (*Cricetulus griseus* and *Mus musculus*) | pCW57.1 CBh-Dll4-2xFLAG-T2A-H2B-mCherry-PGK-Blasticidin | This paper | | Constitutively expressed Dll4 ligand in cell lines. Used to make lentivirus. |
| Transfected construct (*Cricetulus griseus* and *Mus musculus*) | pCW57.1 CBh-Jag1-2xFLAG-T2A-H2B-mCherry-PGK-Blasticidin | This paper | | Constitutively expressed Jag1 ligand in cell lines. Used to make lentivirus. |
| Transfected construct (*Cricetulus griseus*) | pCW57.1 CBh-Jag2-2xFLAG-T2A-H2B-mCherry-PGK-Blasticidin | This paper | | Constitutively expressed Jag2 ligand in cell lines. Used to make lentivirus. |
| Transfected construct (*Cricetulus griseus*) | PB-CMV-Lfng- BGHpA-EF1-Puromycin | *LeBon et al., 2014*; https://doi.org/10.7554/eLife.02950. | | Constitutively expressed Lfng in cell lines treated with siRNA |
| Transfected construct (*Cricetulus griseus*) | PB-CMV- Lfng(D289E)-BGHpA-EF1-Puromycin | This paper *Luther et al., 2009*; https://doi.org/10.1074/jbc | | Constitutively expressed dLfng (D289E) in cell lines treated with siRNA. Sequence from *Luther et al., 2009* |
| Transfected construct (*Cricetulus griseus*) | PB-CMV7-IFP2.0- BGHpA-SV40-Neomycin | Addgene (for IFP2.0 gene) | Cat# 59427 | IFP2.0 gene cloned into piggyBac plasmid. Vector cotransfected with fringe plasmids. |
| Transfected construct (*Mus musculus*) | pX330 (CRISPR-Cas9 plasmid system) | *Nandagopal et al., 2019*; https://doi.org/10.7554/eLife.37880; *Cong et al., 2013*; https://doi.org/10.1126/science.1231143. | | Plasmid used to insert RNA guide sequences for use in CRISPR knockdown in C2C12 cells |
| Transfected construct (*Cricetulus griseus*) | PB-PGK-Dll1- 2xFLAG-T2A-H2B-mCherry-P2A-Hygromycin | This paper | | Constitutively expressed Dll1 ligand in CHO-K1 cell lines |
| Transfected construct (*Cricetulus griseus*) | PB-PGK-Dll4- 2xFLAG-T2A-H2B-mCherry-P2A-Hygromycin | This paper | | Constitutively expressed Dll4 ligand in CHO-K1 cell lines |
| Transfected construct (*Cricetulus griseus*) | PB-PGK-Jag1- 2xFLAG-T2A-H2B-mCherry-P2A-Hygromycin | This paper | | Constitutively expressed Jag1 ligand in CHO-K1 cell lines |
| Transfected construct (*Cricetulus griseus*) | PB-PGK-Jag2- 2xFLAG-T2A-H2B-mCherry-P2A-Hygromycin | This paper | | Constitutively expressed Jag2 ligand in CHO-K1 cell lines |
| Antibody | PE anti-human Notch1 MHN1-519 (Mouse monoclonal) | BioLegend | Cat# 352105 | Used in soluble ligand binding assay (1:400) |
| Antibody | PE anti-human Notch2 MHN2-25 (Mouse monoclonal) | BioLegend | Cat# 348303 | Used in soluble ligand binding assay (1:400) |
| Antibody | Goat anti-human IgG Alexa Fluor 594 (Goat polyclonal) | Thermo Fisher Scientific | Cat# A-11014 | Secondary antibody used to pre-cluster ligands in soluble ligand binding assay (1:1000) |
| Antibody | Anti-Notch2 (Rabbit monoclonal) | Cell Signaling Technology | Cat# 5732 | WB (1:800) |
| Antibody | Anti-GAPDH (Rabbit monoclonal) | Cell Signaling Technology | Cat# 2118 | WB (1:3000) |

*Appendix 1 Continued on next page*

*Appendix 1 Continued*

| Reagent type (species) or resource | Designation | Source or reference | Identifiers | Additional information |
|---|---|---|---|---|
| Antibody | Rabbit anti-FLAG M2 | Cell Signaling Technology | Cat# 14793 | WB (1:750) |
| Antibody | Mouse anti-Sodium Potassium ATPase Alpha 1 | Novus Biologicals | Cat# NB300-146 | WB (1:1000) |
| Antibody | Amersham ECL Rabbit IgG, HRP-linked whole Ab | Cytiva | Cat# NA934 | WB (1:2000) |
| Sequence-based reagent | siRNA: AllStars negative control | QIAGEN | Cat# 1027281 | |
| Sequence-based reagent | siRNA: Custom Select siRNA hamster Rfng | Thermo Fisher Scientific | Cat# 4399666 ID# s553138 | Sequence (5' to 3') GCUGUAAAAUGUCAGUGGAtt |
| Sequence-based reagent | siRNA: Custom Select siRNA hamster Lfng | Thermo Fisher Scientific | Cat# 4399665 ID# 555728 | Sequence (5' to 3') AGCUAAUGAUGAUAAGGGAtt |
| Sequence-based reagent | siRNA: Silencer Select siRNA mouse N1 | Thermo Fisher Scientific | Cat# 4390771 ID# s70699 | |
| Sequence-based reagent | siRNA: Custom Stealth siRNA mouse N2 | Thermo Fisher Scientific | Cat# 10620312 ID# 359602D12 | Sequence (5' to 3') GACCUUCACCCAUCCUGCAAGUUCA |
| Sequence-based reagent | siRNA: Stealth siRNA mouse N3 | Thermo Fisher Scientific | Cat# 1320001 ID# mss207111 | |
| Sequence-based reagent | siRNA: Silencer Select siRNA mouse Rfng | Thermo Fisher Scientific | Cat# 4390771 ID# s72908 | |
| Peptide, recombinant protein | Cas9 Protein | PNA Bio Inc | Cat# CP01 | |
| Peptide, recombinant protein | Recombinant Human Dll1ext-Fc fusion proteins | *Sprinzak et al., 2010*; https://doi.org/10.1038/nature08959. | | Kindly provided by Irwin Bernstein, MD at Fred Hutchinson Cancer Research Center |
| Peptide, recombinant protein | Recombinant Human IgG1 Fc Protein | Bio-Techne/ R&D Systems | Cat# 110-HG-100 | |
| Peptide, recombinant protein | Recombinant Human DLL1 Fc Chimera Protein | Bio-Techne/ R&D Systems | Cat# 10,184-DL-050 | |
| Peptide, recombinant protein | Recombinant Human DLL4 Fc Chimera Protein | Bio-Techne/ R&D Systems | Cat# 10185-D4-050 | |
| Peptide, recombinant protein | Recombinant Human Jag1 Fc Chimera Protein | Bio-Techne/ R&D Systems | Cat# 1277-JG-050 | |
| Peptide, recombinant protein | Recombinant Human Jag2 Fc Chimera Protein | Bio-Techne/ R&D Systems | Cat# 1726-JG-050 | |
| Chemical compound, drug | Lipfectamine RNAiMAX Transfection Reagent | Thermo Fisher Scientific | Cat# 13778075 | |
| Chemical compound, drug | Polyplus- transfection jetOPTIMUS DNA Transfection Reagent | Genesee Scientific | Cat# 55-250 | |
| Chemical compound, drug | DAPT | Sigma-Aldrich | Cat# D5942 | |
| Chemical compound, drug | 4-epi tetracycline hydrochloride | Sigma-Aldrich | Cat# 37918 | |
| Commercial assay or kit | Megashortscript T7 transcription kit | Thermo Fisher Scientific | Cat# AM1354 | |
| Commercial assay or kit | RNeasy Mini Kit | QIAGEN | Cat# 74104 | |

*Appendix 1 Continued on next page*

*Appendix 1 Continued*

| Reagent type (species) or resource | Designation | Source or reference | Identifiers | Additional information |
|---|---|---|---|---|
| Commercial assay or kit | QIAshredder | QIAGEN | Cat# 79656 | |
| Commercial assay or kit | iScript cDNA Sythesis Kit | Bio-Rad | Cat# 1708890 | |
| Commercial assay or kit | MycoStrip | InvivoGen | Cat# rep-mys-100 | |
| Commercial assay or kit | Pierce Cell Surface Protein Biotinylation and Isolation Kit | Thermo Fisher Scientific | Cat# A44390 | |
| Software, algorithm | RNA-sequencing read trimming and quality control | TrimGalore | https://www.bioinformatics.babraham.ac.uk/projects/trim_galore/ | |
| Software, algorithm | RNA-sequencing read alignment | HISAT2 v2.1.0 (via Galaxy) *Kim et al., 2019* | http://daehwankimlab.github.io/hisat2/; https://usegalaxy.org/ | |
| Software, algorithm | RNA-sequencing transcript abundance calculation | StringTie v1.3.4 (via Galaxy) | http://ccb.jhu.edu/software/stringtie/; https://usegalaxy.org/ | |
| Software, algorithm | Import of flow cytometry .fcs files for data processing in Python3 | FlowCal | https://flowcal.readthedocs.io/en/latest/; *Castillo-Hair et al., 2016* | |

