## [Editor Report · eLife Assessment]

This **valuable** study significantly enhances our understanding of how various ligands and receptors interact within the Notch signaling pathway. By developing novel cell-based assay systems, the authors systematically analyzed the effects of different ligand-receptor combinations on pathway activation. The **convincing** data reveal intriguing and unexpected differences and provide a foundation for interpreting Notch signalling in both normal and disease-related contexts.

---

## [Referee Report · Reviewer #1 (Public review)]

Summary:

The Notch signaling pathway plays important roles in many developmental and disease processes. Although well-studied there remain many puzzling aspects. One is the fact that as well as activating the receptor through a trans-activation, the transmembrane ligands can interact with receptors present in the same cell. These cis-interactions are usually inhibitory, but in some cases, as in the assays used here, they may also be activating. With a total of 6 ligands and 4 receptor there are potentially a wide array of possible outcomes when different combinations are co-expressed in vivo. Here the authors set out to make a systematic analysis of the qualitative and quantitative differences in the signaling output from different receptor ligand combinations, generating sets of "signaling" (ligand expressing) and "receiving" (receptor +/- ligand expressing cells).

The readout of pathway activity is transcriptional, relying on the fusion of GAL4 in the intracellular part of the receptor. Positive ligand interactions result in proteolytic release of Gal4 that turns on expression of H2B-citrine. As an indicator of ligand and receptor expression levels, they are linked via TA to H2B mCherry and H2B mTurq expression respectively. The authors also manipulate expression of the glycosyltransferase Lunatic-Fringe (LFng) that modifies the EGF repeats in the extracellular domains impacting on their interactions. The testing of multiple ligand receptor combinations at varying expression levels is a tour de force, with over 50 stable cell lines generated, and yields valuable insights although as a whole, the results are quite complex.

Strengths:

Taking a reductionist approach to test systematically differences in the signaling strength, binding strength and cis-interactions from the different ligands in the context of the Notch1 and Notch 2 receptors (they justify well they choice of players to test via this approach) produces a baseline understanding of the different properties and leads to some unexpected and interesting findings. Notably:

- Jag1 ligand expressing cells failed to activate Notch1 receptor although were capable of activating Notch2. Conversely, Jag2 cells elicited the strongest activation of both receptors. The results with Jag1 are surprising also because it exhibits some of the strongest binding to plate bound ligands. The failure to activate Notch1 has major functional significance and it will be important in future to understanding the mechanistic basis.

- Jagged ligands have the strongest ciis-inhibitory effects and the receptors differ in their sensitivity to cis-inhibition by Dll ligands. These observations are in keeping with earlier in vivo and cell culture studies. More referencing of those would better place the work in context but it nicely supports and extends previous studies that were conducted in different ways.

- Responses to most trans-activating ligands showed a degree of ultrasensitivity but this was not the case for cis-interactions where effects were more linear. This has implications for the way the two mechanisms operate and for how the signaling levels will be impacted by ligand expression levels.

- Qualitatively similar results are obtained in a second cell line, suggesting they reflect fundamental properties of the ligands/receptors.

Weaknesses:

One weakness is that the methods used to quantify the expression of ligands and receptors rely on co-translation of tagged nuclear H2B proteins. These may not accurately capture surface levels/correctly modified transmembrane proteins. In general, the multiple conditions tested partly compensate for the concerns - for example as Jag1 cells do activate Notch2 even if they do not activate Notch1 some Jag1 must be getting to the surface. But even with Notch2, Jag1 activities are on the lower side, making it important to clarify, especially given the different outcomes with the plated ligands. Similarly, is the fact that all ligands "signalled strongest to Notch2" an inherent property or due to differences in surface levels Notch 2 compared to Notch1?.. The results would be considerably strengthened by calibration of the ligand/receptor levels (and ideally their sub-cellular localizations). Assessing the membrane protein levels would be relatively straightforward to perform on som eof the basic conditions because their ligand constructs contain Flag tags, making it plausible to relate surface protein to H2B, and there are antibodies available for Notch1 and Notch2

In the revised version this has been addressed to some extent. A figure showing the relationship between co-translated mTurquiose and surface receptor expression for some clones (Figure 1-figure supplement 1B) goes some way to address the concerns that differences in Notch1 and Notch 2 could be due to the receptor levels. The data analyzing surface ligand levels is more equivocal, (a Western blot for biotinylated surface proteins), as the levels detected vary substantially between Dll1 and Dll4 (the latter barely detectable). But as a signal for surface expression of Jag1 was obtained this rules-out one concern that this ligand was failing to reach the surface. A discussion of the caveats of the approach is warranted, to make clear the limitations.

Cis-activation as a mode of signaling has only emerged from these synthetic cell culture assays raising questions about its physiological relevance. Cis-activation is only seen at the higher ligand (Dll1, Dll4) levels, how physiological are the expression levels of the ligands/receptors in these assays? Is it likely that this would make a major contribution in vivo? Is it possible that the cells convert themselves into "signaling" and "receiving" sub-populations within the culture by post-translational mechanism. Again some analysis of the ligand/receptors in the cultures would be a valuable addition to show whether or not there are major heterogeneities.

It is hard to appreciate how much cell to cell variability in the "output" there is. For example, low "outputs" could arise from fewer cells becoming activated or from all cells being activated less. As presented, only the latter is considered. That maybe already evident in their data, but not easy for the reader to distinguish from the way they are presented. For example, in many of the graphs, data have been processed through multiple steps of normalization. Some discussion/consideration this point is needed.

Impact:

Overall, cataloguing of the outcomes from the different ligand-receptor combinations, both in cis and trans, yields a valuable baseline for those investigating their functional roles in different contexts. There is still a long way to go before it will be possible to make a predictive model for outcomes based on expression levels, but this work gives an idea about the landscape and the complexities. This is especially important now that signaling relationships are frequently hypothesised based on single cell transcriptomic data. The results presented here demonstrate that the relationships are not straightforward when multiple players are involved.

---

## [Referee Report · Reviewer #2 (Public review)]

Summary:

In this manuscript the authors extend their previous studies on trans-activation, cis-inhibition (PMID: 25255098) and cis-activation (PMID: 30628888) of the Notch pathway. Here they create a large number of cell lines using CHO-K1 and C2C12 cells expressing either Notch1-Gal4 or Notch2-Gal4 receptors which express a fluorescent protein upon receptor activation (receiver cells). For cis-inhibition and cis-activation assays, these cells were engineered to express one of the four canonical Notch ligands (Dll1, Dll4, Jag1, Jag2) under tetracycline control. Some of the receiver cells were also transfected with a Lunatic fringe (Lfng) plasmid to produce cells with a range of Lfng expression levels. Sender cells expressing all of the canonical ligands were also produced. Cells were mixed in a variety of co-culture assays to highlight trans-activation, cis-activation, and cis-inhibition. All four ligands were able to trans-activate Notch1 and Notch 2, although Jag1 transactivated Notch1 weakly. Lfng enhanced trans-activation of both Notch receptors by Dll1 and Dll4, and inhibited both receptors by Jag 1 and Jag2. Cis-expression of all four ligands were predominantly inhibitory, but Dll1 and Dll4 showed strong cis-activation of Notch2. Interestingly, cis-ligands preferentially inhibited trans-activation by the same ligand, with varying effects on other trans-ligands.

Strengths:

This represents the most comprehensive and rigorous analysis of the effects of canonical ligands on cis- and trans-activation, and cis-inhibition, of Notch1 and Notch2 in the presence or absence of Lfng so far. Studying cis-inhibition and cis-activation is difficult in vivo due to the presence of multiple Notch ligands and receptors (and Fringes) that often occur in single cells. The methods described here are a step towards generating cells expressing more complex arrays of ligands, receptors and Fringes to better mimic in vivo effects on Notch function.

In addition, the fact that their transactivation results with most ligands on Notch1 and 2 in the presence or absence of Lfng were largely consistent with previous publications provides confidence that the author's assays are working properly.

Weaknesses:

In the original version, there was a major concern about quantifying the amount of Notch receptors and ligands on the cell surface (especially Jag1) based on total fluorescence. The authors have added data to demonstrate that most of the receptors and ligands are on the cell surface, allaying most of these concerns.

---

## [Author Response]

The following is the authors’ response to the original reviews.

**Reviewer #1 (Public Review):**

Summary:The Notch signaling pathway plays an important role in many developmental and disease processes. Although well-studied there remain many puzzling aspects. One is the fact that as well as activating the receptor through trans-activation, the transmembrane ligands can interact with receptors present in the same cell. These cis-interactions are usually inhibitory, but in some cases, as in the assays used here, they may also be activating. With a total of 6 ligands and 4 receptors, there is potentially a wide array of possible outcomes when different combinations are co-expressed in vivo. Here the authors set out to make a systematic analysis of the qualitative and quantitative differences in the signaling output from different receptor-ligand combinations, generating sets of "signaling" (ligand expressing) and "receiving" (receptor +/- ligand expressing cells).The readout of pathway activity is transcriptional, relying on the fusion of GAL4 in the intracellular part of the receptor. Positive ligand interactions result in the proteolytic release of Gal4 that turns on the expression of H2B-citrine. As an indicator of ligand and receptor expression levels, they are linked via TA to H2B mCherry and H2B mTurq expression respectively. The authors also manipulate the expression of the glycosyltransferase Lunatic-Fringe (LFng) that modifies the EGF repeats in the extracellular domains impacting their interactions. The testing of multiple ligand-receptor combinations at varying expression levels is a tour de force, with over 50 stable cell lines generated, and yields valuable insights although as a whole, the results are quite complex.Strengths:Taking a reductionist approach to testing systematically differences in the signaling strength, binding strength, and cis-interactions from the different ligands in the context of the Notch1 and Notch 2 receptors (they justify well the choice of players to test via this approach) produces a baseline understanding of the different properties and leads to some unexpected and interesting findings. Notably:- Jag1 ligand expressing cells failed to activate Notch1 receptor although were capable of activating Notch2. Conversely, Jag2 cells elicited the strongest activation of both receptors. The results withJag1 are surprising also because it exhibits some of the strongest binding to plate-bound ligands. The failure to activate Notch1 has major functional significance and it will be important in the future to understand the mechanistic basis.- Jagged ligands have the strongest cis-inhibitory effects and the receptors differ in their sensitivity to cis-inhibition by Dll ligands. These observations are in keeping with earlier in vivo and cell culture studies. More referencing of those would better place the work in context but it nicely supports and extends previous studies that were conducted in different ways.- Responses to most trans-activating ligands showed a degree of ultrasensitivity but this was not the case for cis-interactions where effects were more linear. This has implications for the way the two mechanisms operate and for how the signaling levels will be impacted by ligand expression levels.- Qualitatively similar results are obtained in a second cell line, suggesting they reflect fundamental properties of the ligands/receptors.

We appreciate the positive and constructive feedback.

Weaknesses:One weakness is that the methods used to quantify the expression of ligands and receptors rely on the co-translation of tagged nuclear H2B proteins. These may not accurately capture surface levels/correctly modified transmembrane proteins. In general, the multiple conditions tested partly compensate for the concerns - for example, as Jag1 cells do activate Notch2 even if they do not activate Notch1 some Jag1 must be getting to the surface. But even with Notch2, Jag1 activities are on the lower side, making it important to clarify, especially given the different outcomes with the plated ligands. Similarly, is the fact that all ligands "signalled strongest to Notch2" an inherent property or due to differences in surface levels of Notch 2 compared to Notch1? The results would be considerably strengthened by calibration of the ligand/receptor levels (and ideally their sub-cellular localizations). Assessing the membrane protein levels would be relatively straightforward to perform on some of the basic conditions because their ligand constructs contain Flag tags, making it plausible to relate surface protein to H2B, and there are antibodies available for Notch1 and Notch2.

We agree that mCherry fluorescence does not provide a direct readout of active surface ligand levels. As the reviewer points out, the ability of Jag1 to activate Notch2 demonstrates that expressed Jag1 is competent for signaling. Further, in some cases, Jag1-Notch2 activation can be comparable to Dll1-Notch2 activation (Figure 2A). Following the reviewer’s suggestion, we performed a Western blot for multiple expression levels for each of three surface ligands (Dll1, Dll4, Jag1) (Figure 2—figure supplement 2). This blot revealed a signal for surface expression of Jag1. Interpretation is complicated by the expected dependence of the efficiency of surface protein purification on the number of primary amines in the protein, which varies among these ligands, and qualitatively correlates with the staining intensity. While this makes quantitative interpretation difficult, this result further supports the notion that Jag1 is present on the cell surface. Finally, we note that high signaling activity need not, in general, directly correlate with surface expression levels. In fact, one study showed an example in which increased ligand activity occurred with decreased basal ligand surface levels (Antfolk et al., 2017). While one would ideally like to know all parameters of the system, including surface protein levels, rates of recycling, etc. the perspective taken here is that the net effect of these many post-translational processing steps can be subsumed into the overall relationship between the expression of the protein (which, in our case, is read out by the co-translational reporter) and its activity, which is relevant for the behavior of developmental circuits, among other systems. To address this comment, we now explicitly mention the limitation of mCherry as a proxy for surface protein, and add a reference to previous work highlighting the relationship between surface levels and ligand activity.

In terms of the dependence of signaling on Notch levels, the metric of signaling activity used here is explicitly normalized by the mTurquoise co-translational reporter of Notch expression to account for differences in receptor expression across receiver clones. We have added a new figure to show the variation in expression (Figure 1—figure supplement 1A) and to demonstrate this normalization (Figure 1—figure supplement 5). Having said that, as the reviewer correctly points out, we cannot directly address the dependence on surface receptor levels with mTurquoise alone. To address this comment, we have added a figure that shows cotranslational and surface receptor expression for a subset of our receiver clones (Figure 1—figure supplement 1B). Although antibody binding strengths may vary, it appears unlikely that higher surface levels could explain most ligands’ preferential activation of Notch2 over Notch1, since Notch2 levels were lower than Notch1 levels in both surface expression and cotranslational expression.

Cis-activation as a mode of signaling has only emerged from these synthetic cell culture assays raising questions about its physiological relevance. Cis-activation is only seen at the higher ligand (Dll1, Dll4) levels, how physiological are the expression levels of the ligands/receptors in these assays? Is it likely that this would make a major contribution in vivo? Is it possible that the cells convert themselves into "signaling" and "receiving" sub-populations within the culture by post-translational mechanism? Again some analysis of the ligand/receptors in the cultures would be a valuable addition to show whether or not there are major heterogeneities.

The cis-activation results in this paper are, as the reviewer points out, conducted in synthetic cell culture assays. Cis-activation is observed across a large dynamic range of ligand expression, possibly including non-physiologically high levels. However, our previous work (Nandagopal et al, eLife 2019) showed that cis-activation does not require over-expression, as it occurred in unmodified Caco-2 and NMuMG cells with their endogenous ligand and receptor expression levels. As shown here in Figure 4B, cis-activation for Notch2 increases monotonically and is substantial even at intermediate ligand concentrations. In other cases, cis-activation is maximal at intermediate concentrations. We agree that the in vivo role remains unclear, and is difficult to determine due to the typical close contacts among cells in tissues. Therefore, these assays do not speak to in vivo relevance. Note that we can, however, rule out the possibility of trans signaling between well-mixed cell populations at these densities (Figure 4A).

It is hard to appreciate how much cell-to-cell variability in the "output" there is. For example, low "outputs" could arise from fewer cells becoming activated or from all cells being activated less. As presented, only the latter is considered. That may be already evident in their data, but not easy for the reader to distinguish from the way they are presented. For example, in many of the graphs, data have been processed through multiple steps of normalization. Some discussion/consideration of this point is needed.

We agree that in different experiments changes in a mean response can reflect changes in fraction of activated cells, or level of activation or some combination of both. In this work, most assays were conducted by flow cytometry, which provides a full distribution of cellular responses. We provided distributions for some experiments in the supplementary figures (i.e., Figure 4—figure supplement 1, and Figure 5—figure supplement 4). The sheer number of experiments and samples prevents us from displaying all underlying histograms. Therefore, we have provided all flow data sets in an extensive archive that is publicly available on data.caltech.edu (https://doi.org/10.22002/gjjkn-wrj28).

Impact:Overall, cataloging the outcomes from the different ligand-receptor combinations, both in cis and trans, yields a valuable baseline for those investigating their functional roles in different contexts. There is still a long way to go before it will be possible to make a predictive model for outcomes based on expression levels, but this work gives an idea about the landscape and the complexities. This is especially important now that signaling relationships are frequently hypothesized based on single-cell transcriptomic data. The results presented here demonstrate that the relationships are not straightforward when multiple players are involved.

We appreciate this concise impact summary, and agree with its conclusions.

**Reviewer #2 (Public Review):**
Summary:In this manuscript, the authors extend their previous studies on trans-activation, cis-inhibition (PMID: 25255098), and cis-activation (PMID: 30628888) of the Notch pathway. Here they create a large number of cell lines using CHO-K1 and C2C12 cells expressing either Notch1-Gal4 or Notch2-Gal4 receptors which express a fluorescent protein upon receptor activation (receiver cells). For cis-inhibition and cis-activation assays, these cells were engineered to express one of the four canonical Notch ligands (Dll1, Dll4, Jag1, Jag2) under tetracycline control. Some of the receiver cells were also transfected with a Lunatic fringe (Lfng) plasmid to produce cells with a range of Lfng expression levels. Sender cells expressing all of the canonical ligands were also produced. Cells were mixed in a variety of co-culture assays to highlight trans-activation, cis-activation, and cis-inhibition. All four ligands were able to trans-activate Notch1 and Notch 2, except Jag1 did not transactivate Notch1. Lfng enhanced trans-activation of both Notch receptors by Dll1 and Dll2, and inhibited Notch1 activation by Jag2 and Notch2 activation by both Jag 1 and Jag2. Cis-expression of all four ligands was predominantly inhibitory, but Dll1 and Dll4 showed strong cis-activation of Notch2. Interestingly, cis-ligands preferentially inhibited trans-activation by the same ligand, with varying effects on other trans-ligands.Strengths:This represents the most comprehensive and rigorous analysis of the effects of canonical ligands on cis- and trans-activation, and cis-inhibition, of Notch1 and Notch2 in the presence or absence of Lfng so far. Studying cis-inhibition and cis-activation is difficult in vivo due to the presence of multiple Notch ligands and receptors (and Fringes) that often occur in single cells. The methods described here are a step towards generating cells expressing more complex arrays of ligands, receptors, and Fringes to better mimic in vivo effects on Notch function.In addition, the fact that their transactivation results with most ligands on Notch1 and 2 in the presence or absence of Lfng were largely consistent with previous publications provides confidence that the author's assays are working properly.

We appreciate the thoughtful comments and feedback.

Weaknesses:It was unusual that the engineered CHO cells expressing Notch1-Gal4 were not activated at all by co-culture with Jag1-expressing CHO cells. Many previous reports have shown that Jag1 can activate Notch1 in co-culture assays, including when Notch1 was expressed in CHO cells. Interestingly, when the authors used Jag1-Fc in a plate coating assay, it did activate Notch1 and could be inhibited by the expression of Lfng.

In our assays, we do in fact also see some signaling of Jag1 to Notch1, especially when dLfng is coexpressed (Figure 2—figure supplement 4, formerly Figure 2—figure supplement 3). While these levels are lower than those observed for other ligand-receptor combinations, they are significantly elevated compared to baseline. In specific natural contexts, it will be important to determine whether the weak but non-zero Jag1-Notch1 signaling acts negatively to suppress signaling from other ligands, or provides weak but potentially functionally important levels of signaling. Evidence for both modes exists in the literature. To address this, we have expanded the discussion of Jag1-Notch1 signaling and added references to other work on Jag1-Notch1 signaling to the Discussion section.

The cell surface level of the ligands was determined by flow cytometry of a co-translated fluorescent protein. Some calibration of the actual cell surface levels with the fluorescent protein would strengthen the results.

This issue was also raised by Reviewers #1 and #3. Please see responses to Reviewer #1, above.

**Reviewer #3 (Public Review):**
Summary:This manuscript reports a comprehensive analysis of Notch-Delta/Jagged signaling inclusive of the human Notch1 and Notch2 receptors and DLL1, DLL4, JAG1, and JAG2 ligands. Measurementsencompassed signaling activity for ligand trans-activation, cis-activation, cis-inhibition, and activity modulation by Lfng. The most striking observations of the study are that JAG1 has no detectable activity as a Notch1 ligand when presented on a cell (though it does have activity when immobilized on a surface), even though it is an effective cis-inhibitor of Notch1 signaling by other ligands, and that DLL1 and DLL4 exhibit cis-activating activity for Notch1 and especially for Notch2. Notwithstanding the artificiality of the system and some of its shortcomings, the results should nevertheless be a valuable resource for the Notch signaling community.Strengths:(1) The work is systematic and comprehensive, addressing questions that are of importance to the community of researchers investigating mammalian Notch proteins, their activation by ligands, and the modulation of ligand activity by LFng.(2) A quantitative and thorough analysis of the data is presented.Weaknesses:(1) The manuscript is primarily descriptive and does not delve into the underlying, mechanistic origin or source of the different ligand activities.

We agree that the goals of this paper were largely to discover the range of signaling modes that occur. A mechanistic analysis would be beyond the scope of this work, but we agree it is an important next step.

(2) The amount of ligand or receptor expressed is inferred from the flow cytometry signal of a co-translated fluorescent protein-histone fusion, and is not directly measured. The work would be more compelling if the amount of ligand present on the cell surface were directly measured with anti-ligand antibodies, rather than inferred from measurements of the fluorescent protein-histone fusion.

This issue was also raised by Reviewers #1 and #2. Please see responses to Reviewer #1, above.

(3) It would be helpful to see plots of the raw activity data before transformation and normalization, because the plots present data after several processing steps, and it is not clear how the processed data relate to the original values determined in each measurement.

We included examples showing how raw data is processed in Figure 4—figure supplement 1 and Figure 5—figure supplement 4. The sheer number of experiments precludes including similar figures for all data sets. However, all raw and processed data and data analysis code is publicly available at (https://doi.org/10.22002/gjjkn-wrj28).

(4) The authors use sparse plating of engineered cells with parental (no ligand or receptor-expressing cell to measure cis activation). However, the cells divide within the cultured period of 22-24 h and can potentially trans-activate each other.

If measured cis-activation signal arises solely from trans-activation, then the measured cis-activation signal per cell should increase with cell density, since trans-activation per cell does depend on cell density (Figure 4A). However, for the strongest cis-activators (Dll1- and Dll4-Notch2), signaling magnitude is similar when these cells are cultured sparsely or at confluence, which would otherwise allow efficient trans signaling (Figure 5A). Thus, for Dll1- and Dll4-Notch2 receivers, total signaling strength per cell depends little or not at all on the opportunity to signal intercellularly. Moreover, cis-activation signal for the Dll1- and Dll4-Notch2 combinations exceeded the maximum trans-signaling levels we could achieve for the same receivers when cis-ligand was suppressed (Figure 4B). These results argue that cis interactions dominate signaling in this context. However, we have not ruled out the possibility that trans-signaling between sister cells after division contributes to the comparatively weak cis-activation observed for Notch1 receivers.

**Reviewer #1 (Recommendations For The Authors):**
As outlined in the public review, there is a question of whether the nuclear H2B accurately reflects the surface levels of the transmembrane proteins (ligand and receptor). Clearly, it would not be feasible to check levels in all of the experimental conditions, but some baseline conditions should be analyzed.

We addressed this above.

**Reviewer #2 (Recommendations For The Authors):**
(1) As mentioned above, it was unusual that Jag1 did not activate Notch1 in co-culture assays, but did activate Notch1 in plate-coating assays. The authors should add some text to the Discussion to explain why they think this is happening in their engineered cells. One possibility is that the CHO cells express Manic fringe (Mfng) which is known to reduce Jag1-Notch1 activation. Data for Mfng levels in CHO cells were not included in Supplemental Table 2. Knocking down all three Fringes in CHO cells might increase Jag1-Notch1 activation.

This is already addressed in a sentence in the results: “Strikingly, while Jag1 sender cells failed to activate Notch1 receivers above background (Figure 2D), plate-bound Jag1-ext-Fc activated Notch1 only ~3-fold less efficiently than it activated Notch2 (Figure 3B-D). This suggests that the natural endocytic activation mechanism, or potential differences in tertiary structure between the expressed and recombinant Jag1 extracellular domains, could play roles in preventing Jag1-Notch1 signaling in coculture.” Regarding the point about Mfng, we added a note to Supplementary Table about other CHO-K1 expression data.

(2) Figure 1-supplemental figure 1: Both the Notch1-Jag1 and Notch1-Jag2 cells show high expression of Jag1 in low 4epi, but any higher concentration reduces to control levels. How much of a problem is this for interpreting your data?

This was not the ideal behavior, but by binning cells by co-translational reporters for ligand expression, we were able to obtain enough cells in intermediate bins. (Note: Figure 1—figure supplement 1 is now Figure 1—figure supplement 2.)

(3) Figure 1C legend: Are these stably-expressing cells or Tet-off cells? Please state in legend.

The figure legend has been updated.

(4) Figure 1E: How long is the knockdown of Rfng and Lfng effective? Does it affect the expression of Lfng later?

siRNA effects generally last for at least 72-96 hours, so we do not anticipate this being an issue.

(5) Page 9: "Lfng significantly decreased trans-activation of both receptors by Jag1 (>2.5-fold)". If there is no Jag1-Notch1 activation, how can Lfng decrease trans-activation?

We added a note in the main text to clarify that while Jag1-Notch1 signaling is relatively low, it can still be detectably decreased.

(6) Figure 4A legend: Please define what "2.5k ea senders and Rec" means. In the text, it says "To focus on cis-interactions alone, we then cultured receiver cells at low density, amid an excess of wildtype CHO-K1 cells" (page 14).

This was clarified in the text.

(7) Page 14: "By contrast, Notch2 was cis-activated by both Dll1 and Dll4, to levels exceeding those produced by trans-activation by high-Dll1 senders (Figure 4B, lower left)." Where is the trans-activation data? 4B, lower right?

We updated this reference in the main text.

(8) Page 16: "For Notch2-Dll1 and Notch2-Dll4, single cell reporter activities correlated with cis-ligand expression, regardless of whether cells were pre-induced at a high or low culture density (Figure 4D)." It appears that Notch2-Dll1 has lower Notch activation at sparse culture than confluent.

We agree that the level signaling is lower in sparse compared to confluent on average. This is explained by the sensitivity of the Tet-OFF promoter to culture density (Figure 4—figure supplement 2). However, the key point of this experiment is the positive correlation, which is consistent with cis-activation, and inconsistent with the pre-generation of NEXT hypothesis diagrammed in Figure 4C, which would not be expected to produce such a correlation.

(9a) For the creation of the C2C12-Nkd cells: Has genomic sequencing been done to confirm editing of Notch2 and Jag1 loci?

We confirmed the knockdown but did not do genomic sequencing.

(9b) The gel in Figure 7-Supplement 1C is not adequate for showing loss of Jag1. It should be repeated.

In this case, we have only the single gel. We added a note in figure legend that no duplicate was performed.

(10) Figure 7A: Which Fringes are expressed in C2C12 cells? You should provide a rationale for knocking down just Rfng.

Figure 7—figure supplement 1A shows the levels of expression in C2C12. Note that Mfng is not highlighted because its levels were undetectable.

(11) Figure 7-Supplement 1D: This is confusing. Notch2 levels are not reduced in the left panel, and Notch1 and Notch2 levels are not reduced in the right panel?

C2C12-Nkd cells exhibit reduced levels of Notch1 and Notch3. This can be seen in Figure 7—figure supplement 1A. Panel D presents the results of additional siRNA knockdown, performed to prevent subsequent up-regulation of Notch1 and Notch3 during the assay. These knockdown results were variable, as shown. The Notch2 siRNA knockdown was not essential for these experiments, but performed despite very low levels of Notch2 to begin with. In the revision, we have added this note to the Methods.

**Reviewer #3 (Recommendations For The Authors):**
(1) The results section of the manuscript is very dense and difficult to follow, as are the figure legends.

We appreciate the criticism, and regret that it is not easier to read in its current form.

(2) The authors could emphasize areas of concordance with published results (where available) to place their artificial, engineered system into a better biological context. Are there any examples of studies in whole organisms where cis-activation plays a role?

We are not aware of examples of cis-activation in whole organisms at this point.

(3) How do the authors rationalize the different responses of Notch1 to cell-presented Jag1 as opposed to immobilized Jag1, where its signal strength is second in rank order on a molar basis?

This comment was addressed above in response to the first recommendation from Reviewer #2.

It is also difficult to understand Figure 2_—_figure Supplement 3B, in which it appears that Jag1 induces a Notch1 reporter response when LFng is knocked down (dLfng), and how those data relate to the inactive response to Jag1 shown in the main figures.

The issue here is a difference of normalization. Figure 2A in the main text is normalized to the sender expression level, i.e. relative signaling strength. By contrast, Figure 2—figure supplement 4B (previously Figure 2—figure supplement 3B) shows absolute signaling activity, which can appear higher because it does not normalize for ligand expression. For Jag1-Notch1 signaling in particular, substantial signaling required very high levels of Jag1. We have added a new figure to demonstrate these two types of normalization (Figure 2—figure supplement 1A).

See the Authr response image 1 below for a direct comparison of these two normalization modes using data from both Figure 2A and Figure 2—figure supplement 4B. Note how the Jag1-Notch1 signaling activities that are nonzero in the top plot go to zero in the bottom plot as a result of normalizing the values to ligand expression.

**Author response image 1.**

Comparison of normalization modes in Figure 2A and Figure 2—figure supplement 4B (formerly 3B).

Normalized trans-activation signaling activities for different ligand-receptor combinations (with dLfng only), either with further normalization to ligand expression (bottom row) or without further normalization (top row). Normalized signaling activity is defined as reporter activity (mCitrine, A.U.) divided by cotranslational receptor expression (mTurq2, A.U.), normalized to the strongest biological replicate-averaged signaling activity across all ligand-receptor-Lfng combinations in this experiment. Saturated data points, defined here as those with normalized signaling activity over 0.75 in both dLfng and Lfng conditions, were excluded. Colors indicate the identity of the trans-ligand expressed by cocultured sender cells. Error bars denote bootstrapped 95% confidence intervals (Methods), in this case sampled from the number of biological replicates given in the legend—n1 (for Notch1) or n2 (for Notch2). See Methods and Figure 2A caption for more details. Note that the only difference between this figure and the new Figure 2—figure supplement 1A is that this figure additionally includes the Jag1-high data from Figure 2—figure supplement 4B.